EMBO
Molecular Medicine

# A novel modulator of IL-6R prevents inflammation-induced preterm birth and improves newborn outcome

France Côté [ID][1,2], Elizabeth Prairie [ID][1,2], Estefania Marin Sierra[2,3], Christiane Quiniou[2], Tiffany Habelrih[1,2], Wendy Xu[4], Béatrice Ferri [ID][1,2], Xin Hou[2], Isabelle Lahaie[2], Nadia Côté [ID][5], Sarah-Eve Loiselle[1,2], Laurence Gobeil[5], Kevin Sawaya[6], Aurélie Faucher [ID][5], Amélie Beaulieu [ID][1], Sandrine Delisle[7], Marie-Pénélope Simard [ID][1], Mohammad Ali Mohammad Nezhady [ID][2,8], Véronique Laplante[1], Allan Reuben[9], Sidi Mohamed Kalaidji[2], Emmanuel Bajon[2], Gael Cagnone [ID][2], Kelycia B Leimert [ID][4], Jean-François Gauchat[1], Luc Gaudreau[5], Sarah Robertson [ID][10], William D Lubell [ID][11], David M Olson [ID][4] & Sylvain Chemtob[1,2][✉]

## Abstract

**Preterm birth (PTB) is a major cause of neonatal mortality and morbidity. Evidence supports a determinant role for interleukin-6 (IL-6) in the pathophysiology of PTB. Our group developed a small peptide, HSJ633, that antagonizes the interleukin-6 receptor (IL-6R). Binding assays performed on HEK-Blue IL-6 cells reveal that HSJ633 appears to bind to IL-6R on a site remote from the IL-6 binding domain. Concordantly, HSJ633 selectively inhibits STAT3 phosphorylation while preserving the activation of cytoprotective AKT, p38, and ERK 1/2. In vivo, in a murine model of LPS-induced PTB, HSJ633 reduces inflammation in gestational and fetal tissues, preserves the integrity of fetal organs, and improves the survival of neonatal progeny when administered before and after the induction of labor by an inflammatory stimulus. Relevantly, the pharmacological inhibition of STAT3 in mice is sufficient to prevent PTB. Findings reveal first-in-class efficacy of a small peptide inhibitor of IL-6R, namely HSJ633, in impeding the inflammatory cascade associated with PTB and mitigating adverse neonatal outcomes.**

**Keywords** Inflammation; Interleukin-6; Neonatal Mortality; Non-competitive Modulator; Preterm Birth
**Subject Categories** Development; Immunology; Pharmacology & Drug Discovery

## Introduction

Preterm birth (PTB) is defined as delivery before 37 weeks of gestation and applies to ~15 million births every year worldwide. One million children succumb before 5 years of age from complications linked to PTB (Blencowe et al, 2013), corresponding to approximately 15.4% of global infant deaths (<5 years old) (Liu et al, 2015). Furthermore, compared to term infants, preterm children experience a greater likelihood of developing short- and long-term morbidities, such as visual impairment (Robinson et al, 1987), bronchopulmonary dysplasia (O'Reilly et al, 2013), necrotizing enterocolitis, and neurodevelopmental disorders (Cilieborg et al, 2011; Inder et al, 2023). Regrettably, no significant improvements have been developed to overcome the high prevalence of PTB and ensuing pathologies (Gotsch et al, 2009; Ouellet et al, 2013; Taguchi et al, 2017).

Despite the distressing consequences, the etiology of PTB is often poorly identified. Several risk factors including microbial infection, immune or hormonal imbalance, lifestyle and environmental exposures, and genetic polymorphisms are known to contribute to spontaneous PTB. Collectively, these converge to induce pro-inflammatory effects (Blencowe et al, 2013; Romero et al, 2001). Approximately 80% of births occurring before 28 weeks of gestation are attributed to infection and/or inflammation (Cappelletti et al, 2016); however, pathogens per se rarely cause PTB and adverse fetal outcomes; instead, the fetal–maternal immune response appears to have a dominant role in triggering PTB (Thaxton et al, 2010). Inflammation in gestational tissues is provoked by different cellular (damage-associated molecular patterns) and bacterial products (pathogen-associated molecular

[1]Department of Pharmacology and Physiology, Université de Montréal, Montreal, QC, Canada. [2]CHU Sainte-Justine Research Center, Montreal, QC, Canada. [3]Department of Pharmacology and Therapeutics, McGill University, Montreal, QC, Canada. [4]Department of Obstetrics and Gynaecology, University of Alberta, Edmonton, AB, Canada. [5]Department of Biology, Université de Sherbrooke, Sherbrooke, QC, Canada. [6]Department of Microbiology and Immunology, McGill University, Montreal, QC, Canada. [7]Department of Biochemistry, Université de Montréal, Montreal, QC, Canada. [8]Molecular Biology Program, Faculty of Medicine, Université de Montréal, Montreal, QC, Canada. [9]Department of Electrical and Computer Engineering, McGill University, Montreal, QC, Canada. [10]Medical Sciences, The University of Adelaide, Adelaide, SA, Australia. [11]Department of Chemistry, Université de Montréal, Montreal, QC, Canada. [✉]E-mail: sylvain.chemtob@umontreal.ca

patterns) (Cappelletti et al, 2016) which activate Toll-like receptors (TLR): notably TLR4 and TLR2 which are respectively stimulated by Gram(−) and Gram(+) bacteria (Thaxton et al, 2010). Upon activation in various tissues, including trophoblasts and uterine immune cells (Beijar et al, 2006; Holmlund et al, 2002; Kumazaki et al, 2004; Mitsunari et al, 2006), TLRs trigger the production of pro-inflammatory cytokines and chemokines, including interleukin (IL)-1β, IL-6, and tumor necrosis factor-α (TNFα). The latter in turn promote leukocyte infiltration and activate inflammatory cells in the uterus and cervix (Arntzen et al, 1998; Gomez-Lopez et al, 2010), ultimately resulting in uterine contractility, cervical ripening and fetal expulsion (Gotsch et al, 2009; Yoon et al, 2001).

Current strategies to prevent preterm labor rely on tocolytics to decrease myometrial contractility and hence delay labor (Olson et al, 2008; Patel and Ludmir, 2019). Largely ineffective in prolonging gestation for more than a couple of days, tocolytics do not improve perinatal outcome (Habelrih et al, 2024). Moreover, tocolytics are unable to curb the uterine inflammatory processes underlying PTB and associated fetal inflammatory response syndrome (FIRS), which is a risk factor of neonatal morbidity and mortality that has been associated with increased levels of IL-6 in fetal plasma (Olson et al, 2008; Simhan and Caritis, 2007).

IL-6, a 26 kDa cytokine with pleiotropic activity (Mihara et al, 2012), plays a central role in the inflammatory cascade underpinning premature parturition: high levels of IL-6 are found in the amniotic fluid of most patients advancing to deliver prematurely (Hee, 2011; Romero et al, 1990; Romero et al, 1993; Silver et al, 1993). Increased concentrations of IL-6 are also observed in contexts of inflammation and infection, as well as upon stimulation by lipopolysaccharide (LPS), TNFα, IL-1β, and other TLR ligands (Prins et al, 2012). IL-6 stimulates uterine contractions by inducing the expression of the oxytocin receptor (*Oxtr*) in rats (Fang et al, 2000). Genetic deficiency of IL-6 delays parturition (Dubinsky et al, 2008; Gutiérrez et al, 2004) and *Il6* null mutant mice are refractory to LPS-induced preterm delivery (Robertson et al, 2010). In women of European descent, the *IL6* rs1800795 genotype is protective against the risk of PTB (Wu et al, 2013). Moreover, the anti-IL-6 receptor antibody inhibits PTB induced by inflammatory stimuli in mice (Farias-Jofre et al, 2023; Wakabayashi et al, 2013); however, these antibodies which act as orthosteric antagonists, also pose a risk to the fragile developing neonate by inhibiting the entire signaling pathway, which is often not necessary and may be detrimental by not biasing signal transduction (Gregory et al, 2013). There is limited data on the exposure to an anti-IL-6 receptor antibody during the second and third trimesters, when transplacental transport is most significant; and the impact of an anti-IL-6 receptor antibody on the developing immune system remains unclear (Nana et al, 2024). Nonetheless, due to the significant role of IL-6 in pregnancy and related complications, inhibition of its receptor represents a promising target for preventing preterm birth and its deleterious effects.

IL-6 is produced by a variety of cells, including monocytes, dendritic cells, T cells, B cells, fibroblasts, endothelial cells, and placental trophoblasts (Mihara et al, 2012; Prins et al, 2012). IL-6 binds to the IL-6 receptor (IL-6R) which subsequently forms a ternary complex with glycoprotein 130 (gp130; also known as IL6ST). The IL-6R forms a hexameric complex with a 2:2:2 IL-6/IL-6R/gp130 stoichiometry (Boulanger et al, 1993). The IL-6 signaling pathways are constituted by the classical pathway involving the membrane receptor (mIL-6R) and the trans-signaling pathway involving the soluble receptor (sIL-6R). The sIL-6R is generated by alternative splicing or by protein cleavage of the mIL-6R (Kishimoto and Kang, 2022). In this ternary complex, gp130 conveys

intracellular signaling (Murakami et al, 1993). As gp130 is ubiquitously expressed, it enables the sIL-6R (trans-signaling pathway) to reach many cells (Silver and Hunter, 2010). Moreover, the soluble form of gp130 (sgp130) is a natural inhibitor of the sIL-6R and IL-6 complex. Thus, it sequesters this complex, preventing it from binding to membrane-bound gp130 (Jostock et al, 2001).

Downstream signaling of IL-6R/gp130 elicits various pathways such as JAK/STAT and SHP2-MAPK which involves p38, ERK, and AKT kinases (Mihara et al, 2012). The transcription factor STAT3 induces in turn a variety of genes implicated in inflammation, including IL-6, SOCS3 (suppressor of cytokine signaling 3), Bcl2, and LPS-binding protein (Kishimoto and Kang, 2022; Mihara et al, 2012; Naugler and Karin, 2008). Inhibitors of the IL-6/IL-6R system have been approved by the FDA for rheumatoid arthritis and Castleman's disease: Tocilizumab (Actemra) and Sarilumab (Kevzara) target the membrane and soluble IL-6 receptors, Siltuximab (Sylvant) targets IL-6 (Prairie et al, 2021). These monoclonal antibody drugs exert orthosteric activity on the IL-6R and interfere with all signaling triggered by IL-6 (Boyce et al, 2018; Sebba, 2008), but are modestly effective in rodent models of inflammation (Hu et al, 2018; Huehnchen et al, 2020; Poutoglidou et al, 2021; Wu et al, 2018). Accordingly, these orthosteric anti-IL-6/IL-6R biologics cause undesirable side effects such as interfering with cell survival and proliferation (Tanaka et al, 2014), producing pronounced undesirable immune suppression (Luchetti et al, 2016), and possibly worsening the rate of prematurity (Hoeltzenbein et al, 2016), thus limiting their use in pregnant women (not approved to prevent PTB).

The modulation of signaling pathways of other cytokines by allosteric agents has been shown to be beneficial in improving neonatal outcome in PTB animal models (Goupil et al, 2010). Biased signaling by such modulators can desirably preserve explicit receptor-coupled transduction to reduce consequential adverse effects (Sayah et al, 2020). Compared to orthosteric inhibitors, allosteric modulators offer greater functional selectivity (Gregory et al, 2013), which provide less unwanted effects by desirably preserving certain receptor-coupled signals. Accordingly, allosteric modulation of IL-6R signaling offers a promising therapeutic avenue for selective intervention in the inflammatory cascade towards PTB.

Thus, the objective of our research is to develop a safe and selective allosteric approach to prevent preterm birth and subsequently reduce the associated neonatal complications. We hypothesize that the IL-6 receptor is a promising target for the prevention of preterm birth and that allosteric antagonists will alleviate the adverse effects of complete IL-6 receptor inhibition in a vulnerable developing fetus. Initially, we performed a screening of allosteric all-D-peptides that target the IL-6 receptor. These peptides were characterized in an established animal model of LPS-induced prematurity, and the peptide demonstrating the most promising therapeutic effect in reducing prematurity (HSJ633) was further investigated.

The nanopeptide HSJ633 appears to selectively bind to IL-6R in vitro and inhibit STAT3 activation, while conserving the pro-reparative and cell survival functions of ERK, p38, and AKT. In murine models of infection-related inflammation-induced PTB, we show for the first time that HSJ633 seems to act as a small pharmacologically selective IL-6R antagonist that reduces inflammation in fetal–maternal tissues while lessening prematurity rates and improves neonatal organ integrity and outcomes when administered before as well as after the onset of labor induced by an inflammatory stimulus. HSJ633 is also effective (ex vivo) in dampening inflammation in human fetal membranes.

# Results

## Identification of IL-6R antagonists by screening for their effects on inflammation and prematurity

A series of small peptides (<10 amino acids) were designed based on specific sequences of the human IL-6/IL-6R complex to target mobile loop regions of the IL-6R structure (Fig. 1A), as described for a number of other receptors (Goupil et al, 2010; Quiniou et al, 2014; Quiniou et al, 2008; Rihakova et al, 2009). Figure 1A shows the region of the IL-6R/IL-6/gp130 complex corresponding to the origin of the HSJ633 sequence, in the structure predicted by AlphaFold; (the precise binding site of HSJ633 is not yet determined). Non-competitive IL-6 receptor antagonists can modulate signaling selectivity in a way that current IL-6R antibodies are unable to (Fig. 1B). Peptide inhibition of down-stream expression of the pro-inflammatory TNF-α was determined as efficacious in vitro and compared to the IL-6R orthosteric antagonist Tocilizumab. In cultured HEK cells overexpressing the IL-6 receptor (HEK-Blue IL-6), treatment with IL-6 (3.86 nM [$EC_{50}$ approximate]) for 6 h induced TNF-α expression (Fig. 1C). Among the peptides, HSJ633 and HSJ639 were the most effective in reducing the expression of TNF-α and exhibited effects consistent with that of the established IL-6R inhibitor Tocilizumab (Fig. 1C). (Curiously, HSJ631 increased expression of TNF-α, suggesting that it acts as a positive allosteric modulator (Fig. 1C)).

Efficacy of HSJ633 and HSJ639 in vivo was studied in a murine model of inflammation-induced PTB triggered on gestational day (GD) 16 by intraperitoneal injection of the Gram(−) bacteria mimic E. coli-derived LPS (total 10 μg) into pregnant CD1 dams (Fig. 1D). Vehicle-treated dams delivered prematurely prior to GD18.5 (term: GD18.5–19.5). Normal duration of gestation was restored in mice which were pre-treated (30 min) with HSJ633 or HSJ639 (2 mg/kg/day [based on average $IC_{50}$, see below, and estimated volume of distribution]) prior to LPS administration (Fig. 1E); rates of prematurity decreased correspondingly from ~93% to 22% (Fig. 1F); consistently, an anti-IL-6R mouse antibody (800 μg/kg/day) was effective in reducing prematurity rates (Fig. 1F). Congruently, LPS-induced uterine inflammation and pro-contractile gene expression on GD17.25 were markedly inhibited by HSJ633 but to a much lesser extent by HSJ639, especially with respect to relevant delivery-associated genes for the oxytocin (Oxtr) and prostanglandin $F_{2α}$ (Ptgfr) receptors, and partially for matrix metallopeptidase 9 (Mmp9, Fig. 1G). Based on greater efficacy, we focused on further characterizing HSJ633 (amino acid sequence: [all-D] VRKFQNSPA).

HSJ633 potently inhibited (average $IC_{50} \approx 1$ nM) IL-6-induced expression of IL-1β, TNF-α, and IL-6 mRNA in HEK-Blue IL-6R-expressing cells (Fig. 2A). The specificity of HSJ633 binding was studied on HEK-Blue IL-6R-expressing cells (Fig. 2B). No binding of [$^{125}$I]-HSJ633 was observed on HEK293 cells devoid of IL-6R. On HEK-Blue IL-6R-expressing cells, [$^{125}$I]-HSJ633 was competitively displaced by non-radiolabeled HSJ633 with an $IC_{50} \approx 12$ nM, (Fig. 2C) but not by non-radioactive IL-6 (Fig. 2D); it is thus inferred that HSJ633 interacts with IL-6R at a site remote from that of the endogenous IL-6 ligand.

There are some differences in homology between the human and murine IL-6 receptors, making it important to assess the effects of HSJ633 on the murine IL-6 receptor. As expected HSJ633 does not affect the IL-6R

pathway in IL-6R-devoid Ba/F3-mgp130 cells and in cells with Ba/F3-polylinker (empty vector); but HSJ633 does suppress IL-6-induced STAT3 activation in murine Ba/F3-mgp130-IL-6Rα cells (Fig. EV1A–D); the latter Ba/F3 cell line possesses the murine form of gp130 and IL-6Rα. This information substantiates the efficacy of HSJ633 on human and murine IL-6R pathway despite homology differences in IL-6R in these two species, and supports the relevance of murine testing envisaging future human application.

## HSJ633 reduces inflammation in human fetal membrane explants

To evaluate the anti-inflammatory efficacy of HSJ633 in human tissues, fetal membrane explants were collected from patients undergoing caesarean section at term (37–42 weeks of gestation) at the Royal Alexandra Hospital (Edmonton, Alberta). Fetal membrane explants were selected for this study as they are directly involved in the inflammatory processes linked to PTB, particularly premature rupture of membranes (PROM) and preterm labor. Fetal membrane explants allow for the study of inflammation in an ex vivo setting mimicking pathophysiological conditions of pregnancy. Human fetal membrane explants were stimulated with the DAMP HMGB1 to complement our work using a pathogen-associated molecular pattern (PAMP: LPS), as sterile inflammation plays a central role in PTB (Jain et al, 2022; Negishi et al, 2022). Stimulation of fetal membranes with HMGB1 significantly increased concentrations of IL-1β and IL-6 (in the supernatant), while treatment with HSJ633 markedly reduced HMGB1-induced secretion of IL-1β and IL-6 (Fig. 3). These findings underscore the efficacy of HSJ633 in reducing PAMP- and DAMP-induced inflammation, including in relevant human fetal membranes.

## Prophylactic inhibition of IL-6R using HSJ633 markedly diminishes inflammation in gestational tissues

To further characterize the mechanism of action of HSJ633 in reducing PTB, inflammation in gestational tissues was evaluated. In vivo, LPS induced a surge in the mRNA and protein concentrations of the pro-inflammatory cytokines IL-1β and IL-6 in the uterus, placenta, and fetal membranes of mice on GD17.25. We first needed to ascertain the efficacy of HSJ633 using a prophylactic approach. The expression of pro-inflammatory cytokines in gestational tissues was abrogated by pre-treatment with HSJ633 (Fig. 4A,B); similar observations were made for TNF-α (Fig. EV2A). LPS-triggered increases in IL-1β and IL-6 protein concentrations in the amniotic fluid and the rise of the inflammatory marker C-reactive protein levels in maternal plasma were all prevented by HSJ633 (Fig. 4C,D). Moreover, HSJ633 significantly suppressed LPS-induced upregulation of various other inflammatory gene products (e.g., Ccl2, Ccl3, Ptgs2, Mmp9, and Mmp3) in gestational tissues (Fig. EV2B).

## HSJ633 improved birth outcomes and preserved organ integrity in progeny when administered prophylactically

Given that HSJ633, when administered prophylactically, reduces gestational inflammation, we studied its impact on improving pup outcomes and preserving major organ integrity. Suppressed inflammation and prolonged gestation using HSJ633 were associated with markedly increased survival (87%), more viable pups at PT1 (9 pups per dams on average) and greater body mass at PT1 and PT7 (post-term) (Fig. 5A–C).

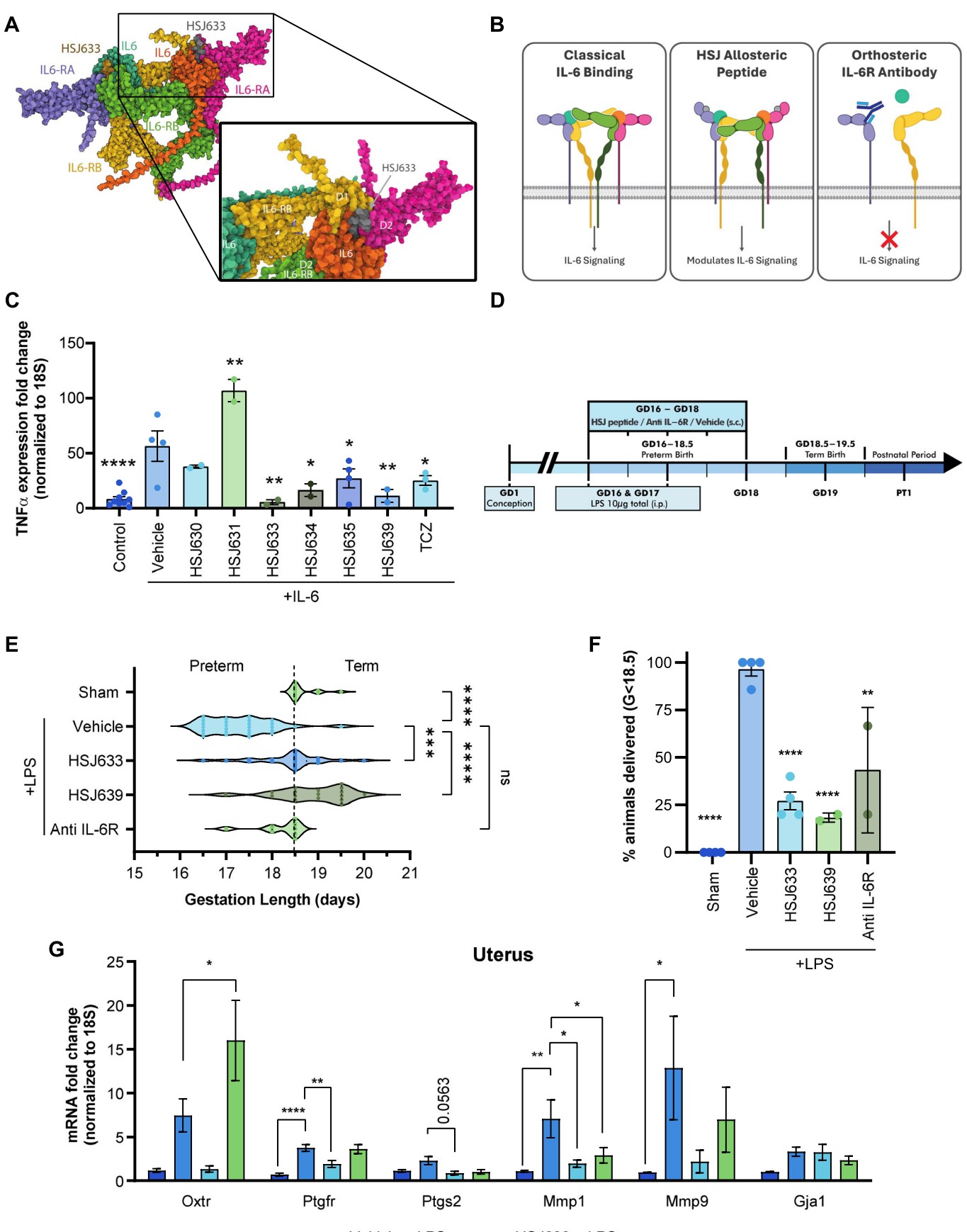

**Figure 1. Generation and screening of small peptide antagonists of IL-6R.**

(A) AlphaFold model of the IL6/IL6RA (refers to IL-6R) /IL6RB (refers to gp130) hexamer, indicating the specific site from which HSJ633 nanopeptide is derived from on the IL6RA sequence; this schema does not imply the actual binding site of HSJ633 (to be determined). IL-6 in blue and orange, IL-6R in pink and purple, IL-6RB (gp130) in green and yellow (Abramson et al, 2024). (B) Schematic diagram depicting signaling induced by IL-6, a binding site for HSJ group peptides remote from the ligand binding site, and antibody against the IL-6 receptor (as per Tocilizumab). The latter interferes with the orthosteric binding site, resulting in inhibition of all signals transduced by IL-6R (right panel); the middle panel shows HSJ peptides that bias signaling of IL-6R and thus modulate transduction. IL-6 in blue and orange, IL-6R in pink and purple, IL-6RB (gp130) in green and yellow. (Boulanger et al, 2003; Kishimoto and Kang, 2022). (C) TNF-α expression is presented as a fold change (normalized to 18S) obtained by RT-qPCR in HEK-Blue IL-6 cells treated with IL-6 (3.86 nM [clinically relevant concentrations]), HSJ630, HSJ631, HSJ633, HSJ634, HSJ635, HSJ639 (1 μM), and tocilizumab (TCZ: 3.44 μM). Values are mean ± SEM of 2-9 samples per group. One-way ANOVA with Dunnett's multiple comparisons test compared to the IL-6 + vehicle group. *$P < 0.05$, **$P < 0.01$, ****$P < 0.0001$. (D) Schematic experimental timeline. HSJ633, HSJ639 (2 mg/kg/day), a murine anti-IL-6 receptor antibody (Anti-IL-6R) (800 μg/kg/day) or vehicle is administered subcutaneously (s.c.) (repeated injection twice a day until GD18) and an intraperitoneal (i.p.) injection of LPS (4 μg on GD16 + 6 μg on GD17 LPS/100 μL saline, total 10 μg) was administered 30 min after the injection of HSJ633, HSJ639, or the anti-IL-6R. (E) Gestational length of mice in the LPS-induced PTB model. In total, 8–28 dams per group. Kruskal–Wallis test with Dunn's multiple comparisons test compared to the LPS + vehicle group. ***$P < 0.001$, ****$P < 0.0001$. (F) Prematurity rate (<18.5 days gestation) of mice in the employed LPS-induced PTB model. Values are mean ± SEM of four different experiments with 2–12 dams per group per experiment. One-way ANOVA with Dunnett's multiple comparisons test compared to the LPS + vehicle group. **$P < 0.01$, ****$P < 0.0001$. (G) mRNA of pro-inflammatory and uterine activation protein genes normalized to 18S in the uterus of mice extracted at G17.25 after LPS induction; histogram displays fold increase of gene products. Values are mean ± SEM of 4 dams per group by one-way ANOVA with Dunnett's multiple comparisons test compared to the LPS + vehicle group. *$P < 0.05$, **$P < 0.01$, ****$P < 0.0001$. Source data are available online for this figure.

Untreated LPS-treated dams had lower neonatal survival rates (33%) and lower viable pups at PT1 (average of 4 pups per dams), consistent with greater prematurity rates (Fig. 1E,F) and inflammation (Figs. 4 and EV2). The administration of the murine anti-IL-6R coherently improved neonatal survival (83%) and viability at PT1, but in contrast to HSJ633, the anti-IL-6R did not significantly improve body weight. Fetuses exposed to LPS and displaying visible signs of inflammation exhibited increased expression of pro-inflammatory cytokines (e.g., IL-1β, IL-6, TNF-α) in major organs (lungs and intestines) on GD17.25 (Fig. EV3) while treatment with HSJ633 curbed LPS-induced rises in cytokines in fetal organs.

Fetomaternal inflammation can elicit deleterious consequences to the fragile immature fetus (Beaudry-Richard et al, 2018; Nadeau-Vallée et al, 2017). Tissue histology was performed at PT7 on LPS-subjected animals to ascertain the consistency with inflammation and neonatal survival. The lungs of neonates from the LPS-treated dams exhibited decreased alveolar density and increased average alveolar area (which impairs gas exchange) (Fig. 5D). HSJ633 and the anti-IL-6R antibody prevented changes in the pulmonary parenchymal morphology induced by LPS (Fig. 5D). Similarly, the intestines of pups displayed detrimental corollaries associated with LPS-triggered inflammation at PT7. Reductions in intestinal diameter and villus height were (partly) prevented by HSJ633 but not by the anti-IL-6R antibody (Fig. 5E). Altogether HSJ633 improved perinatal outcomes with lasting effects on distressed pups subjected to inflammation.

### Distribution of HSJ633 within the fetal compartment

Efficacy in reducing inflammation in the fetal–placental compartment justified investigating the distribution of HSJ633 across the fetal–maternal interface. The biodistribution of HSJ633 labeled with FITC (HSJ633-FITC [~1.4 kDa]) was ascertained in the placenta 4 h post-administration of LPS on GD17, a time which corresponded to a gradual surge of LPS-induced inflammation (Hobbs et al, 2018; Naegelen et al, 2015). In the absence of LPS, the fluorescent signal intensity of HSJ633-FITC in the placenta was equivalent to the autofluorescence observed in the naïve control tissue (Fig. EV4). Upon LPS treatment, free FITC caused an increase in signal intensity in the placenta, likely attributed to inflammation-associated placental permeabilization (Fricke et al, 2018). Placental fluorescent

intensity was slightly further augmented in LPS-treated dams injected with HSJ633-FITC (Fig. EV4), suggesting that the peptide conjugate permeated across the placenta during the acute inflammatory event.

### HSJ633 administered in a treatment modality reduces gestational tissue inflammation, preterm birth, and neonatal mortality

It has been demonstrated in this study that HSJ633 reduces prematurity, decreases gestational inflammation, and improves pup outcomes when administered prophylactically. This approach allows targeting women at risk of preterm birth. However, it is equally important to develop a therapeutic option for women in whom labor induction has already begun. Therefore, the efficacy of HSJ633 in reducing prematurity and gestational inflammation as well as improving birth outcomes following labor induction by an inflammatory stimulus was evaluated. In the LPS-induced preterm birth model, we observed an increase in the expression of pro-inflammatory genes (*Il1b*, *Il6*, *Ptgs2*, and *Ccl2*) in reproductive tissues—placenta, uterus, and fetal membranes on GD17.25 (Fig. 6A). Administration of HSJ633 up to 6 h after preterm labor induction was able to reduce inflammatory gene expression in the placenta; in the uterus, *Il6* gene expression was decreased when HSJ633 was administered 2 h following LPS administration (Fig. 6A); a comparable efficacy of HSJ633 was observed for the post-LPS cytokine surge in the amniotic fluid (Fig. EV5A–C). Hence, HSJ633 effectively reduces cytokine concentrations up to 6 h post-LPS injection (Fig. 6B), implying that HSJ633 administered several hours post-LPS injection is effective in mitigating the ensuing inflammatory cascade.

Dams subjected to intraperitoneal LPS injections undergo premature parturition (84.9% vs 0% in the sham group) (GD < 18.5) (Fig. 1D). HJS633 given 2 h (but not 6 h) post-LPS significantly reduced the rate of prematurity (39.9% in HSJ633-2h vs 80% in HSJ633-6h) (Fig. 7A–F), coherent with uterine activation protein (UAP) expression (Fig. EV5A). Neonatal survival rate was also improved by HSJ633 administered 2 h post LPS (24.6% in LPS group vs 67.5% in HSJ633-2h vs 20% in the HSJ633-6h) (Fig. 7C); likewise, a greater pup survival rate per litter is also observed in the 2h-post LPS HSJ633 group compared to vehicle (7.5 pups/dam vs

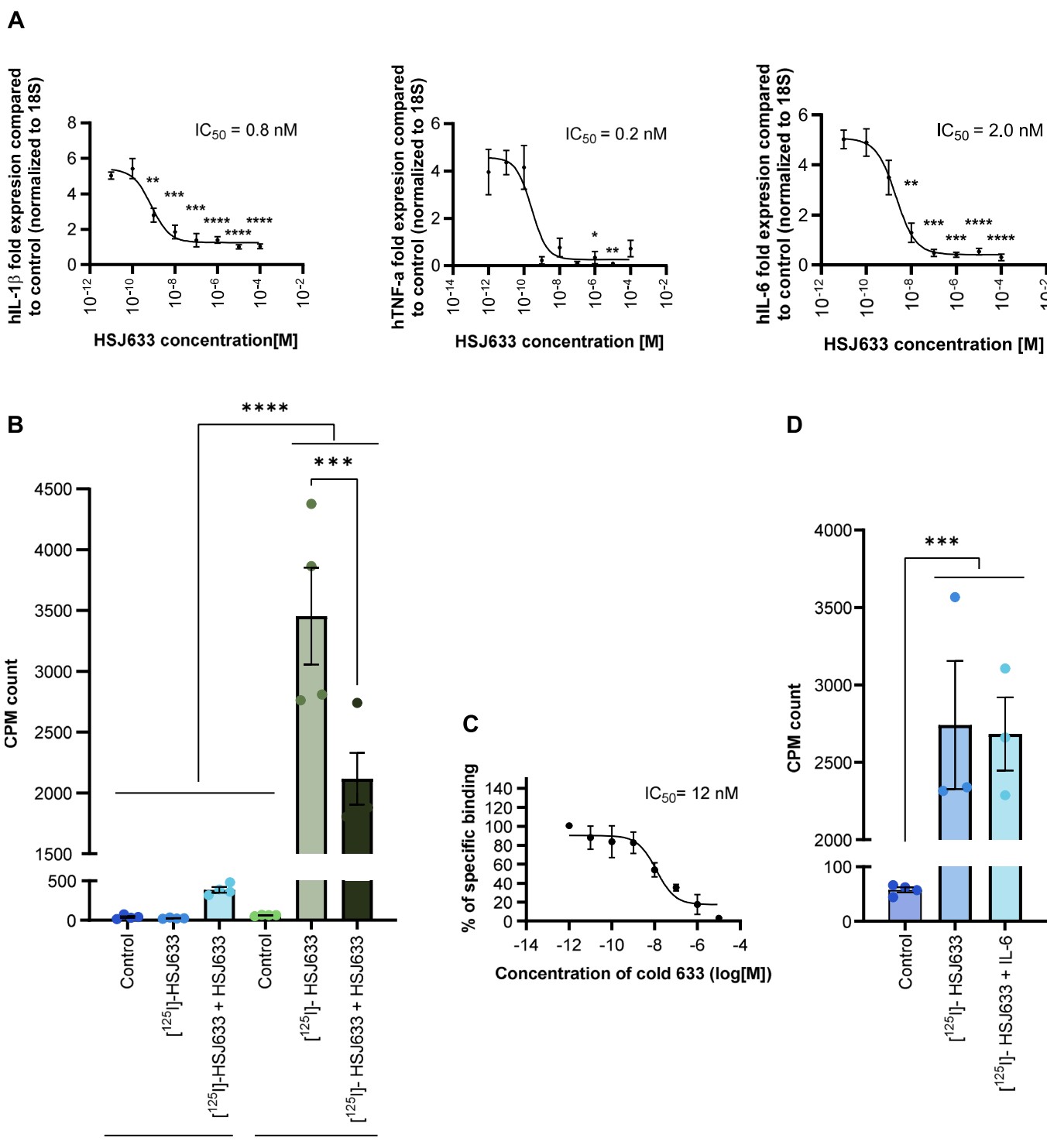

1.9 pups/dam) (Fig. 7C). This improved newborn viability was associated with a greater body mass at PT1 and PT7 (Fig. 7D).

We further evaluated the integrity of major organs in newborns given HSJ633 in a treatment modality. Maternal administration of HSJ633 2 h post-LPS protected the pulmonary parenchymal integrity as indicated by decreased alveolar space and greater alveolar density (Fig. 7E). Administration of HSJ633 up to 6 h post-LPS (at PT7) avoided inflammation-associated damage to the intestines attested by less dilated intestines and greater villus height (Fig. 7F). Hence HSJ633 given 2 h (or more) after induction of labor with LPS reduced prematurity, enhanced neonatal survival and preserved organ integrity in the progeny.

◀ **Figure 2. Pharmacological characterization of HSJ633.**

(A) Inhibition of IL-6-induced cytokines by HSJ633. HEK-Blue IL-6 cells were stimulated with different concentrations of HSJ633 ($10^{-11}$–$10^{-4}$ M) or vehicle and induced by either IL-6 (3.86 nM [$EC_{50}$ consistent]) or vehicle for 6 h. Expression of human IL-1β, TNFα and IL-6 mRNA was quantified by RT-qPCR and normalized to 18S. Values are mean ± SEM of 3–8 samples per group. **$P < 0.01$, ***$P < 0.001$, ****$P < 0.0001$. (B) Specific binding of HSJ633 to IL-6R-expressing cells. HEK-Blue IL-6 and HEK293 cells (devoid of IL-6R) were incubated with unlabeled HSJ633 ($10^{-4}$ M) or vehicle and then exposed to either [$^{125}$I]-HSJ633 ($10^{-8}$ M) or vehicle for 1.5 h. Radioactivity was measured and graphed as CPM counts. Values are mean ± SEM of four samples per group. ***$P < 0.001$, ****$P < 0.0001$. (C) Displacement of bound [$^{125}$I]-HSJ633 with cold HSJ633. Values are mean ± SEM of 3–4 samples per group. (D) Specific binding of [$^{125}$I]-HSJ633 ($10^{-8}$ M) to HEK-Blue IL-6 cells in the absence or presence IL-6 ($10^{-4}$ M). IL-6 does not displace binding of HSJ633. Values are mean ± SEM of 3–4 samples per group. ***$P < 0.001$, by one-way ANOVA with Dunnett's multiple comparisons test. Source data are available online for this figure.

**A**

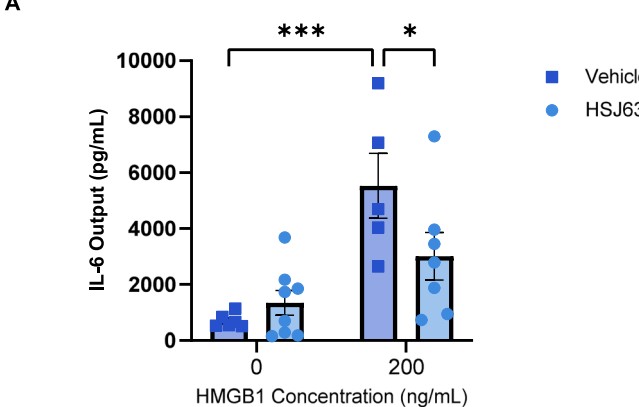

**B**

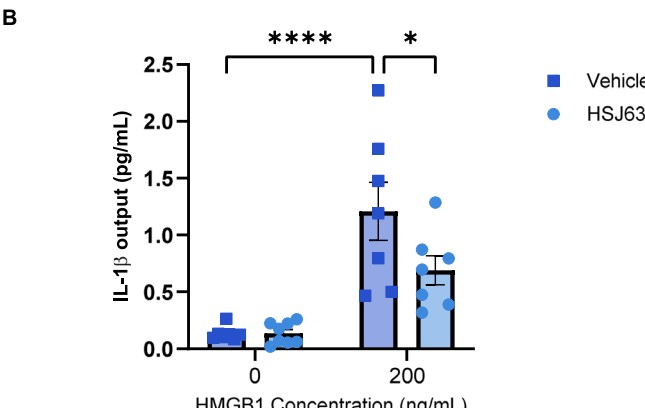

**Figure 3. HSJ633 reduces inflammation in human fetal membrane explants.**

Placentas were collected from term not-in-labor deliveries (elective cesareans) between 37 and 42 weeks at the Royal Alexandra Hospital in Edmonton, Alberta and intact fetal membranes were removed from the placenta. Human fetal membrane explants were pre-incubated with HSJ633 and stimulated with HMGB1 at 0 and 200 ng/mL. (A) IL-6 and (B) IL-1β output in the supernatant were evaluated in a multiplex assay. Values are mean ± SEM of 5–8 samples per group. Two-way ANOVA with Fisher's LSD test compared to HMGB1 0 and 200 ng/mL, and HMGB1 200 ng/mL with and without HSJ633. *$P < 0.05$, ***$P < 0.001$, ****$P < 0.0001$. Source data are available online for this figure.

## Mechanism of action of HSJ633: HSJ633 modulates the IL-6 signaling pathway predominantly through STAT3

This study demonstrates that HSJ633 is effective in reducing prematurity and gestational inflammation, as well as improving pup

outcomes, when administered both before and after labor induction by an inflammatory stimulus. HSJ633 thus exhibits promising therapeutic potential for preventing preterm birth. We determined the mechanism of action of HSJ633 on IL-6R. Canonically, IL-6R signals via MAPK, AKT and STAT3 (Mihara et al, 2012; Murakami et al, 1993). The effects of HSJ633 on IL-6R signaling were characterized in HEK-Blue IL-6 cells. Stimulation of HEK-Blue IL-6 cells with IL-6 elicited phosphorylation of p38, ERK1/2, AKT, and STAT3 (Fig. 8A–D). Phosphorylation of these proteins was blocked by the pre-incubation of cells with the IL-6R inhibitor Tocilizumab, an orthosteric antagonist used as a positive control, confirming that the observed protein activation was IL-6R-dependent (Fig. 8A–D). Interestingly, out of all these signal transducers, only STAT3 phosphorylation was reduced by the pre-treatment of HEK-Blue IL-6 cells with HSJ633 ($IC_{50} \approx 12$ nM; Fig. 8A–E); STAT3 activation is also detected in the placenta and is reduced by prophylactic administration of HSJ633 and to some extent by post-LPS HSJ633 treatment (Fig. 8F,G). Tocilizumab, an orthosteric antagonist, inhibits the entire IL-6R signaling cascade, whereas HSJ633, a non-competitive antagonist, selectively reduces STAT3 activation. This biased signaling induced by HSJ633 aligns with the preservation of pro-survival and growth factor-sensitive pathways elicited through p38, ERK, and AKT (Chung and Kondo, 2011; Vasudevan and Garraway, 2010) and is consistent with the allosteric modulatory effects of HSJ633 (Fig. 2D).

We distinguished soluble IL-6R and membrane-bound IL-6R in triggering the inflammatory cascade using primary amniotic epithelial cells (devoid of cell membrane-bound IL-6R). IL-6-induced STAT3 phosphorylation required co-treatment with soluble IL-6R. HSJ633 inhibition of IL-6-triggered STAT3 phosphorylation was confirmed in relevant primary human amniotic epithelial cells (Fig. EV6A). The importance of STAT3 in prolonging gestation (in inflammatory model) was ascertained using the STAT3 inhibitor NSC74859 (Lai et al, 2017; Zhang et al, 2013) (Fig. 8H). Collectively, these results infer that the beneficial effects of HSJ633 in preventing PTB can be ascribed to the selective inhibition of IL-6R-triggered STAT3 phosphorylation critical in amplifying inflammation (Kishimoto and Kang, 2022; Mihara et al, 2012; Naugler and Karin, 2008). To complement these results, Ba/F3-mgp130-IL-6Rα cells were treated with Hyper IL-6 (sIL-6R and IL-6) with or without HSJ633, and the activation of STAT3 was studied. Both HSJ633 and an anti-IL-6R antibody significantly reduced STAT3 activation when stimulated with Hyper IL-6 (Fig. EV6B). Furthermore, HSJ633 alone did not affect STAT3 activation (Figs. EV6B and EV1D).

To investigate the role of soluble IL-6R in PTB, inhibition of this receptor by soluble gp130 (sgp130) was studied. Prolonged gestation equivalent to that observed with HSJ633 (GD18.3,

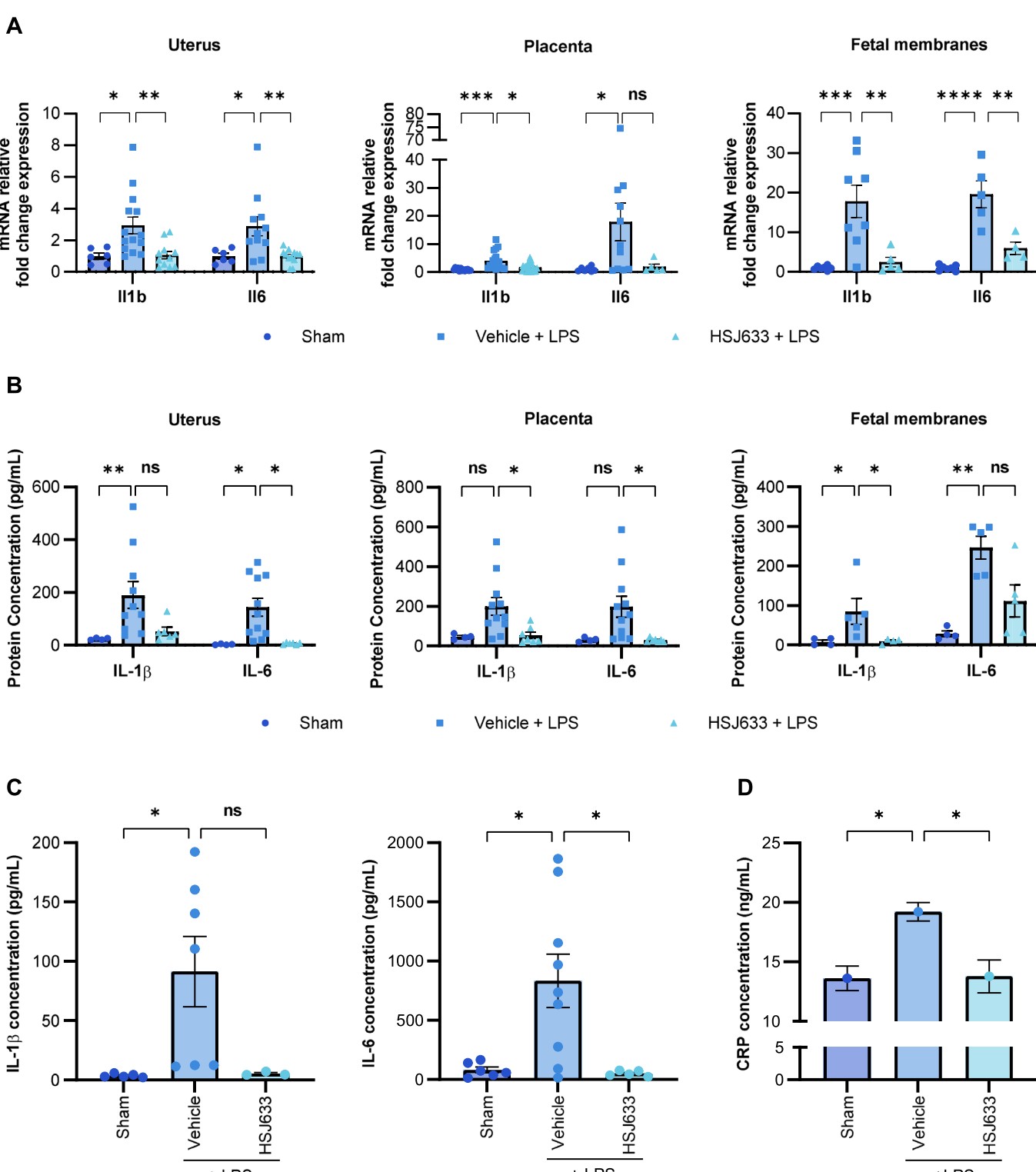

22.9% of PTB) was observed in LPS-treated dams injected in vivo with soluble gp130 (GD19, 20% of PTB) which binds soluble IL-6R and prevents it from inserting into the membrane to interact with membrane-bound gp130 (Fig. EV6C). It is thus suspected that the prime target of HSJ633 is the soluble IL-6R, most often described as the inflammatory IL-6R pathway (Prairie et al, 2021).

## Discussion

Fetal–maternal immune tolerance is critical for maintaining pregnancy. Intolerance leads to PTB and adverse gestational outcomes (Cappelletti et al, 2016). Pro-inflammatory cytokines, notably IL-6 (and interactive IL-1β), exert essential roles on

**Figure 4. Prophylactic administration of HSJ633 prevents LPS-induced inflammatory surge in gestational tissues and fluids.**

(A) mRNA of Il1b and Il6 in uterus, placenta, and fetal membranes extracted on GD17.25 after LPS induction (normalized to Actb). Values are mean ± SEM of 6–14 dams per group for the uterus, 5–21 dams per group for the placenta and 4–8 dams per group for fetal membranes. *$P < 0.05$, **$P < 0.01$, ***$P < 0.001$, ****$P < 0.0001$. (B) IL-1β and IL-6 protein expression on GD17.25 after LPS induction in the placenta, uterus, and fetal membranes. Values are mean ± SEM of 4–11 dams per group for the uterus and the placenta and 4–5 dams per group for fetal membranes. *$P < 0.05$, **$P < 0.01$. (C) IL-1β and IL-6 quantification in amniotic fluid extracted on GD17.25 after LPS induction. Values are mean ± SEM of 3–9 dams per group with one-way ANOVA with Dunnett's multiple comparisons test compared to LPS + vehicle group. *$P < 0.05$. (D) CRP quantification in maternal plasma extracted on GD17.25 after LPS induction. Values are mean ± SEM of 3–4 dams per group. IL-1β, IL-6, and CRP levels were determined by ELISA. *$P < 0.05$. One-way ANOVA with Dunnett's multiple comparisons test compared to LPS + vehicle group was performed when the distribution was normal and Kruskal–Wallis test with Dunn's multiple comparisons test compared to the LPS + vehicle group when data did not display normal distribution. Source data are available online for this figure.

pregnancy outcomes and its ensued complications (Ishihara and Hirano, 2002; Prins et al, 2012). The central role of IL-6 in parturition is evidenced by its higher levels in amniotic fluid compared to other cytokines (Hee, 2011; Liu et al, 2018; Romero et al, 1990; Silver et al, 1993) and prolonged gestation in IL-6-deficient subjects (Dubinsky et al, 2008; Gutiérrez et al, 2004). The currently FDA-approved (not for PTB) anti-IL-6R drugs are monoclonal antibodies. These biologics act as orthosteric inhibitors, suppress all downstream signals of IL-6R, and risk causing adverse effects by interfering with cell survival and immune-vigilance (Luchetti et al, 2016; Tanaka et al, 2014). The nanopeptide HSJ633 is a small molecule that exhibits anti-IL-6 activity. HSJ633 interacts with IL-6R without competing with IL-6. HSJ633 biases signaling, preserves IL-6R-coupled p38, ERK, and AKT phosphorylation which are involved in cell survival, (Manning and Toker, 2017; McCubrey et al, 2007; Yang et al, 2014), and selectively inhibits STAT3 activation. Potent in inhibiting the induction of inflammatory mediators in vitro and in vivo, HSJ633 administered before and after inducing preterm labor by an inflammatory stimulus, prolongs gestation, improves fetal organ integrity, and increases neonatal survival in a murine model of PTB induced by the clinically relevant pro-inflammatory TLR-4 stimulant, E. coli-derived LPS. Evaluating novel compounds in pregnant women poses ethical and practical challenges, given the involvement of both the mother and the fetus; HSJ633 reduced inflammation in relevant fetal membrane explants stimulated with the pertinent DAMP HMGB1. HSJ633 is the first validated small molecule antagonist of IL-6R, which exhibits favorable properties in inhibiting uteroplacental and fetal inflammation in murine and human tissues, and for extending gestation length as well as mitigating fetal and neonatal mortality and morbidity.

In the pathophysiology of PTB, uteroplacental inflammation is a key process which induces uterine activation, ultimately leading to the expulsion of the fetus (Cappelletti et al, 2016). High concentrations of pro-inflammatory cytokines in gestational tissues, predominantly IL-6, activate pro-labor pathways (Cappelletti et al, 2016). Treatments currently available, namely tocolytics, only interfere with uterine contractions, delay delivery by a mere couple of days, and fail to mitigate fetal inflammation or preserve fetal tissue integrity (Miller et al, 2022), thus not affecting newborn outcome. The targeting of upstream factors that are responsible for uterine activation and contractility embodies a rational approach for delaying labor and improving neonatal outcome (Coler et al, 2021), desirable to regulatory agencies. In this light, IL-6R is an appropriate target for PTB intervention due to the prominent role of IL-6 in inflammation and PTB. In addition to its high concentrations in amniotic fluid during parturition, IL-6 induces

uterine activation proteins, including the prostaglandin signaling pathway (Mitchell et al, 1991) and Oxtr expression in pregnant subjects (Fang et al, 2000). Blocking the signaling of IL-6R using an anti-IL-6 antibody has been found to be effective in a pre-clinical model of PTB (Wakabayashi et al, 2013). However, such a strategy risks inhibiting desirable IL-6 effects on cell protection (and immune-vigilance). The design of a non-competitive biased IL-6R modulator offers the potential to avoid unwanted effects while preserving the favorable IL-6R-associated signaling pathways. Among the designed peptides, HSJ633 was consistently the most effective.

Fetal and neonatal tissues are particularly vulnerable to maternal inflammation. Inflammatory stress is involved in the pathogenesis of various neonatal complications such as bronchopulmonary dysplasia (Balany and Bhandari, 2015) and necrotizing enterocolitis (Thompson and Bizzarro, 2008). In subjects exposed to acute maternal inflammation, immune cells and pro-inflammatory cytokines, including IL-6 and IL-1β, increase in the lung parenchyma (Balany and Bhandari, 2015; Kallapur et al, 2001). Inflammatory stress causes alveolar damage correlating to a phenotype of bronchopulmonary dysplasia (Kramer et al, 2009; Kunzmann et al, 2013; Viscardi, 2012). Moreover, the fetal and premature neonatal intestine is vulnerable to inflammation, particularly that in response to bacterial lipopolysaccharides (Jilling et al, 2006; Leaphart et al, 2007; Sharma et al, 2007).

Reducing maternal inflammation provides a coherent explanation for the observed benefits of our treatment on fetal organs. Alternatively, HSJ633 may act directly on the placenta and fetus. In any case, HSJ633 significantly reduces LPS-induced prematurity. The biodistribution of HSJ633 using FITC-conjugated HSJ633 suggests that it crosses the placenta under inflammatory conditions. Under non-inflammatory conditions, HSJ633-FITC is not detected in the placenta, possibly due to integrity of villous endothelial barrier and to the polarity of both HSJ633 and FITC, as small nonpolar molecules tend to cross the placenta more readily. In contrast, inflammation increases placental permeability (to various molecules), potentially facilitating the passage of a positively charged polar molecule such as HSJ633. Yet, polar molecules can cross the placenta via different transporters (Feghali et al, 2015). Mass spectrometry could clarify whether or not HSJ633 crosses the placenta to exert its anti-inflammatory effects in the fetal compartment, and would be the approach of choice in human pharmacokinetic trials.

In a distinct feature of the present study, HSJ633 maintained fetal/neonatal integrity. The benefits of HSJ633 were correlated with the marked reduction of STAT3 phosphorylation and the preservation of the activation of p38, ERK1/2 and AKT in vitro.

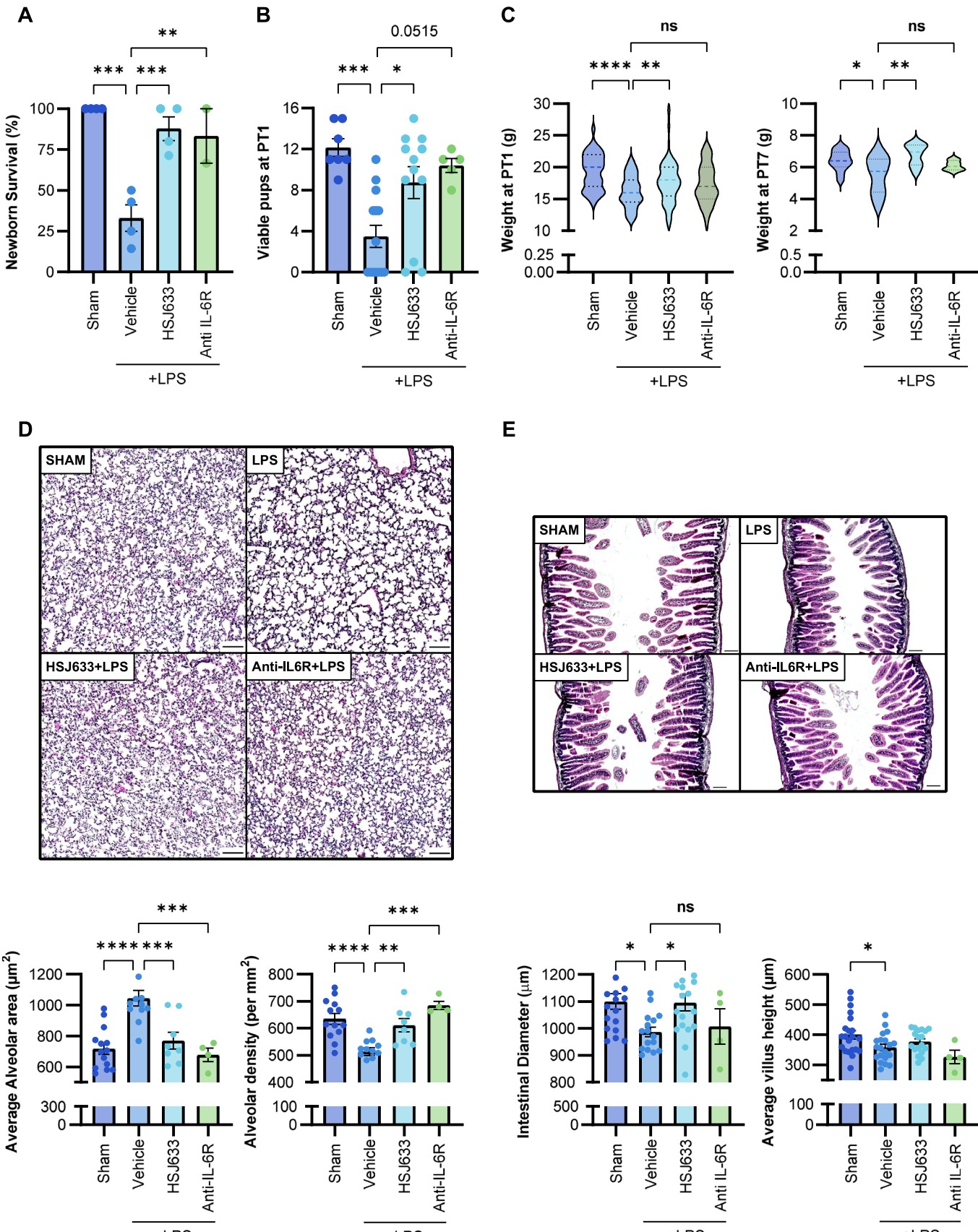

**Figure 5.  Antenatal HSJ633 enhances neonatal survival and preserves organ integrity in progeny when administered prophylactically.**

(A) Neonatal survival rate. Values are mean ± SEM of 2–4 different experiments with 3–8 dams per group per experiment. If two or more pups are alive, the litter is considered viable. One-way ANOVA with Dunnett's multiple comparisons test compared to LPS + vehicle group. **$P < 0.01$, ***$P < 0.001$. (B) Viable pups at PT1. Values are mean ± SEM of 6–15 litter per group. Kruskal–Wallis test with Dunn's multiple comparisons test compared to the LPS + vehicle group. *$P < 0.05$, ***$P < 0.001$. (C) Pup body weights at PT1 and PT7 (age corrected). Values are mean ± SEM of 49–105 pups for the left panel with Kruskal–Wallis test with Dunn's multiple comparisons test compared to the LPS + vehicle group. Values are mean ± SEM of 4–12 pups for right panel with one-way ANOVA with Dunnett's multiple comparisons test compared to LPS + vehicle group. *$P < 0.05$, **$P < 0.01$, ***$P < 0.001$. (D) Representative histological images of lung parenchyma stained with hematoxylin and eosin. Lungs were collected at PT7. Scale bars = 100 µm. Measurement of alveolar density (1 mm$^2$) and average alveolar area (µm$^2$) quantified on regions of interest of 1 mm$^2$ are presented on histograms to the right of histology panels. Values are mean ± SEM of 4–13 pups per group. **$P < 0.01$, ***$P < 0.001$, ****$P < 0.0001$. (E) Representative histological images of intestines collected at PT7 and stained with hematoxylin and eosin; scale bars = 100 µm. Intestinal diameters (µm) of the regions of interest were quantified; villus height (µm) was measured using ZEN software: ten villi by region of interest were compiled; quantifications are presented on histograms to the right of images. Values are mean ± SEM of 4–18 pups per group. One-way ANOVA with Dunnett's multiple comparisons test compared to LPS + vehicle group for (D, E). *$P < 0.05$. Source data are available online for this figure.

The SHP2/MAPK and PI3K/AKT pathways involve p38, ERK1/2 and AKT in cell proliferation and survival (Manning and Toker, 2017; McCubrey et al, 2007; Yang et al, 2014). In the placenta, STAT3 signaling is not entirely diminished as demonstrated in vitro. HSJ633 reduces the activation of IL-6-induced STAT3 but does not decrease the activation of all STAT3. This discrepancy in the reduction observed in the placenta compared to HEK-Blue IL-6 cells could be attributed to these differential effects on STAT3 activation. However, this demonstrates a significant role of STAT3 mediated by the IL-6R. Thus, reducing STAT3 activation solely through modulation of IL-6 by HSJ633 is sufficient to mitigate preterm birth. By postulating on the biased mechanism of action of HSJ633, the allosteric binding may induce a change in the conformational dynamics of the IL-6/IL-6R/gp130 complex that selectively conceals STAT3 binding sites (Kishimoto and Kang, 2022); whereas the gp130 binding site (YXXV motif) involved in the SHP2/MAPK pathway may be unaffected by HSJ633, which selectively modulates the JAK/STAT signaling pathway involved in prematurity and inflammation. Our findings are consistent with the complex biological effects exerted by the IL-6-induced STAT3 signaling pathway, which physiologically governs epithelial regeneration, angiogenesis, and exaggerated inflammation induced by inflammatory lymphoid cells (Willson et al, 2013; Ding et al, 2008). The IL-6-induced STAT3 pathway is also implicated in pathological (innate) inflammation (Jo et al, 2021; Matsuda, 2023), which is underscored by the present study in which the STAT3 inhibitor, NCS74859, proved to be effective in preventing LPS-induced PTB. The greater susceptibility to inflammation of *Il6*-deficient mice compared to wild type has been attributed to a marked deficiency in STAT3 (Grivennikov et al, 2009). Activation of IL-6R classical pathway is essential for intestinal epithelial cell proliferation, and the complete inhibition of classical signaling could lead to intestinal damage (Aden et al, 2016; Gout et al, 2011). All-in-all, HSJ633 simply modulates IL-6-coupled downstream signaling, including that of STAT3, and preserves the integrity of major fetal/neonatal organs.

Complete or partial inhibition of the IL-6 receptor by HSJ633 reduces maternal inflammation and thus prematurity levels. According to the study by Sakurai et al, MR-16, a rat anti-mouse IL-6 receptor antibody, has no impact on fertility, embryonic development required for implantation or pre/postnatal development (Sakurai et al, 2012). However, the administration of IL-6 receptor inhibitors, such as Tocilizumab, is not suggested for women wishing to become pregnant or who are pregnant, due to

limitations on the safety of these drugs; some women undergoing treatment with Tocilizumab for rheumatoid arthritis (RA) may have been exposed to Tocilizumab before and during early pregnancy. The increased use of Tocilizumab in pregnant women during the COVID-19 pandemic provides new data on the safety of this agent. In the study by Hoeltzenbein et al, pregnancies which were prospectively reported following exposure to Tocilizumab showed a 31.1% increase rate in preterm birth, in comparison to the general population (Hoeltzenbein et al, 2016); further data is required to establish the risk-benefit balance for the fetus and newborn (Jiménez-Lozano et al, 2021; Jorgensen et al, 2022). Notably, anti-IL-6 antibodies have been shown to cause a decrease in respiratory branching and cell proliferation in lungs (Nogueira-Silva et al, 2006), possibly related to inhibition of the activated cytoprotective p38-MAPK. These observations highlight the benefits of a biased ligand such as HSJ633 which preserves the MAPK pathways.

The murine and human IL-6 receptors exhibit some structural differences. HSJ633 has been formulated based on the human IL-6 receptor. MR-16, a murine anti-IL-6R antibody, fails to impede human T cell proliferation stimulated by IL-6, suggesting its lack of recognition of the human IL-6R (Okazaki et al, 2002). Similarly, Tocilizumab exhibits no hindrance on the proliferation and activation of STAT3 in Ba/F3-gp130-mIL-6R (expressing the murine IL-6R). Both Tocilizumab and MR-16 act as orthosteric antagonists, competitively interfering with IL-6 for receptor occupancy (Lokau et al, 2020). The moderate conservation observed within the D2 and D3 domains of the IL-6R across species (particularly human and mouse) accounts for the inefficacy of these molecules in alternative organisms (Grötzinger et al, 1997). However, HSJ633 diverges as a non-competitive antagonist of the IL-6R, distinct from Tocilizumab or MR-16, which exert competitive actions against IL-6. Consequently, notwithstanding the moderate conservation at the D2 and D3 sites among species, HSJ633, operating as a non-competitive antagonist, attenuates STAT3 activation within cells harboring the murine IL-6 receptor by presumably inducing conformational changes in the receptor, thereby eliciting biased signaling. The specific site of interaction of HSJ633 with IL-6R and impact on its structural conformation is under investigation and remain to be precisely determined.

The actions of IL-6 are conveyed by soluble and membrane forms of the receptor, both of which are present in gestational tissues (Prins et al, 2012), and form complexes with the ubiquitous subunit gp130 (Xu and Neamati, 2013). Cells devoid of the

**A**

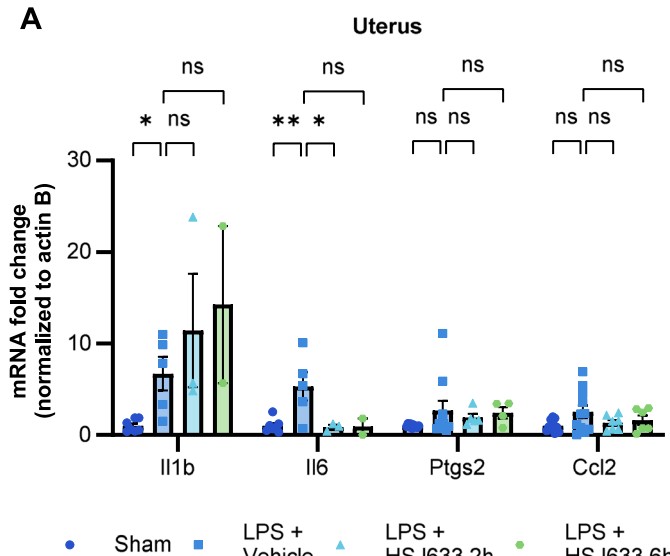

**B**

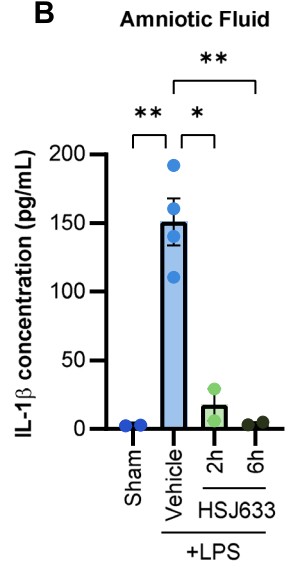

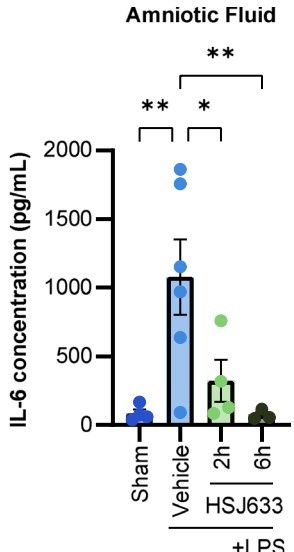

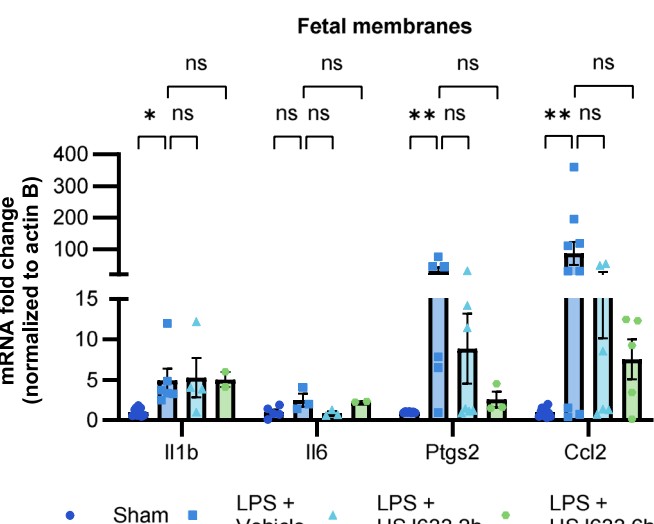

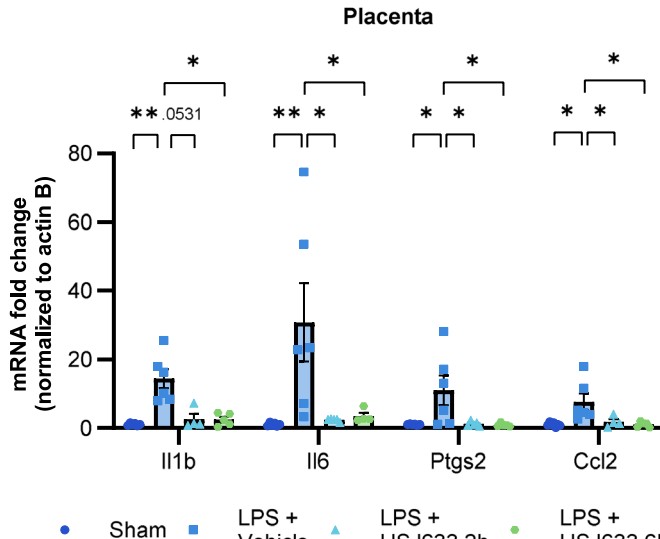

**Figure 6.  HSJ633 administered 2 h after LPS injection prevents inflammation in gestational tissues and fluids.**

(A) mRNA of *Il1b*, *Il6*, *Ptgs2*, and *Ccl2* in uterus, placenta, and fetal membranes extracted on GD17.25 after LPS induction (normalized to *Actb*). Values are mean ± SEM of 2–11 dams per group for the uterus, 4–7 dams per group for the placenta and 2–11 dams per group for fetal membranes. One-way ANOVA with Dunnett's multiple comparisons test compared to the LPS + vehicle group was performed when the distribution was normal, and Kruskal–Wallis test with Dunn's multiple comparisons test compared to the LPS + vehicle group when data did not display normal distribution. *$P < 0.05$, **$P < 0.01$. (B) Quantification of IL-1β and IL-6 by ELISA in amniotic fluid extracted on GD17.25 after LPS induction. Values are mean ± SEM of 2–6 dams per group. One-way ANOVA with Dunnett's multiple comparisons test compared to the LPS + vehicle group was performed. *$P < 0.05$, **$P < 0.01$. Source data are available online for this figure.

membrane form of IL-6R can still respond to IL-6 through interaction with soluble IL-6R and membrane-bound gp130. Soluble IL-6R is generated by proteolytic cleavage [catalyzed by ADAM metallopeptidase domain 10 (ADAM10) and ADAM17] and by alternative splicing (Rose-John, 2012). Expressed in various tissues, including trophoblasts and decidual cells (Yang et al, 2012), ADAM10/17 can cleave soluble IL-6R in cells remote from those experiencing IL-6 activation. In gestational tissues, HSJ633 may likely modulate the signaling of soluble and membrane-bound IL-6R, as the peptide was designed from a common region shared by both forms. Although the distinctions in the modulation of both classical and trans-signaling of IL-6 are not known, soluble IL-6R was required to elicit STAT3 activation in human amniotic epithelial cells. Soluble gp130 has the capacity to sequester soluble IL-6R and was found to replicate the efficacy of HSJ633. Prominent roles may be inferred for soluble IL-6R in LPS-induced PTB and the mitigating effects of HSJ633.

We have shown the efficacy of HSJ633 in preventing preterm birth when administered both prior to and following labor induction. Prophylactic administration involves targeting women at risk of preterm birth, which is more challenging to accomplish due to the lack of predictive tools. Utilizing both a preventative and therapeutic approach facilitates the targeting of women at risk of preterm labor and those who have not been identified. Thus, HSJ633 exhibits a significant reduction in preterm birth when administered at 2 h post-labor induction in a murine model, corresponding to an estimated 30 h delay after labor induction by an inflammatory stimulus in human. Consequently, women at risk of preterm birth and those with initiated inflammatory cascades could potentially benefit from HSJ633 treatment. As anticipated for any anti-inflammatory, optimal efficacy of HSJ633 is reached when the peptide is administered before the initiation of uterine contractions, as HSJ633 does not function as a tocolytic but rather attenuates inflammation. Consequently, an early intervention is evidently ideal. The observed expression of UAPs at 6 h post-labor induction with LPS uncovers the inefficacy of HSJ633 in preventing preterm delivery when administered this late. An anti-IL-6R administered 6 h post-labor induction with LPS also fails to exhibit efficacy in reducing prematurity (Farias-Jofre et al, 2023); this is consistent with our results whereby HSJ633 administered 6 h post-LPS injection did not reduce PTB. This time point reflects the activation of UAPs in mice and in turn the end of the inflammatory cascade and is therefore considered as a late phase for intervention. However, in an IL-1α model of sterile inflammation, an anti-IL-6R administered 6 h post-labor induction reduced rates of prematurity and neonatal mortality (Farias-Jofre et al, 2023). Thus, the efficacy of drugs targeting the IL-6 receptor may be contingent upon the specific processes of labor induction affecting outcomes when administered late.

## Conclusion

Antagonism of the major pro-inflammatory IL-6R using the first-in-class peptide, HSJ633 is hereby presented as an innovative and rational approach to prevent PTB and its corollary detrimental effects on the fetus. HSJ633 displays pharmacological selectivity through biased signaling. In this context, HSJ633 inhibited STAT3 activation while preserving cytoprotective pathways linked to p38, ERK, and AKT, which are preferably left untouched for the fragile fetus. Altogether, HSJ633 is a promising unprecedented biased allosteric IL-6R modulator that suppresses antenatal inflammation and prevents unfavorable neonatal outcomes associated with PTB.

## Limitations of the study

Although frequently employed (Miller et al, 2022), the LPS-induced preterm labor model may not precisely reproduce the pathogenesis of preterm labor in humans. However, this model of intraperitoneal LPS injection is frequently used to represent infection leading to prematurity by several prominent groups in the field (Chin et al, 2016; Deng et al, 2016; Manuel et al, 2019; Shynlova et al, 2014; Sun et al, 2016; Sundaram et al, 2013).

## Methods

**Reagents and tools table**

| Reagent/resource | Reference or source | Identifier or catalog number |
|---|---|---|
| **Experimental models** | | |
| HEK-Blue IL-6 cells (*H. sapiens*) | InvivoGen | hekb-hil6 |
| HEK293 (*H. sapiens*) | ATCC | CRL-1573 |
| Placenta | Royal Alexandra Hospital | |
| Ba/F3 (*M. musculus*) | Creative Bioarray | CSC-C2045 |
| CD-1 (*M. musculus*) | Charles River Laboratories | N/A |
| **Recombinant DNA** | | |
| mouse gp130 | Open Biosystems | |
| mouse IL6Rα | Open Biosystems | |
| pMX retroviral expression vector | Kitamura et al, 2003 | |
| **Antibodies** | | |
| InVivoMAb anti-mouse IL-6R | BioXCell | #BE0047, AB_1107588 |

| Reagent/resource | Reference or source | Identifier or catalog number |
| --- | --- | --- |
| PRACTEMRA | La Roche Limited | |
| Mouse anti-phospho-STAT3 | Cell Signaling Technology | 9138 |
| Rabbit anti-phospho-STAT3 | | 9145 |
| Rabbit anti-phospho-AKT | | 9271 |
| Rabbit anti-phospho-ERK1/2 | | 9101 |
| Rabbit anti-phospho-p38 | | 9212 |
| Mouse anti-STAT3 | | 9139 |
| Rabbit anti-STAT3 | | 4904 |
| Rabbit anti-AKT | | 9272 |
| Rabbit anti-ERK1/2 | | 4695 |
| Rabbit anti-p38 | | 4511 |
| Rabbit anti-gp130 | | 3732 |
| Mouse beta Actin Antibody (C4) | Santa Cruz | sc-47778 |
| Rabbit Anti-cytochrome c oxidase 4 | Proteintech | 11242-1-AP |
| Mouse anti-GAPDH | | 60004-1-Ig |
| Anti-mIL-6R α | R&D systems | AF1830 |
| GAPDH Loading Control Monoclonal Antibody (GA1R) | Thermo Fisher Scientific | MA5-15738 |
| Donkey anti-goat IgG-HRP | Santa Cruz | Sc-2020 |
| Anti-rabbit IgG, HRP-linked Antibody | Cell Signaling Technology | 7074 |
| IRDye® 680RD Donkey anti-Mouse IgG Secondary Antibody | LI-COR Biosciences | 926-68072 |
| Goat Anti-Mouse IgG (H + L)-HRP Conjugate | Bio-Rad Laboratories | 1721011 |
| **Oligonucleotides and other sequence-based reagents** | | |
| PCR Primers | This study | Table EV1 |
| **Chemicals, enzymes, and other reagents** | | |
| Sucrose | Bio Basic | SB0498 |
| iScript reverse transcription superMix | Bio-Rad Laboratories | 1708841 |
| iTaq Universal SYBR Green Supermix | | 1725124 |
| Clarity MAX Western ECL Substrate | | 1705062 |
| Bio-Plex Pro Reagent Kit III with Flat Bottom Plate | | 171-304090 M custom 4-plex |
| Laemmli-SDS | Boston Bioproducts | BP-110R |
| Lipopolysaccharide O111:B4 (LPS) | Caymen Chemical Company | 19661 |
| HMGB1 | Millipore Sigma | SRP6265 |
| Milliplex MAP | | #HCYTA-60K, #HCYP2MAG-62K |
| Radioimmunoprecipitation assay buffer (RIPA) | Cell Signalling Technology | 9806 |
| Peptides (HSJ633, HSJ630, HSJ631, HSJ634, HSJ635, HSJ639, HSJ633-FITC) | Elim Biopharmaceuticals | |
| Tween-20 | Fisher Scientific | BP337-500 |
| PlusOne Mini Dialysis Kit | GE Healthcare | |

| Reagent/resource | Reference or source | Identifier or catalog number |
| --- | --- | --- |
| HEK_Blue Selection | InvivoGen | hb-sel |
| QUANTI-Blue™ Solution | | Rep-qbs |
| MycoStrip | | rep-mys |
| Intercept® (PBS) Blocking Buffer | LI-COR Biosciences | 927-70001 |
| Human IL-6 | Peprotech | 200-06 |
| Recombinant murine IL-6 | | 216-16 |
| Murine IL-3 | | 213-13-10UG |
| RNeasy Plus Mini Kit | Qiagen | 74136 |
| rmgp130/Fc Chimera | R&D Systems | 468-MG |
| IL-1β/IL-1F2 Quantikine | | MLB00C |
| mouse C-Reactive Protein/CRP | | MCRP00 |
| mouse IL-6 Quantikine | | M6000B |
| mouse TNF-α Quantikine | | MTA00B |
| rmIL-6/IL-6R α (Hyper-IL-6) | | 9038-SR/CF |
| Western Lightning Plus ECL | Revity | NEL104001EA |
| cOmplete EDTA-free Protease Inhibitor Cocktail | Roche | 11873580001 |
| Dispase II solution | Sigma-Aldrich | 42613-33-2 |
| Phenylmethanesulfonyl fluoride (PMSF) | | P7627 |
| sIL-6R | | SRP3097 |
| Triton-X100 | | X100-500ML |
| Bovine Serum Albumin (BSA) | | A9647-100G |
| DAPI | | D9542 |
| Polybrene | | TR-1003 |
| DNAse I | Thermo Fisher Scientific | EN0525 |
| Sulfo-SHPP (Water-Soluble Bolton–Hunter Reagent) | | 27712 |
| NSC74859 | TOCRIS Bioscience | 4655 |
| RiboZol RNA Extraction Reagent | VWR | N580-200ML |
| DMEM F12 | Gibco | 11330032 |
| Antibiotic/Antimycotic for AEC | | 15240062 |
| DMEM for AEC | Hyclone, Cytiva Life Sciences | SH30284-01 |
| RPMI 1640 | Wisent Inc. | 350-000-CL |
| DMEM (HEK, HEK-Blue IL-6) | | 319-005-CL |
| Fetal bovine serum (FBS) | | 080850 |
| Trypsin/EDTA | | 325-542-EL |
| Penicillin/Streptomycin | | 450-201-EL |
| Puromycin Dihydrochloride | | 400-160-EM |
| Tris buffered saline 10 X (TBS) | | 311-030-LL |
| Phosphate Buffered saline 10× (PBS) | | 311-012-CL |
| **Software** | | |
| ImageLab | Bio-Rad Laboratories | |
| Image Studio V3.0 | LI-COR Biosciences | |
| Bio-Plex 200 software version 6.1 | | |

| Reagent/resource | Reference or source | Identifier or catalog number |
| --- | --- | --- |
| ImageJ | https://imagej.nih.gov/ij/index.html | |
| GraphPad Prism V10 | GraphPad Software | |
| **Other** | | |
| Pierce™ C18 Spin Tips & Columns | Thermo scientific | 89870 |
| Pierce Iodination Tubes | Thermo scientific | 28601 |
| Packard Cobra-II Auto Gamma Counter | | |
| Light Cycler 96 | Roche Life Science | |
| ChemiDoc | Bio-Rad Laboratories | |
| NanoDrop 1000 spectrophotometer | Thermo Fisher Scientific | |
| Odyssey LI-COR Biosciences Infrared Imaging System | LI-COR Biosciences | |
| Zeiss AxioScan.Z1 | Zeiss | |
| Bio-Plex 200 suspension array system | Bio-Rad Laboratories | |

## Animals

Pregnant CD-1 mice were acquired from Charles River Laboratories on gestational day 11 and were acclimatized over a period of 5 days before treatment interventions. The experimental protocols used in this study were approved by the Animal Care Committee of the CHU Sainte-Justine Hospital (#2023-5580 and #2020-2755) based on the principles of the *Guide for the Care and Use of Experimental Animals* of the Canadian Council on Animal Care. The animals were allowed free access to standard laboratory water and chow and were housed under a 12:12 light:dark cycle.

## Peptides design

A series of small peptides (<10 amino acids) were designed based on specific sequences of the human IL-6/IL-6R complex to target mobile loop regions of the IL-6R structure, as described for a number of other receptors, notably FP, IL-1R, V2R, IL-23R (Goupil et al, 2010; Quiniou et al, 2014, 2008; Rihakova et al, 2009). The regions of the IL-6 receptor were pinpointed through a combination of crystallography and modeling data (Boulanger et al, 2003), which were further validated using hydrophobic and flexibility profiles. Additionally, computational analysis involving homology domains through tools like ProDom, PROSITE, Predict Protein, and ProtScale supported these findings. The predictive structure of the IL-6/IL-6R/gp130 hexameric receptor complex was generated using AlphaFold (Abramson et al, 2024). Amino acid sequences of IL-6, IL-6R, and gp130, excluding their cytoplasmic domains, were used for the multimeric modeling. The resulting AlphaFold model also included HSJ633 polypeptides (9 amino acids), derived from the IL-6Rα sequence and located within the fibronectin domain (Abramson et al, 2024). After identifying loops and hinge regions resulting from the analysis described above, small portions of the primary structure of the IL-6 receptor loops were extracted (9–12 amino acids), and a set of six corresponding homologous peptides were generated. These peptides are composed of D-octa-deca

sequences in both sense (NH3-COOH) orientations as described by Quiniou et al (Quiniou et al, 2008).

## Human fetal membrane (hFM) explants treatment and multiplex assay

Placentas were collected from term not-in-labor deliveries (elective cesareans) between 37-42 weeks at the Royal Alexandra Hospital in Edmonton, AB. The exclusion criteria for this study were patients with multiple pregnancies, clinical infection, diabetes mellitus, immunological problems, intrauterine growth restriction, preeclampsia, or bleeding; this avoided confounding determinants in timing and degree of inflammation. We excluded women under 18 years of age or >40 years. Placentas were collected from eligible study participants after informed written consent, and all experiments conformed to the principles set out in the WMA Declaration of Helsinki and the Department of Health and Human Services Belmont Report. Ethics approval for this study was received from the University of Alberta Research Ethics Board, Study ID Pro00069209. Fetal membranes were obtained as previously described by Dr. Leimert (Leimert et al, 2019).

Freshly obtained hFM were treated with HSJ633 ($1 \times 10^{-6}$ M) for 1 h prior to treatment with HMGB1 (0 and 200 ng/mL) for 24 h. Treatments were cultured in duplicate, and supernatants were collected and pooled with supernatant from the matching sample's duplicate. Human FM tissues were removed, weighed, and snap frozen. Multiplex assay was performed as described by Dr. Leimert (Leimert et al, 2019). To account for variability in explant weight and thickness between samples from the same placenta, cytokine outputs were normalized based on the weight of each explant. Specifically, the median explant weight for each patient was standardized to a value of 1, and cytokine levels were expressed as a ratio relative to this standardized weight.

## Extraction of amniotic epithelial cell (AEC)

Intact placentas were collected from pregnant, non-laboring women undergoing elective caesarian sections at term (37–42 weeks of gestation) at the Royal Alexandra Hospital in Edmonton, Alberta. From the fetal membranes, the amnion was separated from the choriodecidua and cut into 5 cm × 2 cm strips. The amnion tissue was then washed with prewarmed PBS before being incubated in EDTA-PBS (0.5 mM) for 15 min at room temperature. The tissue was washed again with PBS and dissociated in 60 mL prewarmed Dispase solution (2 g/L in PBS) for 45 min at 37 °C. The strips were then carefully transferred to DMEM (with high glucose and L-glutamine, without sodium pyruvate or phenol red) + 10% FBS + antibiotics/antimycotics (10,000 U/mL penicillin, 10,000 U/mL streptomycin, 25 μg/mL Amphotericin B) and vigorously shaken for 4 min to dissociate the epithelial cells. The resulting medium was centrifuged at 900 relative centrifugal force (rcf) for 10 min at room temperature, and the cell pellet was resuspended in DMEM + 10% FBS + antibiotics/antimycotics and filtered through a 40 μm filter. The cell suspension was diluted to a final concentration of $4 \times 10^5$ cells/mL and plated in six-well plates. The cells were cultured at 37 °C with 5% $CO_2$ until 80–90% confluence, with a fresh medium change every 48 h. At 90% confluence, the cells were serum-starved by replacing the medium

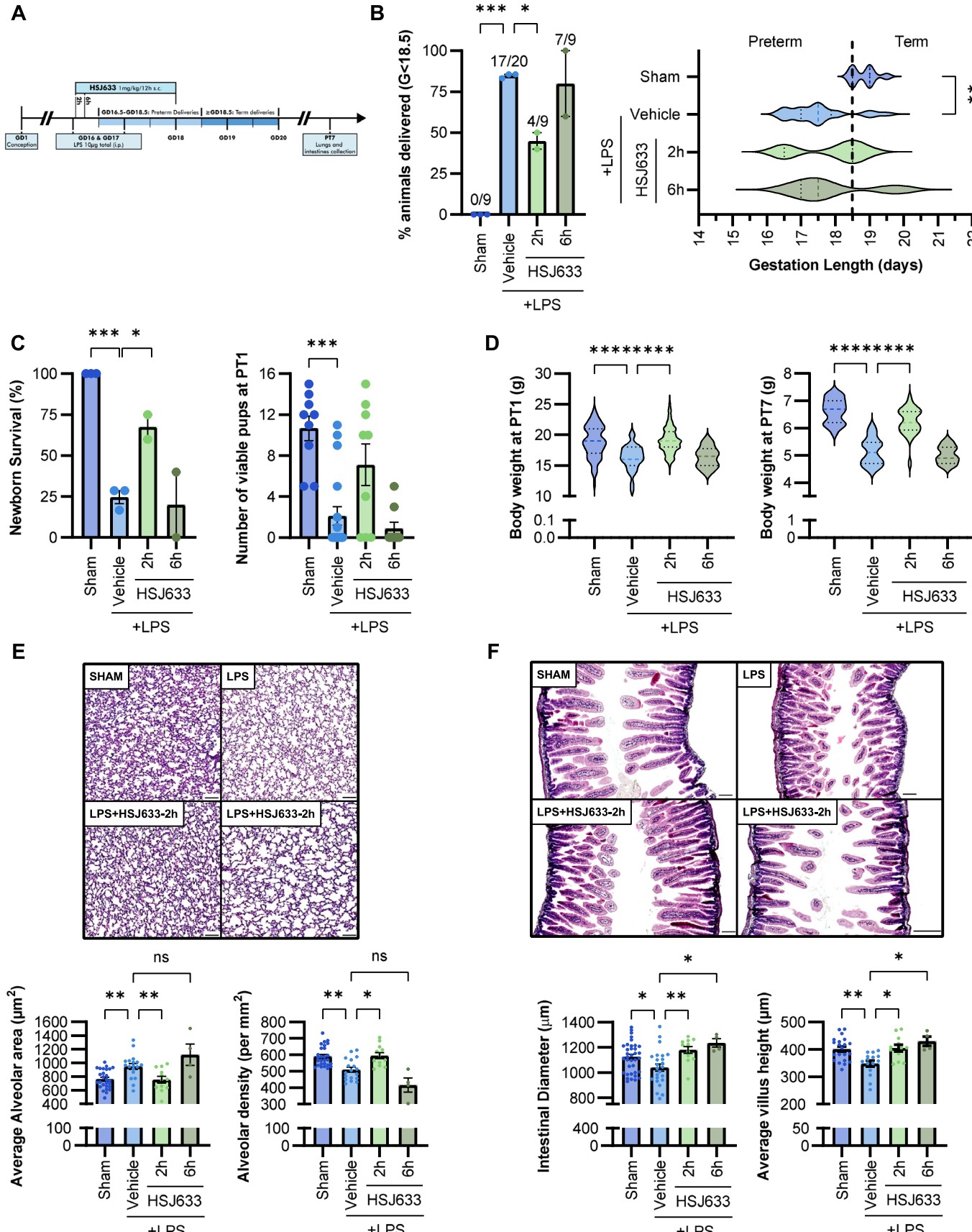

◄ **Figure 7. HSJ633 administered 2 h after inducing preterm labor enhances neonatal survival and preserves organ integrity in the progeny.**

(A) Schematic experimental timeline. An intraperitoneal (i.p.) injection of LPS (4 µg on GD16 + 6 µg on GD17 LPS/ 100 µL saline, total 10 µg) was administered. HSJ633(2 mg/kg/day), a murine anti-IL-6 receptor or vehicle is administered subcutaneously (s.c.) (repeated injection twice a day until GD18) 2 and 6 h after the LPS injection. (B) Prematurity rate (<18.5 days gestation) of mice in our LPS-induced PTB model. Values are mean ± SEM of 2–3 different experiments with 2–7 dams per group per experiment. One-way ANOVA with Dunnett's multiple comparisons test compared to the LPS + vehicle group. Gestational length of mice in the LPS-induced PTB model. In total, 9–20 dams per group. Kruskal–Wallis test with Dunn's multiple comparisons test compared to the LPS + vehicle group. *$P < 0.05$, **$P < 0.01$, ***$P < 0.001$ (C) Neonatal survival rate and viable pups at PT1. Values are mean ± SEM of 2–3 different experiments with 2 to 7 dams per group per experiment. If two or more pups are alive, the litter is considered viable. One-way ANOVA with Dunnett's multiple comparisons test compared to LPS + vehicle group for neonatal survival and Kruskal–Wallis test with Dunn's multiple comparisons test compared to the LPS + vehicle group for the number of viable pups at PT1. *$P < 0.05$, ***$P < 0.001$. (D) Pup body weights at PT1 and PT7 (age corrected). Values are mean ± SEM of 8–73 pups for left panel and 7–24 pups for right panel with one-way ANOVA with Dunnett's multiple comparisons (PT7) or Kruskal–Wallis test with Dunn's multiple comparisons (PT1) test compared to the LPS + vehicle group. ****$P < 0.0001$. (E) Representative histological images of lung parenchyma stained with hematoxylin and eosin. Lungs were collected at PT7. Scale bars = 100 µm. Measurement of alveolar density (1 mm$^2$) and average alveolar area (µm$^2$) quantified on regions of interest of 1 mm$^2$ are presented on histograms to the right of histology panels. Values are mean ± SEM of 4–27 pups per group. Kruskal–Wallis test with Dunn's multiple comparisons test compared to the LPS + vehicle group for the right panel and one-way ANOVA with Dunnett's multiple comparisons compared to the LPS + vehicle group for the left panel. *$P < 0.05$, **$P < 0.01$. (F) Representative histological images of intestines collected at PT7 and stained with hematoxylin and eosin; scale bars = 100 µm. Intestinal diameters (µm) of the regions of interest were quantified; villus height (µm) was measured using ZEN software: ten villi by region of interest were compiled; quantifications are presented on histograms to the right of images. Values are mean ± SEM of 4–38 pups per group. One-way ANOVA with Dunnett's multiple comparisons test compared to LPS + vehicle group was performed when the distribution was normal and Kruskal–Wallis test with Dunn's multiple comparisons test compared to the LPS + vehicle group when data did not display normal distribution. *$P < 0.05$, **$P < 0.01$, ***$P < 0.001$. Source data are available online for this figure.

with DMEM + antibiotics/antimycotics for 24 h before experimentation.

## Amniotic epithelial cells (AEC) treatment

Following 24 h serum starvation, AECs were treated with human recombinant IL-6 (100 ng/mL) for 15 min or 6 h with or without 1 µM HSJ633 or TCZ. Samples receiving HSJ633 or TCZ treatment were pre-treated with the same concentration of the antagonist for 1 h prior to agonist stimulation. IL-6 treatments were also supplemented with or without sIL-6R (100 ng/mL) to account for endogenous sIL-6R lost or washed away during tissue collection and culture procedures. Treatments were terminated by removing the cell culture medium. Total proteins were immediately isolated from AECs treated for 15 min for protein phosphorylation analysis. Total proteins were isolated by shaking the cells in RIPA buffer + cOmplete, EDTA-free Protease Inhibitor Cocktail final concentration 1× for 15 min, then centrifuging the cell lysate at 12,000 rcf, 4 °C for 10 min. Proteins were stored at −80 °C until analysis by immunoblotting.

## Generation of transgenic Ba/F3 cell lines

cDNAs coding for mouse gp130 (IMAGE ID 6834623) and mouse IL6Rα (IMAGE ID 5135770) were acquired from Open Biosystems. All cDNAs were subcloned into the pMX retroviral expression vector (Kitamura et al, 2003). Standard molecular biology methods were used to generate all plasmids. Cells from the IL3-dependent murine pro-B cell line Ba/F3 were transduced in the presence of polybrene (8 µg/ml) using VSV-G pseudo-typed retroviruses. Stable transgenic Ba/F3 derivatives expressing mouse gp130 or an empty vector (polylinker) construct were selected with puromycin (1 µg/ml). Ba/F3 cells expressing both IL6Rα and gp130 (IL6R) were positively selected using mouse IL6 for mIL-6R.

## Cell culture—HEK293, HEK-Blue IL-6, Ba/F3 cells

HEK293 and HEK-blue IL-6 cells were purchased and were used at fewer than 15 passages and tested mycoplasma negative. Cells were cultured in DMEM growth medium supplemented with 10% fetal bovine serum (FBS), 50 U/mL penicillin, and 50 µg/mL streptomycin. HEK-blue IL-6 cell media was also supplemented with HEK-Blue Selection (1:250). Ba/F3-polylinker, Ba/F3-mgp130, and Ba/F3-mgp130-IL-6Rα cells were cultured in RPMI 1640 growth medium supplemented with 10% FBS, 50 U/mL penicillin, and 50 µg/mL streptomycin. These cells were maintained in murine IL-3 at 10 ng/mL and puromycin at 1 µg/mL. Cells were cultured in standard experimental conditions (37 °C, 5% CO$_2$).

## IL-6 dose–response curves in vitro

For in vitro dose–response experiments, HEK-blue IL-6 cells were allowed to acclimate for two passages to complete the stabilization of the receptor before experiments. Prior to experiments, cells at 70% of confluence were added to six-well plates, then starved overnight before media was changed 30 min prior to beginning the stimulation. After the 30 min pre-incubation, the cells were treated with HSJ633 (from $1 \times 10^{-12}$ to $1 \times 10^{-4}$ M) allowed to sit for 15 min, treated with 3.86 nM of IL-6, and incubated for 6 h. Cell lysis and mRNA extraction were performed by adding RiboZol. Samples were stored at −20 °C until RNA extraction and quantification (see section below).

## HSJ633 iodine labeling

The Bolton–Hunter reaction was utilized to conjugate the N-hydroxysuccinimide ester of 3-(4-hydroxyphenyl) propionic acid, a tyrosine-like residue, to primary amines as per the manufacturer's instructions. Briefly, the peptide was dissolved in modification buffer (pH 9.0), then 100 µL of Bolton–Hunter Reagent was added and incubated for 3 h with periodic mixing. After incubating, the solution was dialyzed in PBS using a PlusOne Mini Dialysis Kit. Then 25 µL of peptide, 80 µL of PBS, and 25 µL of I$^{125}$ were combined in a precoated tube for 15 min. The reaction was stopped, and C-18 spin columns were used to perform washes and collect the iodine-marked peptide. Radioactivity measurements were made for each batch of radioactive peptides using a Gamma counter (Packard Cobra-II Auto Gamma Counter).

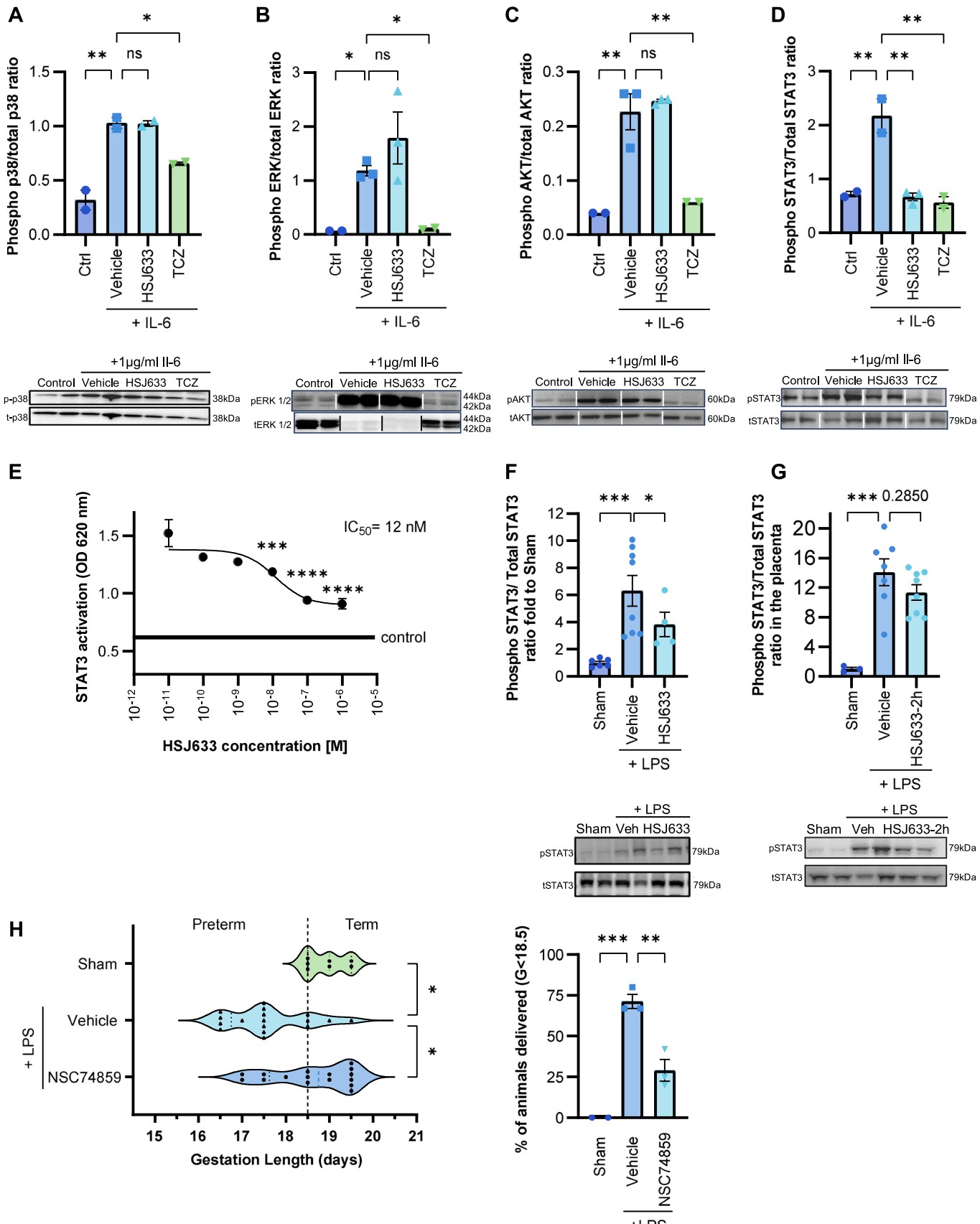

◄ **Figure 8. HSJ633 selectively inhibits STAT3 activation, critically involved in PTB.**

(A–D) Quantification of canonical IL-6R-coupled signaling in HEK-Blue IL-6 cells. Cells were pre-incubated with HSJ633 (1 μM), TCZ (3.44 μM) or vehicle, and subsequently stimulated with IL-6 (3.86 nM). Membranes were blotted for total and phospho-p38, ERK, AKT and STAT3. Values are mean ± SEM of 2–3 samples per group. One-way ANOVA with Dunnett's multiple comparisons test compared to IL-6 + vehicle group. *$P < 0.05$, **$P < 0.01$. (E) Dose-dependent effect of HSJ633 on phospho-STAT3 inhibition was quantified by QUANTI-Blue Assay. Values are mean ± SEM of 3–4 samples per group. ***$P < 0.001$, ****$P < 0.0001$. (F, G) Proteins were extracted from the placenta 6 h after LPS injection. HSJ633 was given 30 min before LPS injection in (E) and 2 h after LPS injection in (F). Membranes were blotted for total and phospho-STAT3. Values are mean ± SEM of 3–8 dams per group. One-way ANOVA with Dunnett's multiple comparisons test compared to LPS + vehicle group. *$P < 0.05$, ***$P < 0.001$. (H) Gestational length and rate of prematurity ($< 18.5$ days gestation) in mice treated with STAT3 inhibitor, NSC74859 (5 mg/kg/day) injected s.c. in mice on GD16 after LPS stimulation. Values are mean ± SEM of 13–17 dams per group for gestation length and Kruskal–Wallis test with Dunn's multiple comparisons test compared to the LPS + vehicle group. Values are mean ± SEM of 2–3 different experiments with 3–7 dams per group per experiment. One-way ANOVA with Dunnett's multiple comparisons test compared to LPS + vehicle group for prematurity rate. **$P < 0.01$, ***$P < 0.001$. TCZ Tocilizumab. Source data are available online for this figure.

## Binding studies

The HEK-blue IL-6 cells were allowed to acclimatize for two passages to complete the stabilization of the receptor. Prior to experiments, media was changed to adjust for a maximum quantity of 500 μL per well. Subsequently, unlabeled HSJ633 ($10^{-4}$ M) was added and allowed to reach equilibrium for 15 min. HEK293 and HEK-Blue IL-6 cells were then treated with the labeled HSJ633 in a quantity equating to 50,000 CPM per well. The reaction was allowed to reach equilibrium by incubating for 1.5 h at room temperature with constant mixing. After the incubation, media was removed and three washes with PBS were performed. Cells were collected by adding 1% NaOH + 1% Triton-X100 and transferred into glass tubes. All samples were measured using a Gamma counter (Packard Cobra-II Auto Gamma Counter).

## IL-6 and hyper IL-6 stimulation of Ba/F3 cells for immunoblotting experiments

To determine the degree of STAT3 phosphorylation with IL-6 stimulation, Ba/F3 cells (Ba/F3-mgp130-IL6Rα; Ba/F3-polylinker; Ba/F3-mgp130) (500,000 per condition) were starved in RPMI 1640 for 4 h and treated with HSJ633 ($10^{-4}$ M) or vehicle for 30 min in RPMI 1640 alone. IL-6 (1.92 nM) was then added to the medium for 30 min to stimulate IL-6R pathways leading to STAT3 activation. To determine the degree of STAT3 phosphorylation with Hyper IL-6 stimulation (IL-6 + sIL-6R), Ba/F3-mgp130-IL6Rα cells (500,000 per condition) were starved in RPMI 1640 for 4 h. Hyper IL-6 (100 ng/mL) was combined with HSJ633 (0.1 mM) or Anti-IL-6R (200 μg/mL) 30 min before stimulation. These combinations or HSJ633 ($10^{-4}$ M) or vehicle was then added to the medium for 30 min to stimulate IL-6R pathways leading to STAT3 activation. Cells were centrifuged at 2000×$g$ for 10 min at 4 °C and washed twice with cold 1× PBS. The pellet was resuspended in RIPA buffer at a 1× concentration, cOmplete EDTA-free Protease Inhibitor Cocktail at a final concentration of 1×, and PMSF at 1 mM. Samples were shaken for 30 min followed by centrifugation at 16,000× $g$ at 4 °C for 20 min to collect the proteins. Proteins were stored at −80 °C until analysis by immunoblotting.

## IL-6 stimulation of HEK-Blue IL-6 cell for RT-qPCR and immunoblotting experiments

For RT-qPCR experiments, cells were plated in 6-well plates at 70% of confluence and starved overnight. Cells were treated respectively with HSJ633, HSJ630, HSJ631, HSJ634, HSJ635, HSJ639 ($1 \times 10^{-6}$ M), or TCZ (3.44 μM, based on validated doses (Sahraoui

et al, 2014)) for TNF-α expression. For hIL-1, hIL-6, and hTNF-α expression, cells were treated with HSJ633 using concentrations ranging from $10^{-11}$ to $10^{-4}$ M, allowing for equilibrium over 15 min. Cells were then treated with 3.86 μM of IL-6 for 6 h for both experiments. Cell lysis was performed by adding RIBOzol. Samples were stored at −20 °C until analysis by quantitative PCR. To determine the degree of phosphorylation of different proteins (STAT3, ERK, AKT, and p38), HEK-Blue IL-6 cells were treated with HSJ633 ($10^{-6}$ M), TCZ (3.44 uM) or vehicle for 30 min in DMEM only. Human IL-6 (3.86 nM) was added to the medium for 15 min to stimulate IL-6R pathways leading to STAT3, ERK, and AKT activation. For p38 activation, 25 min of stimulation with IL-6 was required. The wells were charged with radioimmunoprecipitation assay buffer (RIPA) at a concentration of 1×, cOmplete EDTA-free Protease Inhibitor Cocktail final concentration 1×, and phenylmethanesulfonyl fluoride (PMSF) 1 mM. Samples were shaken for 30 min followed by centrifugation at 12,000 RPM at 4 °C for 20 min to harvest the proteins. Proteins were stored at −80 °C until analysis by immunoblotting.

## Immunoblotting from cells and tissue

A total of 15 μL of protein from HEK-Blue IL-6 cells combined with Laemmli-SDS 1× (reducing) was loaded onto SDS-PAGE gel and electro-transferred onto polyvinylidene difluoride membranes. After blocking with BSA 5%-TBST for 1 h, membranes were incubated with the following antibodies (Ab): either mouse anti-phospho-STAT3 (1:1000), rabbit anti-phospho-AKT (1:1000), rabbit anti-phospho-ERK1/2 (1:1000), rabbit anti-phospho-p38 (1:1000), rabbit anti-cytochrome c oxidase 4 (1:5000) or mouse anti-GAPDH (1:20,000, Proteintech). After washing, membranes were incubated for 1 h with secondary antibodies conjugated to HRP (1:3000 Anti-rabbit IgG, HRP-linked Antibody and 1:5000 Goat Anti-Mouse IgG (H + L)-HRP Conjugate). Samples were normalized by membrane stripping (15 g glycine, 1 g SDS, 10 mL Tween-20 at pH 2,2) and incubation with Ab (mouse anti-STAT3 (1:1000), rabbit anti-AKT (1:1000), rabbit anti-ERK1/2 (1:1000) or rabbit anti-p38 (1:1000)). After washing, membranes were incubated for 1 h with secondary antibodies conjugated to HRP (1:3000 Anti-rabbit IgG, HRP-linked antibody and 1:5000 goat anti-mouse IgG (H + L)-HRP Conjugate). Western Lightning Plus ECL or Clarity MAX Western ECL Substrate were used for protein detection using the ChemiDoc MP, and densitometric analysis were completed with ImageLab from Bio-Rad Laboratories.

Protein lysates from Ba/F3 cells (10 μL), prepared in 1× Laemmli buffer under reducing conditions, were resolved by SDS-PAGE and

transferred onto polyvinylidene difluoride (PVDF) membranes. Following a 1-hour blocking step in 5% BSA in TBST at room temperature, membranes were incubated overnight with primary antibodies (goat anti-IL-6R (1:2000), rabbit anti-gp130 (1:1000), rabbit anti-phospho-STAT3 (1:1000), mouse anti-GAPDH (1:10,000), or mouse anti-β-actin (1:2000)). After washing, membranes were incubated for 1 h with secondary antibodies conjugated to HRP (1:3000 Anti-rabbit IgG, HRP-linked antibody or 1:5000 Goat Anti-Mouse IgG (H + L)-HRP Conjugate or 1:1000 Donkey anti-goat IgG-HRP antibody). Samples were normalized by membrane stripping (15 g glycine, 1 g SDS, 10 mL Tween-20 at pH 2.2) and incubation with an Ab (rabbit anti-STAT3 (1:1000)). Western Lightning Plus ECL or Clarity MAX Western ECL Substrate were used for protein detection using the ChemiDoc MP, and densitometric analysis were completed with ImageLab from Bio-Rad Laboratories.

For mouse placenta, 5 μL of total protein was combined with Laemmli-SDS 1× (reducing) was loaded onto SDS-PAGE gel and electro-transferred onto polyvinylidene difluoride membranes. After blocking with BSA 5%-TBST for 1 h, membranes were incubated with an antibody (rabbit anti-phospho-STAT3 (1:1000) or rabbit anti-cytochrome c oxidase 4 (1:5000)). After washing, membranes were incubated for 1 h with secondary antibodies conjugated to HRP (1:3000 Anti-rabbit IgG, HRP-linked antibody). Samples were normalized by membrane stripping (15 g glycine, 1 g SDS, 10 mL Tween-20 at pH 2.2) and incubation with an Ab (rabbit anti-STAT3 (1:1000)). Western Lightning Plus ECL or Clarity MAX Western ECL Substrate were used for protein detection using the ChemiDoc MP, and densitometric analysis were completed with ImageLab from Bio-Rad Laboratories.

For treatment on amniotic epithelial cells, total protein (45 μL) was combined with 1× loading buffer (250 mM Tris-HCl containing 4% sodium dodecyl sulfate, 10% glycerol, 2% β-mercaptoethanol, and 0.002% bromophenol blue), separated by SDS-PAGE and electro-transferred onto nitrocellulose membranes. Membranes were incubated with Intercept (PBS) Blocking Buffer for 1 h at room temperature before being incubated with the following primary antibodies: mouse anti-phospho-STAT3 (1:1000), mouse anti-STAT3 (1:1000), or mouse anti-GAPDH (1:5000, Thermo Fisher Scientific). After washing, the membranes were then incubated in 680RD donkey anti-mouse antibodies (1:5000 from LI-COR Biosciences) for 1 h at room temperature. Protein bands were detected and quantified using the Odyssey LI-COR Biosciences Infrared Imaging System and application software V3.0.

## STAT3 QUANTI-Blue assay

HEK-Blue IL-6 cells were treated with HSJ633 ($10^{-11}$ to $10^{-6}$ M) for 30 min before the IL-6 stimulation (3.86 nM). After IL-6 stimulation, cells were incubated for 4 h at 37 °C. Levels of secreted alkaline phosphatase in cell culture supernatant were determined by the QUANTI-Blue assay, according to the manufacturer's instructions. The plate was read at a wavelength of 620–655 nm using a microplate reader.

## LPS-induced PTB models

Timed pregnant CD-1 mice at days 16 and 17 of gestation were anesthetized with isoflurane and given intraperitoneal (i.p.) injections of LPS (4 μg in 100 μL saline on GD16 and 6 μg in

100 μL saline on GD17, total 10 μg) or vehicle (100 μL). Doses of LPS were selected based on previous reports (Chang et al, 2011; Elovitz and Mrinalini, 2004; Fidel et al, 1994; Sadowsky et al, 2006; Salminen et al, 2008; Urakubo et al, 2001), preliminary dose–response experiments, and knowing that a systemic maternal infection is associated with the onset of labor (Romero et al, 2001). A total of 100 μL of HSJ633 or HSJ639 (2 mg/kg/day), an anti-IL-6R antibody (800 μg/kg/day) or vehicle was injected subcutaneously (s.c.) 30 min before injections of LPS (to allow distribution of drugs to target tissues) and repeated twice daily until GD18.5.

To determine the maximal delay of administration of HSJ633, LPS was injected as previously described, and HSJ633 was either administered 2 h or 6 h after the LPS injection at GD16. Dams were consistently observed for delivery until term (G18.5–19.5), and the body weight and survival rate (if 2 or more pups are alive, the litter is considered alive) of the neonates were recorded. The prematurity rate was assessed by determining the number of dams delivering prematurely (gestation length before 18.5 days) out of the total number of dams in the same group. The pups were sacrificed at PT7 for histological analysis.

Fetomaternal analyses were conducted according to the following model: pregnant CD-1 mice on day 16 were injected intraperitoneally with a total of 10 μg of LPS in 100 μL of saline. The same conditions (prophylaxis and treatment), as well as the dose of HSJ633, are described earlier. The animals were sacrificed 30 h (G17.25) after the LPS injection, and an intracardiac puncture was performed to collect maternal systemic blood, which was deposited in tubes containing heparin to prevent clotting. Blood plasma (obtained by centrifugation), amniotic fluid, uterus, placenta, fetal membranes, as well as lungs and intestines of fetal samples, were snap frozen in liquid nitrogen and kept at −80 °C for subsequent RNA purification or protein extraction.

In a separate experiment, placentas from CD-1 mice exposed to a total of 10 μg of LPS with or without HSJ633 in prophylaxis, as well as HSJ633-2h, were also collected 6 h post-LPS injection to assess STAT3 activation by western blot.

In order to show that HSJ633 administration takes place after the induction of inflammation in gestational tissues, a time course of LPS was conducted. Gestational tissues (placenta, uterus, and fetal membranes) were collected at 1 h, 2 h, and 6 h post-LPS injection (totaling 10 μg in 100 μL saline i.p. on GD16) for mRNA extraction.

## STAT3 inhibitor and soluble gp130 efficacy in LPS-induced PTB

Thirty minutes before administering LPS, on GD16, NSC74859 (5 mg/kg/day solubilized in 20% DMSO + water, injected twice a day s.c.) or rmgp130/Fc Chimera (10 μg i.p. on GD16 and 10 μg s.c. on GD16.5) were injected into distinct dams. The same LPS-induction PTB model as described above was used for these experiments (4 μg on GD16 + 6 μg on GD17, i.p.). Delivery was assessed until term (GD18.5–19.5).

## Tissue and cell RNA extraction and reverse transcription-quantitative PCR

Samples (30 mg) of uterus, placenta, fetal membranes, as well as the fetal lung, brain, and intestine, were thawed and deposited in lysis buffer according to the manufacturer's protocol (Qiagen RNeasy

Plus Mini Kit). Cell lysis and RNA extraction were performed directly into RiboZol. RNA was extracted from tissue fragments and cells according to the manufacturer's protocol. RNA was quantified with a NanoDrop 1000 spectrophotometer and samples (500 ng) were treated with DNAse I (1U). RNA was used to synthesize cDNA using iScript reverse transcription superMix. Quantitative gene expression analysis was performed on a Light Cycler 96 with iTaq Universal SYBR Green Supermix. Gene expression levels were normalized to Actb or 18S. Primer sequences are provided in Table EV1.

## Protein extraction and murine ELISA

For immunoblotting, 50 mg of placentas was placed in RIPA buffer final concentration 1×, cOmplete, EDTA-free Protease Inhibitor Cocktail final concentration 1×, and phenylmethanesulfonyl fluoride (PMSF) final concentration 1 mM. Tissues were disrupted by sonication. Proteins were obtained by centrifugation at $10,000 \times g$, 5 min at 4 °C. Proteins were stored at $-80$ °C until analysis by immunoblotting.

For ELISA, samples (60 mg) of the uterus, placentas, and fetal membranes were placed in RIPA buffer final concentration 1×, cOmplete, EDTA-free Protease Inhibitor Cocktail final concentration 1×, and phenylmethanesulfonyl fluoride (PMSF) final concentration 1 mM. Tissues were disrupted by sonication. Proteins were obtained by centrifugation at $12,000 \times g$, 20 min at 4 °C. ELISAs were performed according to the manufacturer's protocols for IL-1β/IL-1F2 Quantikine, mouse C-Reactive Protein/CRP, mouse IL-6 Quantikine, and mouse TNF-α Quantikine kits. The reaction was stopped, and the plate was read at 450 nm with a wavelength correction of 540 nm.

## Lung and intestine histology

Histology was performed as described by Habelrih et al (Habelrih et al, 2023). Images were captured using ×10 objective with a Zeiss AxioScan.Z1.

## HSJ633 biodistribution

HSJ633-FITC (1 mg/kg), FITC alone (1 mg/kg), LPS (10 µg i.p.) + FITC (1 mg/kg), LPS (10 µg i.p.) + HSJ633-FITC (1 mg/kg) or vehicle were given by s.c. injection to pregnant mice on GD17. Mice were euthanized after 4 h to analyze tissue distribution of the fluorescent-tagged compounds. Placentas were collected and fixed in 4% paraformaldehyde for 1 day and transferred into 30% sucrose for another day. Placental cryosections were stained with DAPI (0.0002 mg/mL) for nuclei. Images were captured using a ×20 objective with a Zeiss AxioScan.Z1. Fluorescence analysis was performed using ImageJ.

## Statistical analysis

Analysis of data was performed using GraphPad Prism (version 10) software. Outliers were identified using the ROUT method with a Q value of 1% and excluded based on the test results. The normality of groups was assessed using the Shapiro–Wilk test. In cases of normally distributed data, a one-way ANOVA was utilized. For comparisons involving a single control, the Dunnett multiple comparison method was employed. In situations where data did not exhibit normal distribution, a Kruskal–Wallis test was applied

with Dunn's multiple comparison. Statistical significance was determined at a $P$ value of less than 0.05. Presentation of results varied depending on sample size means ± SEM were presented for normally distributed data and for non-normally distributed data. A one-way ANOVA was conducted despite the small sample size in a very limited number of experiments due to practical constraints in data collection. The analysis was deemed necessary to address the primary research question and explore potential group differences. While small sample sizes may limit statistical power, results were interpreted cautiously, and additional metrics, such as effect sizes, were provided to support the findings. The images and analysis for biodistribution and for histology were conducted under blinded conditions, without knowledge of group assignments.

## Graphics

Figure EV5C was created using BioRender.com, while Figs. 1B,D, 7A, and synopsis were made by Kevin Sawaya with Adobe Illustrator. All other graphs were generated using GraphPad Prism.

# Data availability

ROI images of intestines and lungs are available on BioImages accession number S-BIAD1505, no access code needed.

## The paper explained

### Problem
Preterm birth occurs before the 37th week of pregnancy and affects 15 million children worldwide each year. This early delivery leads to many complications for the newborn. For instance, organs like the lungs, brain, eyes, and intestines can be affected by being born too soon. Although the exact cause of preterm birth is unknown, there are several risk factors which converge on inflammation. The latter is shown to elicit labor and preterm birth. Current medications only tackle uterine contractions, but have no impact on deleterious inflammation, and thus are essentially ineffective in prolonging gestation or improving newborn outcome. Hence, discovery of safe medications that target in utero inflammation is highly relevant.

### Results
We have developed a small molecule called HSJ633, which acts as an antagonist to the interleukin-6 receptor, as it reduces inflammation in maternal and fetal tissues. When administered before or after the onset of labor, HSJ633 reduces preterm birth and neonatal mortality in pre-clinical models. It also improves birth outcomes by desirably increasing body mass and preserving the structural integrity of offspring's intestines and lungs. Importantly, HSJ633 is also effective in inhibiting inflammation in relevant human fetal membranes. HSJ633 is an unprecedented pharmacologically selective modulator of IL-6R, as it inhibits STAT3 activation while desirably preserving activity of cytoprotective signals—ERK, p38, and AKT.

### Impact
HSJ633 is groundbreaking in the context of preterm birth, by prolonging gestation and improving progeny outcome. HSJ633 represents a novel approach, which provides a safer profile to currently available biologics against IL-6.

The source data of this paper are collected in the following database record: biostudies:S-SCDT-10_1038-S44321-025-00257-9.

## Peer review information

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

## Acknowledgements

The authors thank the employees of the CHU Sainte-Justine animal facility. Funding was provided by CIHR (application number 421180). CIHR, FRQ, and Université de Montréal (Mérite Scholarship) funded this project by providing scholarships to the authors. FC, TH, EP, NC, and BF are recipients of bursaries from CIHR and/or FRQ. SC is the recipient of a Canada Research Chair and the Leopoldine Wolfe Chair in Translational studies on vision research.

## Author contributions

**France Côté**: Conceptualization; Formal analysis; Supervision; Validation; Investigation; Visualization; Methodology; Writing—original draft; Writing—review and editing. **Elizabeth Prairie**: Formal analysis; Validation; Investigation; Visualization; Project administration. **Estefania Marin Sierra**: Conceptualization; Formal analysis; Investigation. **Christiane Quiniou**: Conceptualization; Methodology; Project administration; Writing—review and editing. **Tiffany Habelrih**: Investigation; Writing—review and editing. **Wendy Xu**: Investigation. **Béatrice Ferri**: Investigation. **Xin Hou**: Investigation; Methodology. **Isabelle Lahaie**: Investigation; Project administration. **Nadia Côté**: Resources; Investigation. **Sarah-Eve Loiselle**: Investigation. **Laurence Gobeil**: Investigation. **Kevin Sawaya**: Investigation; Visualization; Writing—review and editing. **Aurélie Faucher**: Investigation. **Amélie Beaulieu**: Investigation. **Sandrine Delisle**: Investigation. **Marie-Pénélope Simard**: Investigation. **Mohammad Ali Mohammad Nezhady**: Writing—review and editing. **Véronique Laplante**: Resources; Investigation. **Allan Reuben**: Software. **Sidi Mohamed Kalaidji**: Investigation. **Emmanuel Bajon**: Writing—review and editing. **Gael Cagnone**: Investigation. **Kelycia B Leimert**: Methodology; Writing—review and editing. **Jean-François Gauchat**: Resources; Investigation. **Luc Gaudreau**: Resources. **Sarah Robertson**: Conceptualization; Funding acquisition; Writing—review and editing. **William D Lubell**: Conceptualization; Funding acquisition; Writing—review and editing. **David M Olson**: Conceptualization; Funding acquisition; Methodology; Writing—review and editing. **Sylvain Chemtob**: Conceptualization; Funding acquisition; Methodology; Writing—review and editing.

Source data underlying figure panels in this paper may have individual authorship assigned. Where available, figure panel/source data authorship is listed in the following database record: biostudies:S-SCDT-10_1038-S44321-025-00257-9.

## Disclosure and competing interests statement

The authors declare no competing interests.

# Expanded View Figures

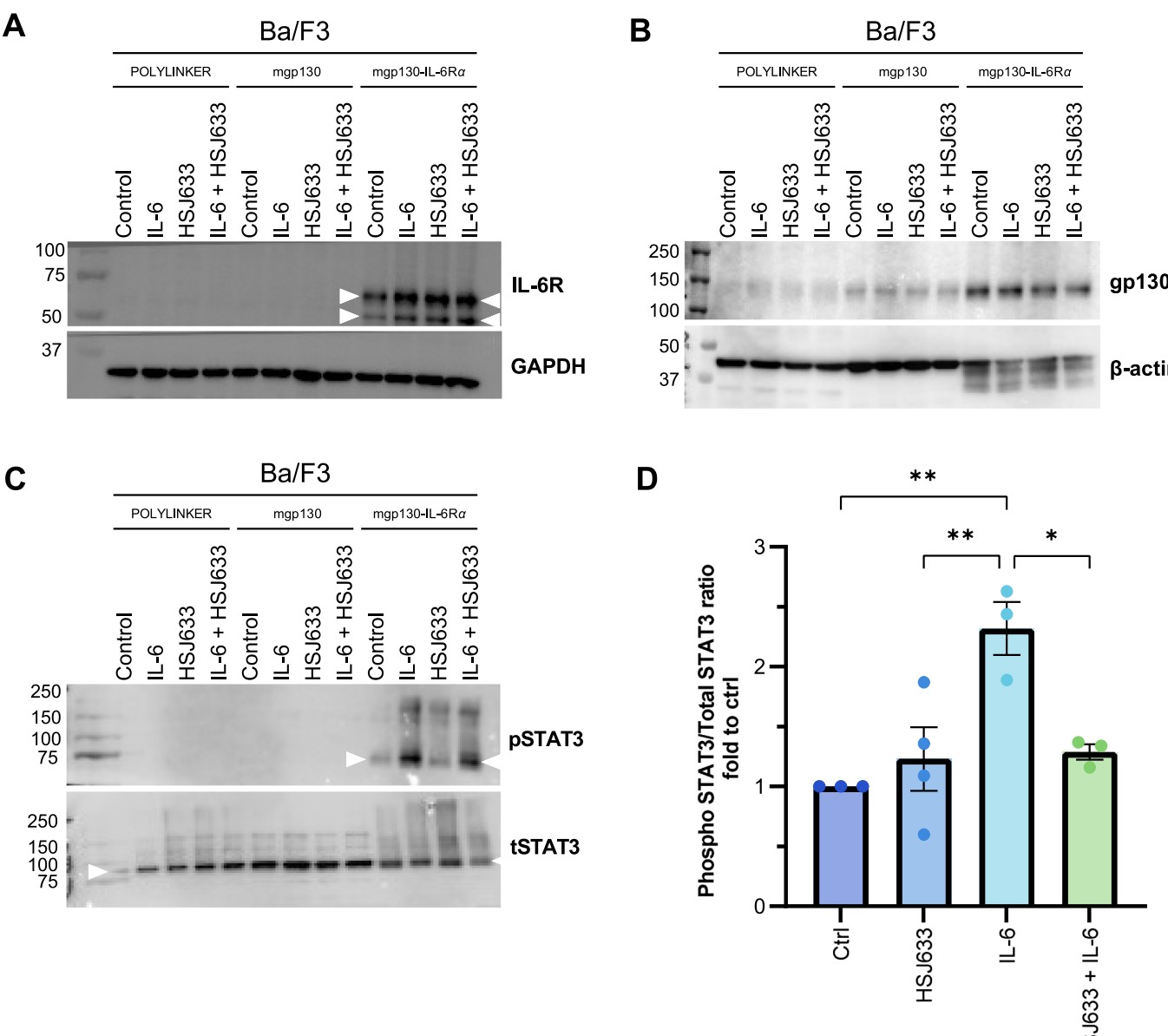

**Figure EV1.  HSJ633 reduces STAT3 activation in murine Ba/F3-mgp130-IL-6Rα cells but not in murine cells without IL-6Rα.**

Representative Western blot displaying the levels of IL-6Rα, gp130 and STAT3 in Ba/F3-polylinker, Ba/F3-mgp130, and Ba/F3-mgp130-IL-6Rα cells treated with IL-6 (1.92 nM), HSJ633 (0.1 mM) and HSJ633 + IL-6. (A) Western blot image showing IL-6Rα and GAPDH bands. (B) Western blot image showing gp130 and β-actin bands. (C) Western blot image showing phospho-STAT3 and total STAT3 bands. (D) Quantification of STAT3 activation in Ba/F3-mgp130-IL-6Rα, presented as mean ± SEM of 3-4 samples per group. Densitometric analysis of the bands indicates a significant reduction in STAT3 activation in HSJ633 + IL-6-treated cells relative to IL-6. One-way ANOVA with Dunnett's multiple comparisons test compared to IL-6. *$P < 0.05$, **$P < 0.01$.

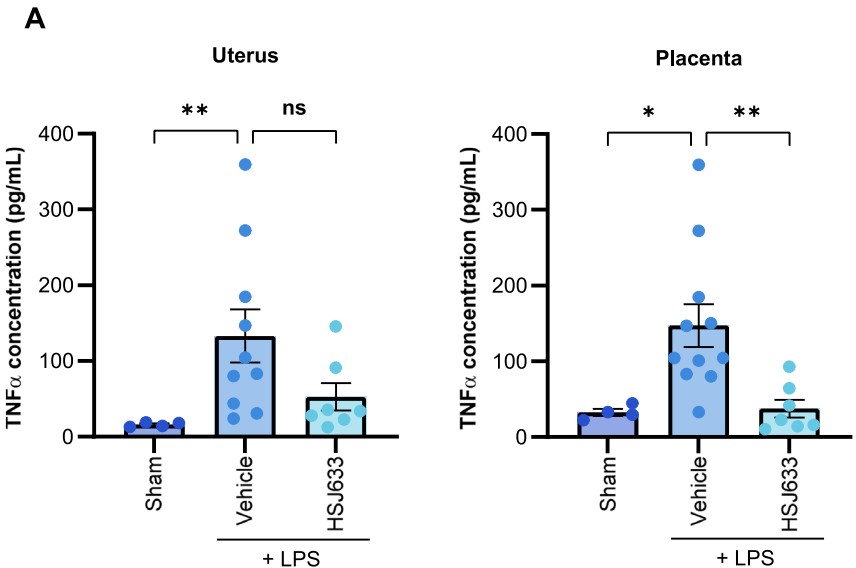

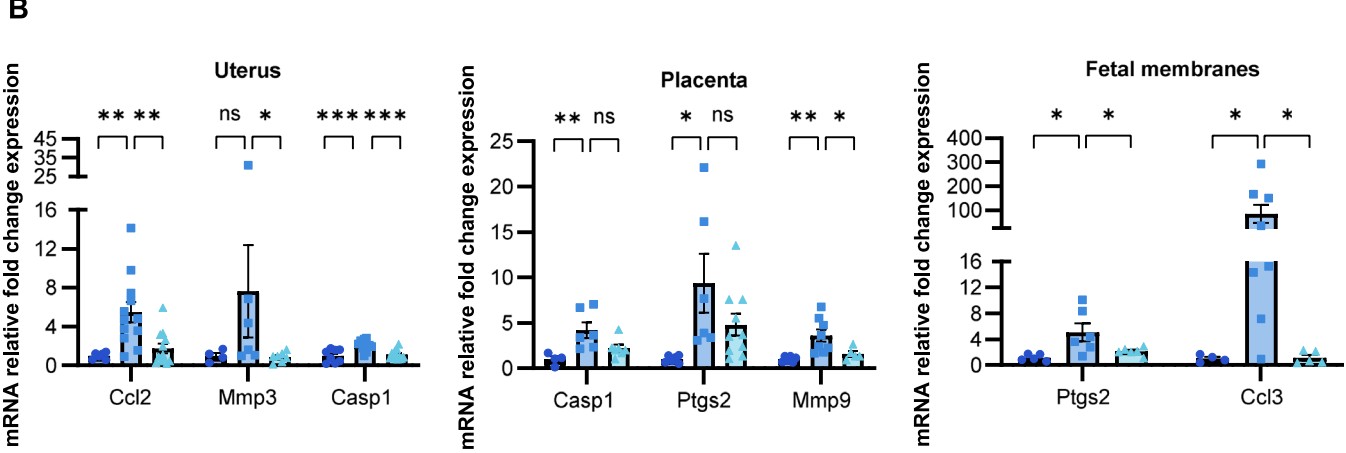

**Figure EV2.  Prophylactic administration of HSJ633 reduces inflammation in reproductive tissues.**

(A) TNF-α concentrations (quantified by ELISA) in the uterus and placenta on GD17.25. Values are mean ± SEM of 4–10 dams per group and one-way ANOVA with Dunnett's multiple comparisons test compared to LPS + vehicle group was performed for the placenta. Values are mean ± SEM of 4–11 dams per group and Kruskal–Wallis test with Dunn's multiple comparisons test compared to the LPS + vehicle group was performed for the uterus. *$P < 0.05$, **$P < 0.01$. (B) mRNA expression on GD17.25 in the uterus, placenta, and fetal membranes of various genes involved in PTB (normalized to Actb). Values are mean ± SEM of 4–12 dams per group for the uterus, 4–10 dams per group in the placenta and 4–8 dams per group for fetal membranes. *$P < 0.05$, **$P < 0.01$, ***$P < 0.001$,

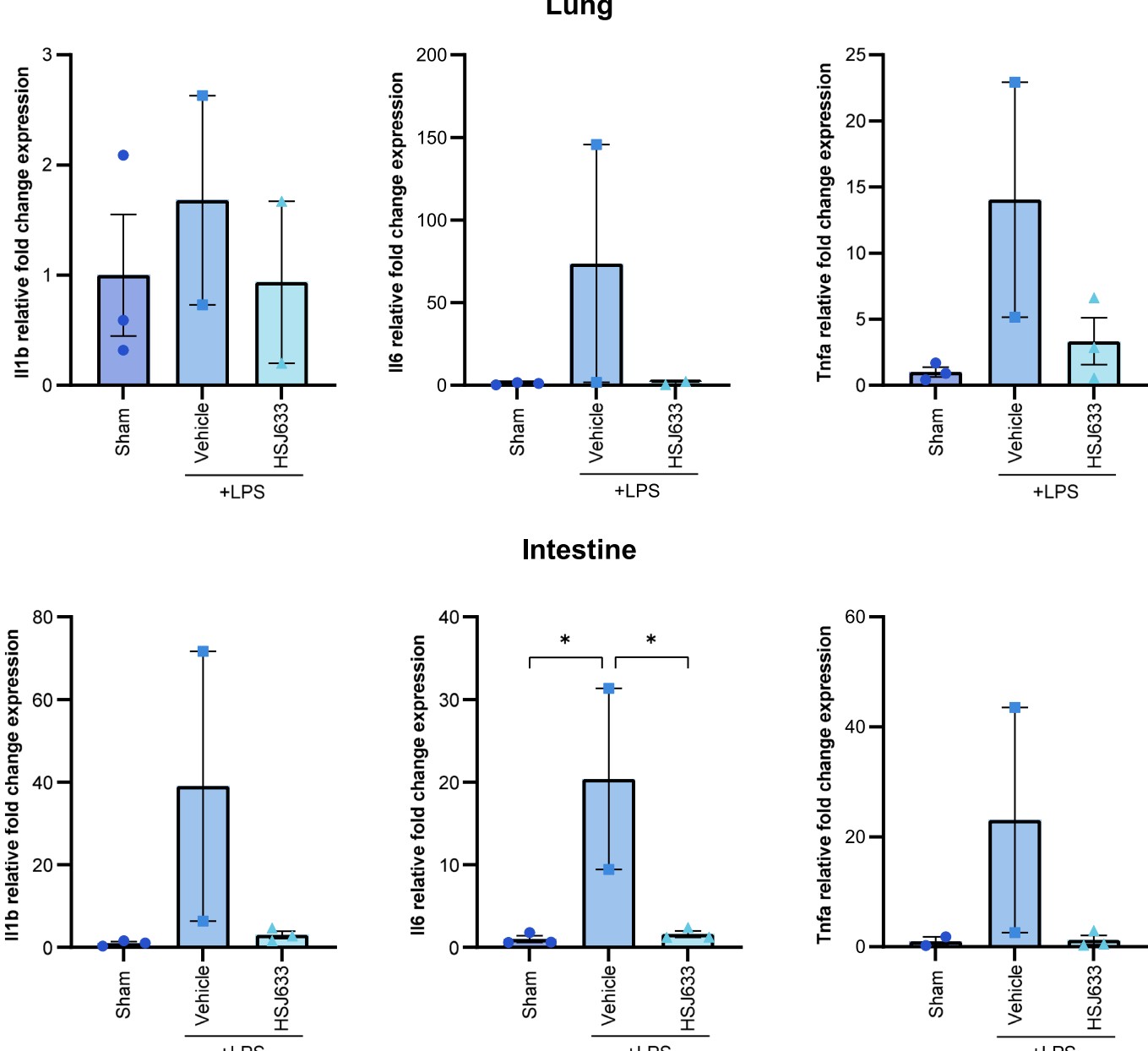

**Figure EV3.  LPS-induced inflammatory cytokines in fetal organs.**

mRNA expression (determined by RT-qPCR) of IL-1β, IL-6 and TNF-α in murine lung and intestine collected on GD17.25 after LPS induction (normalized to Actb). Values are mean ± SEM of 2–5 samples per group. One-way ANOVA with Dunnett's multiple comparisons test compared to LPS + vehicle group. *P < 0.05.

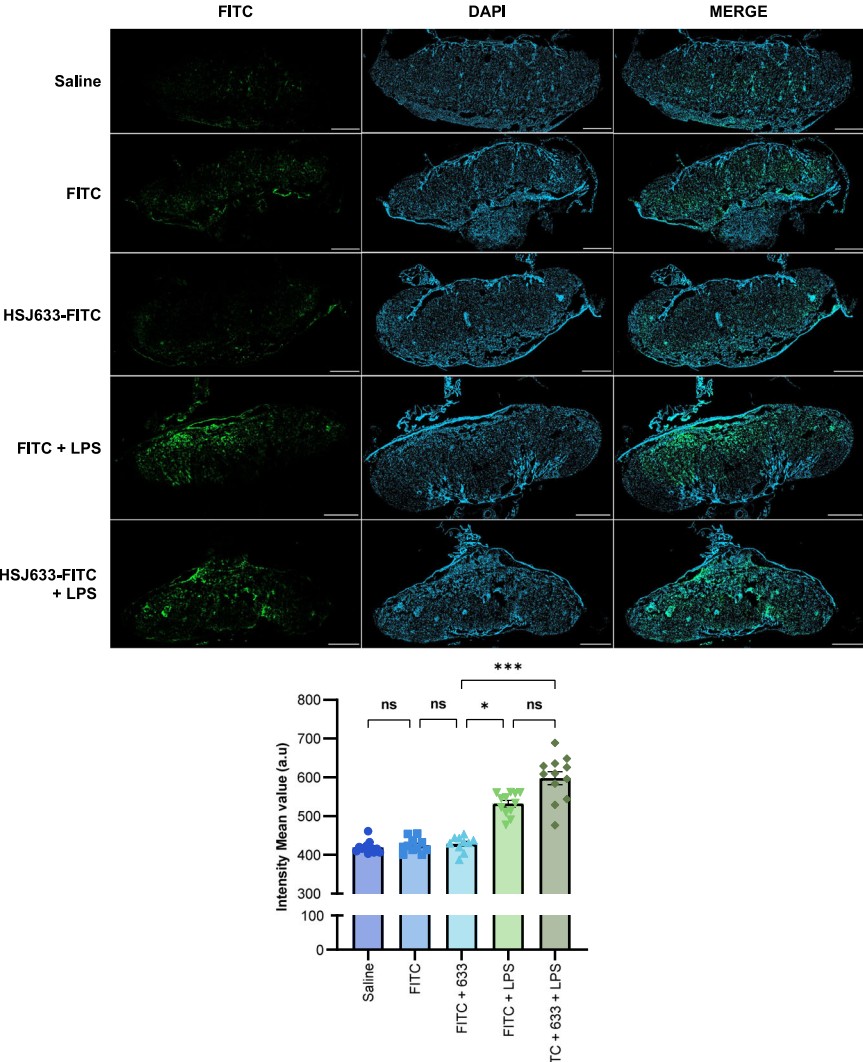

Figure EV4.   Distribution of HSJ633 to fetal placental compartment.

Biodistribution of prophylactic HSJ633-FITC 4 h post subcutaneous injection in pregnant mice on GD17 in the placenta. Green fluorescence was quantified. Values are mean ± SEM of 11–12 samples per group. 1000 μm scale bars. *$P < 0.05$, ***$P < 0.001$ by the Kruskal–Wallis test with Dunn's multiple comparisons test compared to the LPS + vehicle group.

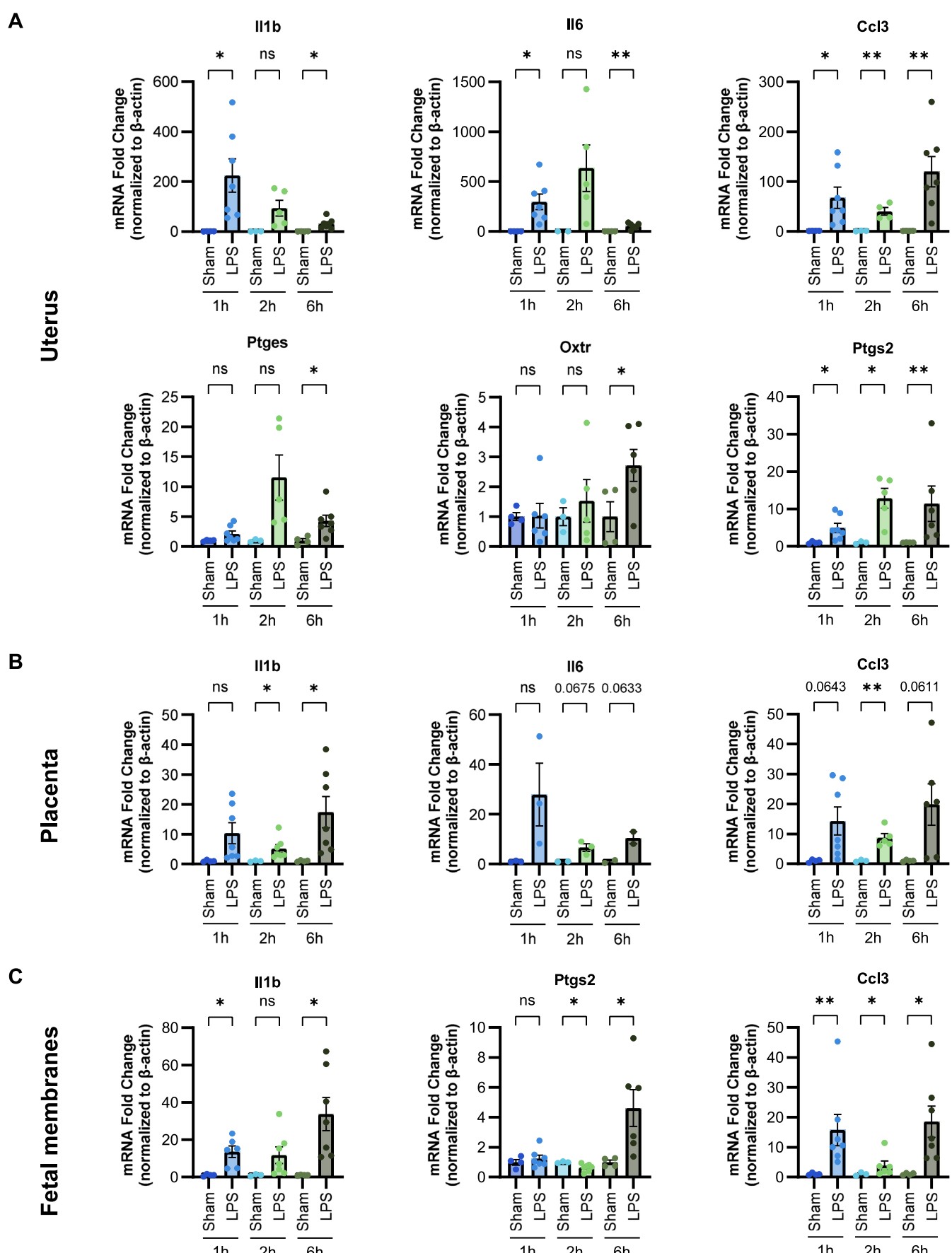

**Figure EV5. LPS induces an inflammatory response as early as 1 h post-administration in gestational tissues.**

(A–C) mRNA expression (determined by RT-qPCR) of IL-1β, IL-6, Ccl3, Ptgs2, Ptges, and Oxtr in the uterus, placenta, and fetal membranes collected 1 h, 2 h and 6 h after LPS injection on GD16 (normalized to Actb). Values are mean ± SEM of 3–7 samples per group. Unpaired *T* test compared to sham group at each time point with Mann–Whitney comparison when data were not distributed normally. *$P < 0.05$, **$P < 0.01$.

**A**

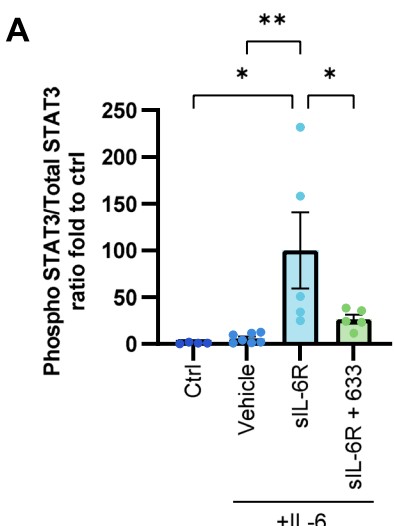

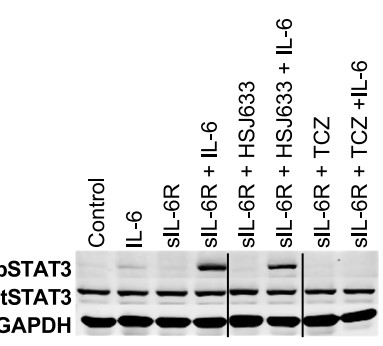

**B**

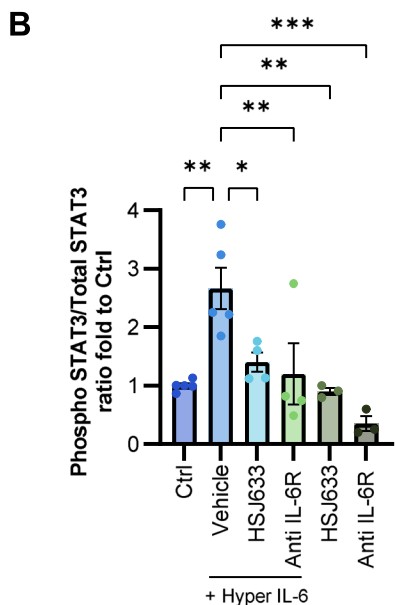

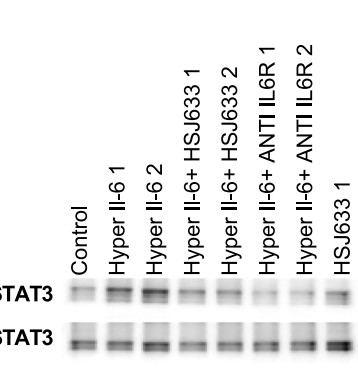

**C**

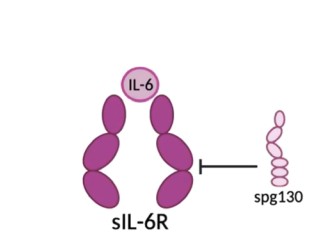

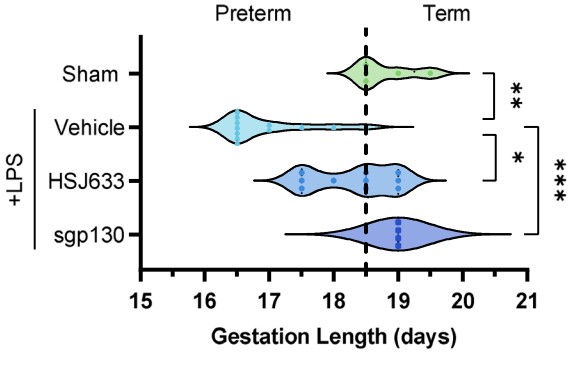

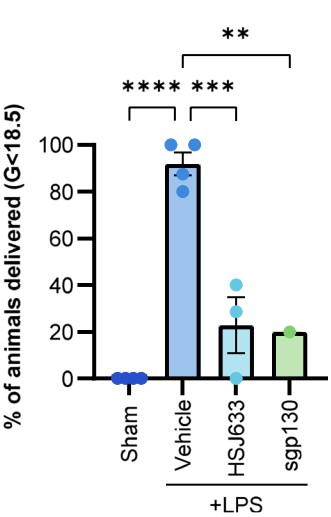

◄ **Figure EV6. Importance of soluble IL-6R in human amniotic epithelial cells and in Ba/F3-mgp130-IL-6Rα IL-6-triggered signaling and in LPS-induced PTB.**

(A) Amniotic epithelial cells were stimulated with IL-6 and/or sIL-6R (100 ng/mL for both) in absence or presence of HSJ633 (1 μM), TCZ (3.44 μM), or vehicle. Membranes were blotted for phospho-STAT3 and normalized to total STAT3. Representative Western blot gel on the right and compiled histogram data on the left. Values are mean ± SEM of 4–7 samples per group. One-way ANOVA with Dunnett's multiple comparisons test compared to sIL-6R + IL-6 group. (B) Quantification of STAT3 activation in Ba/F3-mgp130-IL-6Rα treated with Hyper IL-6 (100 ng/mL), HSJ633 (0.1 mM) and Anti-IL-6R (200 μg/mL). Representative Western blot displaying the levels of STAT3 in Ba/F3-mgp130-IL-6Rα cells treated with hyper IL-6. Quantification of STAT3 activation presented as mean ± SEM of 3–5 samples per group. Densitometric analysis of the bands indicates a significant reduction in STAT3 activation in HSJ633+Hyper IL-6-treated cells relative to Hyper IL-6. One-way ANOVA with Dunnett's multiple comparisons test compared to Hyper IL-6. *$P < 0.05$, **$P < 0.01$, ***$P < 0.001$. (C) Gestational length (left panel), data are presented as mean ± SEM of 4–12 dams per group and Kruskal–Wallis test with Dunn's multiple comparisons test compared to the LPS + vehicle group for gestation length. Rate of prematurity (born on GD < 18.5; right panel) in mice treated with soluble gp130 (sgp130) injected s.c. on GD16. Data are presented as mean ± SEM of 1–4 different experiments with 2 to 8 dams per group per experiment. One-way ANOVA with Dunnett's multiple comparisons test compared to the LPS + vehicle group. *$P < 0.05$, **$P < 0.01$, ***$P < 0.001$, ****$P < 0.0001$. TCZ Tocilizumab.

