## [Peer Review File · EMBO Molecular Medicine]

A novel modulator of IL-6R prevents inflammation-induced preterm birth and improves newborn outcome

France Côté, Elizabeth Prairie, Estefania Marin Sierra, Christiane Quiniou, Tiffany Habelrih, Wendy Xu, Béatrice Ferri, Xin Hou, Isabelle Lahaie, Nadia Côté, Sarah-Eve Loiselle, Laurence Gobeil, Kevin Sawaya, Aurélie Faucher, Amélie Beaulieu, Sandrine Delisle, Marie-Pénélope Simard, Mohammad Ali Mohammad Nezhady, Véronique Laplante, Allan Reuben, Sidi Mohamed Kalaidji, Emmanuel Bajon, Gael Cagnone, Kelycia Leimert, Jean-François Gauchat, Luc Gaudreau, Sarah Robertson, William Lubell, David Olson, and Sylvain Chemtob

Corresponding author: Sylvain Chemtob (sylvain.chemtob@umontreal.ca)

Review Timeline:

Submission Date:	4th Jul 24
Editorial Decision:	14th Aug 24
Revision Received:	24th Jan 25
Editorial Decision:	12th Feb 25
Revision Received:	11th May 25
Editorial Decision:	14th May 25
Revision Received:	17th May 25
Accepted:	20th May 25

Editor: Zeljko Durdevic

Transaction Report:

14th Aug 2024

Dear Dr. Chemtob,

Thank you for the submission of your manuscript to EMBO Molecular Medicine and please accept my apologies for the delay in getting back to you due to the holiday season. We have now received feedback from the two reviewers who agreed to evaluate your manuscript. Both referees recognize interest of the study but also raise important concerns that should be addressed in a major revision. If you would like to discuss further the points raised by the referees, I am available to do so via email or video. Let me know if you are interested in this option.

We would welcome the submission of a revised version within three months for further consideration. Please let us know if you require longer to complete the revision.

Please use this link to login to the manuscript system and submit your revision: <https://embomolmed.msubmit.net/cgi-bin/main.plex>

I look forward to receiving your revised manuscript.

Yours sincerely,

Zeljko Durdevic

We require:

- 1) A .docx formatted version of the manuscript text (including legends for main figures, EV figures and tables). Please make sure that the changes are highlighted to be clearly visible.
- 2) Individual production quality figure files as .eps, .tif, .jpg (one file per figure). For guidance, download the 'Figure Guide PDF': (<https://www.embopress.org/page/journal/17574684/authorguide#figureformat>).
- 3) A .docx formatted letter INCLUDING the reviewers' reports and your detailed point-by-point responses to their comments. As part of the EMBO Press transparent editorial process, the point-by-point response is part of the Review Process File (RPF), which will be published alongside your paper.
- 4) A complete author checklist, which you can download from our author guidelines (<https://www.embopress.org/page/journal/17574684/authorguide#submissionofrevisions>). Please insert information in the checklist that is also reflected in the manuscript. The completed author checklist will also be part of the RPF.
- 5) Please note that all corresponding authors are required to supply an ORCID ID for their name upon submission of a revised manuscript.
- 6) It is mandatory to include a 'Data Availability' section after the Materials and Methods. Before submitting your revision, primary

datasets produced in this study need to be deposited in an appropriate public database, and the accession numbers and database listed under 'Data Availability'. Please remember to provide a reviewer password if the datasets are not yet public (see <https://www.embopress.org/page/journal/17574684/authorguide#dataavailability>).

13) Author contributions: You will be asked to provide CRediT (Contributor Role Taxonomy) terms in the submission system. These replace a narrative author contribution section in the manuscript.

14) A Conflict of Interest statement should be provided in the main text.

16) Include a Reagents and Tools Table as part of the Methods section, which can be downloaded from our author guidelines (<https://www.embopress.org/page/journal/17574684/authorguide#structuredmethods>)

***** Reviewer's comments *****

Referee #1 (Comments on Novelty/Model System for Author):

While the ip LPS model for preterm birth has limitations, it is widely accepted, as the authors discuss.

Referee #1 (Remarks for Author):

This manuscript is well written and has several strengths, namely that the novel compound discussed, HSJ633, improves neonatal outcomes along with lengthening gestation and that it also has a selective effect on downstream pathways, presumably making it more likely to be safe for both the mothers and neonates. However, I have identified several issues that should be addressed before the manuscript is considered for publication. I have divided my concerns into major and minor points.

Major Issues

The histologic changes in the lungs and intestines are convincing and a very interesting part of the story. However, the data in Supplemental Figure 3 indicate that inflammatory cytokines are increased in the LPS control animals and then reduced with 633 treatment, especially IL-6 levels in the intestines. Can you explain why there are no changes in inflammatory cell infiltrates seen in any of the tissue sections?

I do not think one can conclude that 633 penetrates the fetal placental compartment in LPS-treated mice based on Supplementary Figure 4. The figure shows that both FITC and FITC-HSJ633 penetrate the fetal placental compartment of LPS-treated mice. But would free HSJ633, the peptide unconjugated with the more polar FITC moiety, penetrate the placenta? If this work is translated to humans, would HSJ633 be modified with a non-polar moiety to enhance its ability to cross the placenta? It's true that 633 produced benefits in the fetuses, but these benefits could have resulted from 633's ability to reduce cytokines in the maternal compartment, thereby indirectly affecting the placenta and fetus. Please comment on this point and consider adding this point to the discussion section.

While administration of 633 prophylactically (Figure 1F) or within 2 h of LPS administration (Figure 6) allows most treated dams to deliver at term, administration of 633 in as little as 6 h after LPS administration has no statistically significant effect on the length of gestation (Figure 6). In the discussion, it is stated that the 2-h period after LPS administration is equivalent to a 30-h period after labor induction in humans. But LPS administration is not the same as "labor induction." Based on Figure 1E, delivery occurs approximately 24 h after LPS is first administered in the Vehicle group. So, 2 h after the first LPS administration is well before labor has been induced. Given this finding, how do you conceive of 633 as a treatment to delay preterm birth? Would it be administered prophylactically, based on specific diagnostic criteria predicting PTB? Or is the main purpose of 633 to reduce the effects of FIRS? Would 633 be given along with other drugs meant to extend gestation?

Figure 6E reveals that 633 administered 6 h after LPS administration causes loss of alveolar density in the lung, although this finding is not mentioned anywhere in the manuscript. What accounts for the 633-induced lung injury? Might there be risk of such lung injury occurring in some patients even if 633 were given promptly after the start of the inflammatory process? Given that 633 has a favorable effect on the lung at 2 h but a harmful effect at 6 h, again would 633 best be administered prophylactically?

While the biased effect of 633 on only the STAT3 pathway is encouraging, were any toxic effects observed in the dams?

The favorable effects of 633 on the lungs and intestines are shown, but were any other organs affected? Were there any unfavorable effects seen in any of the organs?

Can you confirm that 633 had no effect on litter size and produced no instances of teratogenic effects?

Minor Issues

In the provided copy of the manuscript Figure 6 is a bit "messed up." There is no Panel B and parts of the figure overlap each other.

Lines 280-281 read: "Neonatal survival rate was also reduced by HSJ633 administered 2 h post LPS (Figure 6C)..." Should that be "Neonatal survival rate was also improved by HSJ633 administered 2 h post LPS (Figure 6C)..."?

The Figure 4 title should end with "prophylactically" rather than "in prophylactic."

In Line 505, when mentioning prominent groups in the field that use ip LPS to induce preterm birth, please add the following two citations:

Manuel CR, Latuga MS, Ashby CR and Reznik SE. Immune Tolerance Attenuates Gut Dysbiosis, Dysregulated Uterine Gene Expression and High-Fat Diet Potentiated Preterm Birth in Mice. *Am J Obstet Gynecol*, 220: 596.e1-596.e28 (2019).

Sundaram S, Ashby CR, Pekson R, Sampat V, Sitapara R, Mantell L, Chen C-H, Yen H, Abhichandani K, Munnangi S, Khadtare N, Stephani RA and Reznik SE. N,N-Dimethylacetamide Regulates the Pro-inflammatory Response Associated with Endotoxin and Prevents Preterm Birth. *Am J Path* 183: 422-430 (2013).

Referee #2 (Comments on Novelty/Model System for Author):

The manuscript by Coté et al is a very interesting study describing the development of a novel way to inhibit IL-6 signaling. The designed inhibitory peptide(s) appears to allosterically modulate IL-6 signaling by mainly inhibiting STAT3 phosphorylation without prevention of IL-6-induced receptor complex formation. Using animal models, the authors show that this peptide(s) significantly prevent LPS-induced preterm birth. This was supported by data showing that LPS-induced preterm birth was also prevented by global inhibition of IL-6 signaling using IL-6R antibodies, by selective inhibition of IL-6 trans-signaling using sgp130 and by selective blocking of STAT3 by NSC74859.

However, the manuscript lacks clarity. I do not agree with the overall arrangement of the experiments and I think the manuscript lacks attention to detail.

Moreover, I would suggest to restructure the manuscript and to start with the in vitro data first and keep going with the in vivo data.

I have to admit that I am not familiar with the in vivo models, therefore most comments concentration on the design and in vitro testing of the peptides.

In general, I do not understand why the in vivo results with the STAT3 and the trans-signaling inhibitors are presented as supplemental data.

Major points:

The design of the peptides was basically described in the first section but I have not understand the rational and the overall design strategy - there is also no section in Mat and Met. Figure 1A and B "describe" the procedure and the principle. However, Figure 1A only shows a very general illustration of the IL-6R structure, including arrows pointing to some loop structures. Figure 1B should illustrate the principle of allosteric versus competitive inhibition of IL-6 signaling, but the illustration it is completely wrong. For instance, IL-6 is not a dimer. IL-6 interacts via site I with IL-6R but also with site II and site III with gp130. The illustration did not show the site II and site III interaction but instead shows some interaction between IL-6R and gp130 which is not present. Overall, peptide design has to be described in much greater detail and the illustration should be correct and help the reader to understand complex formation and the inhibitory principles.

HSJ631 seems to accelerate TNF secretion (Figure 1C), HSJ631 might be an allosteric (?) activator? This should at least be commented. But more data should be included to validate this finding - compare to Figure 2A or supplemental Figure 1 for example.

Characterization of the biological activity of HSJ631 in Supplemental Figure 1C is not convincing. IL-6/sIL-6R stimulation should be included. pSTAT3 appears to be too small (around 75 kDa) or too large (around 150 kDa). Different concentrations must be used of cytokines and inhibitors. Direct binding of HSJ631 to murine and human IL-6R must be shown. Binding kinetics with purified proteins and peptides must be performed, e.g. by surface plasmon resonance (SPR) or other methods which also give binding constants. Binding experiments on cells in Figure 2B-D is not convincing. More IL-6 concentrations must be used in Figure 2D.

The binding epitopes of the peptides must be characterized in detail.

The presented blots in Figure 7 are cropped and assembled. To directly compare activation of p38, ERK1/2, AKT and STAT3 all samples have to be analyzed on a single gel with appropriate controls. Moreover, 1 µg/ml IL-6 was used. Typically, concentrations between 1 and 10 ng/ml are sufficient.

What about the other STAT proteins and the inhibitory capacity of HSF633.

The authors state: "STAT3 activation is also detected in the placenta and is inhibited by prophylactic administration of HSJ633 and by post-LPS HSJ633 treatment (Figure 7E, F)." However, HSF633 appears to not interfere with STAT3 phosphorylation in this experiment. The reduction in STAT3 phosphorylation is not significant. Moreover, in Figure 7E and F also p38, ERK1/2, AKT should be analyzed to strengthen the point of the authors that HSJ631 selectively blocks STAT3 but not the other signaling pathways.

Supplemental Figure 6A - it appears to me that HSJ633 did not block IL-6 trans-signaling. Maybe there is a partial block but from the experiment presented using only one concentration of IL-6, sIL-6R and HSJ633, I would say that this compound did not affect IL-6 trans-signaling? This must be further investigated using HEK and Ba/F3 cells and additional concentrations. Why is this important? In Figure 6B, the authors data suggest that IL-6 trans-signaling mainly is responsible for preterm birth because the IL-6 trans-signaling inhibitor prevents preterm birth. From the data presented it is, however, not clear if sgp130 is as effective as HSJ633. Therefore, if HSJ633 did not interfere with IL-6 trans-signaling but IL-6 trans-signaling drives preterm birth, then why is HSJ633 effective?

Minor points:

What are "Ba/F3-polylinker cells"?

STAT3 is not a kinase. If referring to kinases in the signaling pathway JAKs should be mentioned - "Stimulation of HEK-Blue IL-6 cells with IL-6 elicited phosphorylation of p38, ERK1/2, AKT, and STAT3 (Figure 7A-D). Phosphorylation of all these kinases was blocked by the pre-incubation of cells with the IL-6R inhibitor Tocilizumab used as a positive control, confirming that the observed kinase activation was IL-6R-dependent (Figure 7A-D)."

Change to order of (Figure 7A-D, G) and (Figure 7E, F) - G should be E, E should be F and F should be G.

Suppl Figure 6A was mentioned and partly described two times in the text. This should be merged.

First and foremost, we would like to extend our sincere gratitude to the reviewers for their constructive feedback. Their insights have undoubtedly added considerable value to our paper. We have addressed each question and made changes where necessary. However, we did not accommodate all of the reviewers' comments, as we felt that some did not align with the objective of this paper; aspects left aside apply to molecular pharmacology of HSJ633 intended for future investigations.

Referee #1 (Comments on Novelty/Model System for Author):

While the ip LPS model for preterm birth has limitations, it is widely accepted, as the authors discuss.

Referee #1 (Remarks for Author):

This manuscript is well written and has several strengths, namely that the novel compound discussed, HSJ633, improves neonatal outcomes along with lengthening gestation and that it also has a selective effect on downstream pathways, presumably making it more likely to be safe for both the mothers and neonates. However, I have identified several issues that should be addressed before the manuscript is considered for publication. I have divided my concerns into major and minor points.

Major Issues

1. The histologic changes in the lungs and intestines are convincing and a very interesting part of the story. However, the data in Supplemental Figure 3 indicate that inflammatory cytokines are increased in the LPS control animals and then reduced with 633 treatment, especially IL-6 levels in the intestines. Can you explain why there are no changes in inflammatory cell infiltrates seen in any of the tissue sections?

H&E staining of histological sections do not reveal infiltration of inflammatory cells but rather tissue morphology. The purpose of showing this histology is to focus on the damaged morphology of major tissues secondary to *in utero* inflammation which occurs in reproductive tissues within hours after LPS exposure and does not last beyond GD17 (Figure EV4A).

2. I do not think one can conclude that 633 penetrates the fetal placental compartment in LPS-treated mice based on Supplementary Figure 4. The figure shows that both FITC and FITC-HSJ633 penetrate the fetal placental compartment of LPS-treated mice. But would free HSJ633, the peptide unconjugated with the more polar FITC moiety, penetrate the placenta? If this work is translated to humans, would HSJ633 be modified with a non-polar moiety to enhance its ability to cross the placenta? It's true that 633 produced benefits in the fetuses, but these benefits could have resulted from 633's ability to reduce cytokines in the maternal compartment, thereby indirectly affecting the placenta and fetus. Please comment on this point and consider adding this point to the discussion section.

We thank the reviewer for his/her insightful comments. We did not include unconjugated HSJ633 since it is not fluorescent and hence the approach utilized cannot be applied herein with free HSJ633. Although polar compounds may be more limited in crossing the placenta, many do via membrane transporters (Feghali *et al*, 2015). To detect free HSJ633 in the placenta (and fetal compartment), one could use mass spectrometry; this approach would complement FITC-conjugated HSJ633 usage in transplacental localization and will of course be the approach of choice in clinical study to determine pharmacokinetic profile. We have added a section discussing this point in the discussion; please refer to page 17, lines 404 to 416.

3. While administration of 633 prophylactically (Figure 1F) or within 2 h of LPS administration (Figure 6) allows most treated dams to deliver at term, administration of 633 in as little as 6 h after LPS administration has no statistically significant effect on the length of gestation (Figure 6). In the discussion, it is stated that the 2-h period after LPS administration is equivalent to a 30-h period after labor induction in humans. But LPS administration is not the same as "labor induction." Based on Figure 1E, delivery occurs approximately 24 h after LPS is first administered in the Vehicle group. So, 2 h after the first LPS administration is well before labor has been induced. Given this finding, how do you conceive of 633 as a treatment to delay preterm birth? Would it be administered prophylactically, based on specific diagnostic criteria predicting PTB? Or is the main purpose of 633 to reduce the effects of FIRS? Would 633 be given along with other drugs meant to extend gestation?

As pointed out by the reviewer, the inflammatory cascade triggers the process of labor (Habelrih *et al*, 2024) which is why we previously specified "30 hours after the onset of labor." For purposes of clarity, we revised the sentence to "30 hours after the induction of an inflammatory stimulus." Figure EV4 shows the expression of several pro-inflammatory genes in gestational tissues by 1 hour after LPS injection, indicative of rapid onset inflammation.

One could foresee prophylactic treatment of HSJ633 to women at risk such as those with a history of preterm birth; this was used for the progesterone trials, but its efficacy turned out to be negligible (Blackwell *et al*, 2020). HSJ633 could also be considered for a treatment approach with women in labor or with early ruptured membranes which imminently leads to preterm delivery, usually within 1-2 weeks. In these circumstances, antibiotics are used to avoid dissemination of sepsis, although the risk of preterm birth is unaffected by anti-microbials (Bowes, 2009). In women, HSJ633 would thus be administered without altering standards of care, including the use of ineffective tocolytics (Coler *et al*, 2021). These comments are now added to the Discussion (last para).

4. Figure 6E reveals that 633 administered 6 h after LPS administration causes loss of alveolar density in the lung, although this finding is not mentioned anywhere in the manuscript. What accounts for the 633-induced lung injury? Might there be risk of such lung injury occurring in some patients even if 633 were given promptly after the start of the inflammatory process? Given that 633 has a favorable effect on the lung at 2 h but a harmful effect at 6 h, again would 633 best be administered prophylactically?

We thank the reviewer for pointing this out. Lung histology had been performed on two additional litters (yielding total n=4) which were inadvertently forgotten. The complete data set indicates statistically that HSJ633 administered 6 h post-LPS is neither significantly harmful nor effective in preserving lung alveoli (Fig. 6E). Prophylactic administration or early post-LPS treatment with HSJ633 provides best efficacy (Fig. 4D).

5. While the biased effect of 633 on only the STAT3 pathway is encouraging, were any toxic effects observed in the dams?

We did not detect adverse effects in the dams treated with HSJ633 or with the STAT3 inhibitor NSC74859.

6. The favorable effects of 633 on the lungs and intestines are shown, but were any other organs affected? Were there any unfavorable effects seen in any of the organs?

We have not observed adverse effects to lung or intestinal histology secondary to HSJ633 exposure. Moreover, neonatal survival and body mass was improved by HSJ633 as shown at birth and 1 week later at PT7, suggestive of general well-being. We also examined the pups' vulnerable brains at PT7,

focusing on the hippocampus and cortex. Importantly, we found that HSJ633 administered prophylactically preserved vascular density in brain; these data were not included in the paper as post-LPS treatment with HSJ633 was not performed. Overall, the present work reveals that major organs of progeny are protected by HSJ633. Full-fledged pre-clinical toxicology studies are planned with a GLP-certified Clinical Research Organization as we later progress towards clinical trial.

7. Can you confirm that 633 had no effect on litter size and produced no instances of teratogenic effects?

As shown in Figures 4B and 6C, we counted the number of living pups at PT1. In both HSJ633 groups (administered either before or after LPS injection), the number of living pups was comparable to that of the sham group. Teratogenicity is unlikely during the fetal stage of development (GD11-GD19.5) when HSJ633 is administered (GD16); no treatments took place during the embryonic period (GD0-10.5). Nonetheless, we proceeded to perform pilot data on toxicity of HSJ633 (administered at GD16) on lungs and gut; to avoid conflicting results with inflammation, these experiments were conducted on sham animals treated or not with HSJ633. Histoanatomical studies reveal no effect of HSJ633 compared to vehicle at PT1 (see results below).

Minor Issues

1. In the provided copy of the manuscript Figure 6 is a bit "messed up." There is no Panel B and parts of the figure overlap each other.

We have made the necessary corrections to Fig. 6.

2. Lines 280-281 read: "Neonatal survival rate was also reduced by HSJ633 administered 2 h post LPS (Figure 6C)..." Should that be "Neonatal survival rate was also improved by HSJ633 administered 2 h post LPS (Figure 6C)..."?

Thank you for pointing out this error in the phrase. Correction was made.

3. The Figure 4 title should end with "prophylactically" rather than "in prophylactic."

Correction was made.

4. In Line 505, when mentioning prominent groups in the field that use ip LPS to induce preterm birth, please add the following two citations: Manuel CR, Latuga MS, Ashby CR and Reznik SE. Immune Tolerance Attenuates Gut Dysbiosis, Dysregulated Uterine Gene Expression and High-Fat Diet Potentiated Preterm Birth in Mice. Am J Obstet Gynecol, 220: 596.e1-596.e28 (2019).

Sundaram S, Ashby CR, Pekson R, Sampat V, Sitapara R, Mantell L, Chen C-H, Yen H, Abhichandani K, Munnangi S, Khadtare N, Stephani RA and Reznik SE. N,N-Dimethylacetamide Regulates the Pro-inflammatory Response Associated with Endotoxin and Prevents Preterm Birth. Am J Path 183: 422-430 (2013).

Suggested references were added (page 23 line 537).

Referee #2 (Comments on Novelty/Model System for Author):

The manuscript by Coté et al is a very interesting study describing the development of a novel way to inhibit IL-6 signaling. The designed inhibitory peptide(s) appears to allosterically modulate IL-6 signaling by mainly inhibiting STAT3 phosphorylation without prevention of IL-6-induced receptor complex formation. Using animal models, the authors show that this peptide(s) significantly prevent LPS-induced preterm birth. This was supported by data showing that LPS-induced preterm birth was also prevented by global inhibition of IL-6 signaling using IL-6R antibodies, by selective inhibition of IL-6 trans-signaling using sgp130 and by selective blocking of STAT3 by NSC74859.

However, the manuscript lacks clarity. I do not agree with the overall arrangement of the experiments and I think the manuscript lacks attention to detail.

Moreover, I would suggest to restructure the manuscript and to start with the in vitro data first and keep going with the in vivo data.

I have to admit that I am not familiar with the in vivo models, therefore most comments concentration on the design and in vitro testing of the peptides.

In general, I do not understand why the in vivo results with the STAT3 and the trans-signaling inhibitors are presented as supplemental data.

We thank the reviewer for his/her comments. The article was organized in the way it is presented since the focus we wanted to convey applies to generating innovative solutions to tackle preterm birth; hence emphasis was placed on the latter starting with Fig 1. In this process, we first screened to identify potential IL-6R antagonist peptides designed on the basis of a long-standing effective approach by reproducing the amino acid sequence of putative mobile loop regions of IL-6R, as successfully achieved for numerous other receptors, such as FP, V2R, IL-23R, IL-1R (Goupil *et al*, 2010; Quiniou *et al*, 2014;

Quiniou *et al*, 2008; Rihakova *et al*, 2009). In line with the reviewer, we then proceeded to characterize selectivity/specificity properties of our lead peptide, HSJ633. This was then followed by *in vivo* characterization of HSJ633 in an established model of preterm birth, by testing in a coherent manner the efficacy of the drug candidate in preventive and treatment modes.

Major points: ALL COMMENTS APPLY TO PURE PHARMACOLOGY IRRESPECTIVE OF IN VIVO DATA

1. The design of the peptides was basically described in the first section but I have not understand the rational and the overall design strategy - there is also no section in Mat and Met. Figure 1A and B "describe" the procedure and the principle. However, Figure 1A only shows a very general illustration of the IL-6R structure, including arrows pointing to some loop structures. Figure 1B should illustrate the principle of allosteric versus competitive inhibition of IL-6 signaling, but the illustration it is completely wrong. For instance, IL-6 is not a dimer. IL-6 interacts via site I with IL-6R but also with site II and site III with gp130. The illustration did not show the site II and site III interaction but instead shows some interaction between IL-6R and gp130 which is not present. Overall, peptide design has to be described in much greater detail and the illustration should be correct and help the reader to understand complex formation and the inhibitory principles.

Fig. 1A & B provide a general graphic representation of IL-6R and its interaction with gp130 and its ligand IL-6. Antibodies to IL-6R interfere (eg. Tocilizumab) with ligand (IL-6) binding (right Fig. 1B panel) resulting with inhibition of all IL-6-coupled signals coherent with orthosteric inhibition; whereas HSJ peptides (middle Fig. 1B panel) bind to allosteric sites remote from the ligand binding region biasing signal transduction; concordant details are presented in the legend to Fig. 1. Notably, IL-6, IL-6R, and gp130 complex form a hexamer; we have revised the schema to best show that IL-6R and gp130 interact via the D3 domain of IL-6R α and gp130, which constitutes the site Iib (Boulanger *et al*, 2003). A section on peptide design was added to the Materials and Methods section.

2. HSJ631 seems to accelerate TNF secretion (Figure 1C), HSJ631 might be an allosteric (?) activator? This should at least be commented. But more data should be included to validate this finding - compare to Figure 2A or supplemental Figure 1 for example.

It does indeed appear that HSJ631 increases the expression of TNF- α . Since we were seeking to identify a negative allosteric modulator that would decrease inflammatory factor generation, we justifiably did not pursue work with HSJ631. In the context of inhibiting inflammation, as would apply for the prevention of preterm birth, HSJ633 and HSJ639 displayed more promising results. This aspect is addressed on page 8 and lines 183-185.

3. Characterization of the biological activity of HSJ631 in Supplemental Figure 1C is not convincing. IL-6/sIL-6R stimulation should be included. pSTAT3 appears to be too small (around 75 kDa) or too large (around 150 kDa). Different concentrations must be used of cytokines and inhibitors. Direct binding of HSJ631 to murine and human IL-6R must be shown. Binding kinetics with purified proteins and peptides must be performed, e.g. by surface plasmon resonance (SPR) or other methods which also give binding constants. Binding experiments on cells in Figure 2B-D is not convincing. More IL-6 concentrations must be used in Figure 2D.

We believe the reviewer is referring to HSJ633 rather than HSJ631. A newly added figure (EV5B) demonstrates that STAT3 activation in Ba/F3 (murine) cells stimulated with hyper-IL6 (IL-6 bound to sIL-

6R) is reduced by HSJ633. Note that the molecular weight of STAT3 monomer is reported to be 79-86 kDa and thus doubled as dimer (Cell Signaling, #9138), consistent with western blot shown in the corresponding figure.

We have provided several lines of evidence suggesting that HSJ633 interacts at high affinity with IL-6R; this includes using radio-labelled HSJ633 in cells with but not in cells without IL-6R, dose-dependent inhibition of inflammatory factors yielding IC₅₀ values in the low nanomolar range as per binding, and STAT3 activation in cells with but not in cells devoid of IL-6R (Fig. 2). Moreover, the concentration of IL-6 used to displace bound-HSJ633 was sufficient as it far exceeded (10,000-fold) the IC₅₀. Accordingly, binding kinetics using purified receptor would yield little additional information in the context of the pre-clinical pharmacologic focus of this study, and thus we feel this issue can be addressed in a future paper on the molecular characterization of HSJ633 binding to IL-6R. Nonetheless, we have refined our terminology to avoid confusion regarding direct binding of HSJ633 to IL-6R.

4. The binding epitopes of the peptides must be characterized in detail.

We appreciate this detail, but identification of binding regions of HSJ633 could entail a long experimental period and would best be part of a separate study on molecular characterization properties of HSJ633, as described above. Again, we feel that the primary goal of this paper consists in demonstrating the efficacy of a novel promising (small peptide) IL-6R modulator HSJ633 in reducing preterm birth and improving offspring outcomes; this aspect is ground-breaking and fulfills an unmet medical need.

5. The presented blots in Figure 7 are cropped and assembled. To directly compare activation of p38, ERK1/2, AKT and STAT3 all samples have to be analyzed on a single gel with appropriate controls. Moreover, 1 µg/ml IL-6 was used. Typically, concentrations between 1 and 10 ng/ml are sufficient.

We agree that a single gel may have been preferred. However, during the membrane preparation, the samples were not all placed next to each other to create a cohesive image for publication. Nonetheless, we do provide the editor our membrane figures that show these results. Importantly, despite using a high dosage of IL-6 (µg/ml) and of HSJ633 (1µM) ERK1/2, AKT, and p38 were unaffected, whereas STAT3 was inhibited.

6. What about the other STAT proteins and the inhibitory capacity of HSF633 (HSJ633).

We focused on STAT3 since it is canonically directly involved in IL-6 signaling, unlike other STAT proteins. Moreover, a STAT3 inhibitor NSC74859 successfully inhibited preterm birth, in line with its purported role.

7. The authors state: "STAT3 activation is also detected in the placenta and is inhibited by prophylactic administration of HSJ633 and by post-LPS HSJ633 treatment (Figure 7E, F)." However, HSF633 (HSJ633) appears to not interfere with STAT3 phosphorylation in this experiment. The reduction in STAT3 phosphorylation is not significant. Moreover, in Figure 7E and F also p38, ERK1/2, AKT should be analyzed to strengthen the point of the authors that HSJ631 (HSJ633) selectively blocks STAT3 but not the other signaling pathways.

There is a reduction in STAT3 activation in the placenta when HSJ633 is administered prophylactically; a tendency towards significant inhibition of STAT3 activation is also observed when HSJ633 is administered 2h post-LPS (Fig. 7F & G). It should be emphasized herein that reproduction of receptor-coupled signaling corroboration in *in vivo* studies is complicated when compared to a controlled environment in cultured cells or tissues, as specific timing of signal activation is difficult to control *in vivo*.

Nonetheless, overall the inhibition of STAT3 *in vivo* and *in vitro*, along with efficacy of STAT3 inhibition (using NSC74859) on preterm birth, provides solid evidence for involvement of STAT3 in this process.

8. Supplemental Figure 6A - it appears to me that HSJ633 did not block IL-6 trans-signaling. Maybe there is a partial block but from the experiment presented using only one concentration of IL-6, sIL-6R and HSJ633, I would say that this compound did not affect IL-6 trans-signaling? This must be further investigated using HEK and Ba/F3 cells and additional concentrations. Why is this important? In Figure 6B, the authors data suggest that IL-6 trans-signaling mainly is responsible for preterm birth because the IL-6 trans-signaling inhibitor prevents preterm birth. From the data presented it is, however, not clear if sgp130 is as effective as HSJ633. Therefore, if HSJ633 did not interfere with IL-6 trans-signaling but IL-6 trans-signaling drives preterm birth, then why is HSJ633 effective?

We thank the reviewer for an excellent point regarding soluble IL-6R (sIL-6R). We felt that an attempt to identify which receptor (membrane-bound vs soluble IL-6R) is involved in conveying preterm birth-associated inflammation is of pharmacologic interest. Specifically, the *in vitro* studies (Fig. EV5) enabled to reveal efficacy of HSJ633 on the soluble receptor, in addition to its effects on the membrane-bound IL-6R [Fig. 2]. The *in vivo* data reveal that capturing sIL-6R (using soluble gp130 [sgp130]) provides protection against preterm birth to the same extent as that observed with HSJ633. Hence, we have added Figure EV5B to demonstrate that Ba/F3 (relevant) murine cells stimulated with hyper-IL6 (IL-6 bound to sIL-6R) in the presence of HSJ633 reduces activation of STAT3. This suggests that HSJ633 effectively decreases STAT3 activation through soluble receptor-mediated trans-signaling. As shown in Figure EV5C, *in vivo* sgp130 results in a preterm birth rate of 20% comparable to the 22.9% for HSJ633. Altogether, HSJ633 reduces STAT3 activation via the soluble receptor, underscoring the relevance of our *in vivo* results with sgp130 in the preterm birth model.

Minor points:

1. What are "Ba/F3-polylinker cells"?

Ba/F3-polylinker cells are those transfected with the empty vector, serving as controls. Definition of polylinker has been clarified in line 599 page 25.

2. STAT3 is not a kinase. If referring to kinases in the signaling pathway JAKs should be mentioned - "Stimulation of HEK-Blue IL-6 cells with IL-6 elicited phosphorylation of p38, ERK1/2, AKT, and STAT3 (Figure 7A-D). Phosphorylation of all these kinases was blocked by the pre-incubation of cells with the IL-6R inhibitor Tocilizumab used as a positive control, confirming that the observed kinase activation was IL-6R-dependent (Figure 7A-D)."

Corresponding changes have been made.

3. Change to order of (Figure 7A-D, G) and (Figure 7E, F) - G should be E, E should be F and F should be G.

Changes in figure 7 have been made as suggested.

4. Suppl Figure 6A was mentioned and partly described two times in the text. This should be merged.

Correction was made to sentence.

Bibliography

- Blackwell SC, Gyamfi-Bannerman C, Biggio JR, Jr., Chauhan SP, Hughes BL, Louis JM, Manuck TA, Miller HS, Das AF, Saade GR *et al* (2020) 17-OHPC to Prevent Recurrent Preterm Birth in Singleton Gestations (PROLONG Study): A Multicenter, International, Randomized Double-Blind Trial. *Am J Perinatol* 37: 127-136
- Boulanger MJ, Chow DC, Brevnova EE, Garcia KC (2003) Hexameric structure and assembly of the interleukin-6/IL-6 alpha-receptor/gp130 complex. *Science* 300: 2101-2104
- Bowes WA (2009) The role of antibiotics in the prevention of preterm birth. *F1000 Med Rep* 1
- Coler BS, Shynlova O, Boros-Rausch A, Lye S, McCartney S, Leimert KB, Xu W, Chemtob S, Olson D, Li M *et al* (2021) Landscape of Preterm Birth Therapeutics and a Path Forward. *J Clin Med* 10
- Feghali M, Venkataramanan R, Caritis S (2015) Pharmacokinetics of drugs in pregnancy. *Semin Perinatol* 39: 512-519
- Goupil E, Tassy D, Bourguet C, Quiniou C, Wisheart V, Pétrin D, Le Gouill C, Devost D, Zingg HH, Bouvier M *et al* (2010) A novel biased allosteric compound inhibitor of parturition selectively impedes the prostaglandin F2alpha-mediated Rho/ROCK signaling pathway. *J Biol Chem* 285: 25624-25636
- Habelrih T, Augustin TL, Mauffette-Whyte F, Ferri B, Sawaya K, Côté F, Gallant M, Olson DM, Chemtob S (2024) Inflammatory mechanisms of preterm labor and emerging anti-inflammatory interventions. *Cytokine Growth Factor Rev* 78: 50-63
- Quiniou C, Domínguez-Punaro M, Cloutier F, Erfani A, Ennaciri J, Sivanesan D, Sanchez M, Chognard G, Hou X, Rivera JC *et al* (2014) Specific targeting of the IL-23 receptor, using a novel small peptide noncompetitive antagonist, decreases the inflammatory response. *Am J Physiol Regul Integr Comp Physiol* 307: R1216-1230
- Quiniou C, Sapieha P, Lahaie I, Hou X, Brault S, Beauchamp M, Leduc M, Rihakova L, Joyal JS, Nadeau S *et al* (2008) Development of a novel noncompetitive antagonist of IL-1 receptor. *J Immunol* 180: 6977-6987
- Rihakova L, Quiniou C, Hamdan FF, Kaul R, Brault S, Hou X, Lahaie I, Sapieha P, Hamel D, Shao Z *et al* (2009) VRQ397 (CRAVKY): a novel noncompetitive V2 receptor antagonist. *Am J Physiol Regul Integr Comp Physiol* 297: R1009-1018

12th Feb 2025

Dear Dr. Chemtob,

Thank you for the submission of your revised manuscript to EMBO Molecular Medicine. We have now heard back from the one referee who agreed to re-evaluate your manuscript. As you will see from the reports below, referee #2 remains critical regarding insufficient characterization of the small peptide inhibitor of IL-6R. Given that the referee #1 did not agree to re-evaluate the manuscript, I have sought external advice on the study from an expert in the field.

Our external advisor acknowledged interest of the study and overall conclusiveness of the data presented, however, in his/her opinion "since the discovery of the role of IL6R has been made before, the development of an allosteric inhibitor alone without further demonstration of specificity and efficacy in an adequate human model does not seem sufficient to be published in EMBO Molecular Medicine". He/She further suggests "the authors could use for example human placental explants (control and from preterm birth models) to confirm the results obtained in the selective inhibition of IL6R signaling and in the reduction of inflammatory processes".

From our side, we appreciate the work that has been done to address the referee criticisms, however, considering comments from our advisor confirmation of the results in a more appropriate human model as suggested by the advisor in an additional and final round of revision would be required for further consideration of the manuscript at EMBO Molecular Medicine. Concerns raised by the referee #2 should be addressed textually, no further biochemical characterization of the small peptide is required. Regarding referee #2 point 1, please provide more information about the peptide design and correct the illustration in Figure 1.

Please also amend the following:

- Correct order of manuscript sections: Abstract, The Paper Explained, Introduction, Results, Discussion, Methods, Acknowledgements, Disclosure and competing interests statement, References, Figure legends, Tables and their legends, Expanded View Figure legends.
- Please address all comments suggested by our data editors listed below:
 - o Figure legends:
 1. Please note that the exact p values are not provided in the legends of figures 1C, E, F, G; 2A, B, D; 3A-D; 4A-E; 5A, B; 6B-F; 7A-H; EV1 D, EV2 A, B; EV3, EV4 B, C.
 2. Please indicate the statistical test used for data analysis in the legends of figures 6E, 7E.
 3. Please note that in figures EV2 A, B; EV3 there is a mismatch between the annotated p values in the figure legend and the annotated p values in the figure file that should be corrected.
 4. Please indicate what */ **/ ***/ **** represents; if this represents p value(s), please specify the exact p value in the legend(s) of figure(s) EV4 A-C.
 5. Please note that n=2 in figures 1C, 5A, B; 7A-D, H.
 - In Western blot images each splice site needs to be marked with a strong white or black line indicating the splice.
 - Please resize the synopsis image to 550 pixels wide x 200-600 pixels high and increase the font size for better readability.
 - The legend for Table EV1 should be removed from the manuscript text and added to the table, at the top of the page.
 - Please make Appendix Figure S1 to an EV Figure.
 - Author contributions: Please remove it from the manuscript and specify author contributions in our submission system. CRediT has replaced the traditional author contributions section because it offers a systematic machine-readable author contributions format that allows for more effective research assessment. You are encouraged to use the free text boxes beneath each contributing author's name to add specific details on the author's contribution. More information is available in our guide to authors:
<https://www.embopress.org/page/journal/17574684/authorguide#authorshipguidelines>
 - In Methods, please add following paragraph:
Graphics:
(some of the... OR Figure #... OR synopsis) Graphics were created with BioRender.com."

Further consideration of a revision that addresses reviewer and advisor concerns will entail an additional round of review. Acceptance or rejection of the manuscript will depend on the completeness of your responses included in the next, final version of the manuscript. For this reason, and to save you from any frustrations in the end, I would strongly advise against returning an incomplete revision.

We would welcome the submission of a revised version within three months for further consideration. Please let us know if you require longer to complete the revision.

I look forward to receiving your revised manuscript.

Yours sincerely,

Zeljko Durdevic

We require:

- 1) A .docx formatted version of the manuscript text (including legends for main figures, EV figures and tables). Please make sure that the changes are highlighted to be clearly visible.
 - 2) Individual production quality figure files as .eps, .tif, .jpg (one file per figure). For guidance, download the 'Figure Guide PDF': (<https://www.embopress.org/page/journal/17574684/authorguide#figureformat>).
 - 3) A .docx formatted letter INCLUDING the reviewers' reports and your detailed point-by-point responses to their comments. As part of the EMBO Press transparent editorial process, the point-by-point response is part of the Review Process File (RPF), which will be published alongside your paper.
 - 4) A complete author checklist, which you can download from our author guidelines (<https://www.embopress.org/page/journal/17574684/authorguide#submissionofrevisions>). Please insert information in the checklist that is also reflected in the manuscript. The completed author checklist will also be part of the RPF.
 - 5) Please note that all corresponding authors are required to supply an ORCID ID for their name upon submission of a revised manuscript.
 - 6) It is mandatory to include a 'Data Availability' section after the Materials and Methods. Before submitting your revision, primary datasets produced in this study need to be deposited in an appropriate public database, and the accession numbers and database listed under 'Data Availability'. Please remember to provide a reviewer password if the datasets are not yet public (see <https://www.embopress.org/page/journal/17574684/authorguide#dataavailability>).
- In case you have no data that requires deposition in a public database, please state so in this section. Note that the Data Availability Section is restricted to new primary data that are part of this study.
- 7) For data quantification: please specify the name of the statistical test used to generate error bars and P values, the number (n) of independent experiments (specify technical or biological replicates) underlying each data point and the test used to calculate p-values in each figure legend. The figure legends should contain a basic description of n, P and the test applied. Graphs must include a description of the bars and the error bars (s.d., s.e.m.). See also 'Figure Legend' guidelines: <https://www.embopress.org/page/journal/17574684/authorguide#figureformat>
 - 8) At EMBO Press we ask authors to provide source data for the main manuscript figures. Our source data coordinator will contact you to discuss which figure panels we would need source data for and will also provide you with helpful tips on how to upload and organize the files.
 - 9) Our journal encourages inclusion of *data citations in the reference list* to directly cite datasets that were re-used and obtained from public databases. Data citations in the article text are distinct from normal bibliographical citations and should directly link to the database records from which the data can be accessed. In the main text, data citations are formatted as follows: "Data ref: Smith et al, 2001" or "Data ref: NCBI Sequence Read Archive PRJNA342805, 2017". In the Reference list, data citations must be labeled with "[DATASET]". A data reference must provide the database name, accession number/identifiers and a resolvable link to the landing page from which the data can be accessed at the end of the reference.

Further instructions are available at .

.

12) Author contributions: You will be asked to provide CRediT (Contributor Role Taxonomy) terms in the submission system. These replace a narrative author contribution section in the manuscript.

13) A Conflict of Interest statement should be provided in the main text.

14) Every published paper now includes a 'Synopsis' to further enhance discoverability. Synopses are displayed on the journal webpage and are freely accessible to all readers. They include a short stand first (maximum of 300 characters, including space) as well as 2-5 one-sentences bullet points that summarizes the paper. Please write the bullet points to summarize the key NEW findings. They should be designed to be complementary to the abstract - i.e. not repeat the same text. We encourage inclusion of key acronyms and quantitative information (maximum of 30 words / bullet point). Please use the passive voice. Please attach these in a separate file or send them by email, we will incorporate them accordingly.

15) Include a Reagents and Tools Table as part of the Methods section, which can be downloaded from our author guidelines (<https://www.embopress.org/page/journal/17574684/authorguide#structuredmethods>)

**** Reviewer's comments ****

Referee #2 (Comments on Novelty/Model System for Author):

Interesting model by the quality of the compound design process and principle in vitro testing is not convincing.

Referee #2 (Remarks for Author):

I am still not convinced by the additional data provided by the authors.

Firstly, some important concerns have not been addressed. Figure 1 is still incorrect and does not add any helpful information about the mode of action of the novel compounds.

At first I said: "1. The design of the peptides was basically described in the first section but I have not understand the rational and the overall design strategy - there is also no section in Mat and Met. Figure 1A and B "describe" the procedure and the principle. However, Figure 1A only shows a very general illustration of the IL-6R structure, including arrows pointing to some loop structures. Figure 1B should illustrate the principle of allosteric versus competitive inhibition of IL-6 signaling, but the illustration it is completely wrong. For instance, IL-6 is not a dimer. IL-6 interacts via site I with IL-6R but also with site II and site III with gp130. The illustration did not show the site II and site III interaction but instead shows some interaction between IL-6R and gp130 which is not present. Overall, peptide design has to be described in much greater detail and the illustration should be correct and help the reader to understand complex formation and the inhibitory principles. "

All my comments are still valid and have not been addressed by the authors. I still believe that a novel compound needs to be characterised in great detail during the testing and publication process.

My original point 3 was not properly addressed:" Characterization of the biological activity of HSJ631 in Supplemental Figure 1C is not convincing. IL-6/sIL-6R stimulation should be included. pSTAT3 appears to be too small (around 75 kDa) or too large (around 150 kDa). Different concentrations must be used of cytokines and inhibitors. Direct binding of HSJ631 to murine and human IL-6R must be shown. Binding kinetics with purified proteins and peptides must be performed, e.g. by surface plasmon resonance (SPR) or other methods which also give binding constants. Binding experiments on cells in Figure 2B-D is not convincing. More IL-6 concentrations must be used in Figure 2D. "

The new figure (EV5B) should show that STAT3 activation in Ba/F3 (mouse) cells stimulated with hyper-IL6 (IL-6 bound to sIL-6R) is reduced by HSJ633. However, the blot presented is far from convincing. Hyper-IL-6 is typically a very good activator of pSTAT3 in Ba/F3 cells and this signal is too low to be taken seriously.

My point 4 was also not addressed, albeit the authors suggested binding of their compounds in Figure 1A: "4. The binding epitopes of the peptides must be characterized in detail. "

My point 5 has not been addressed, I think the samples should be analysed properly and even lower concentrations of IL-6 (less than 1µg/ml) should be tested: "5. The presented blots in Figure 7 are cropped and assembled. To directly compare activation of p38, ERK1/2, AKT and STAT3 all samples have to be analyzed on a single gel with appropriate controls. Moreover, 1 µg/ml IL-6 was used. Typically, concentrations between 1 and 10 ng/ml are sufficient."

****** Editor's comments ******

1. From our side, we appreciate the work that has been done to address the referee criticisms, however, considering comments from our advisor confirmation of the results in a more appropriate human model as suggested by the advisor in an additional and final round of revision would be required for further consideration of the manuscript at EMBO Molecular Medicine.

We thank the Editors' comments. We agree that the most important query raised applies to application of our results to an appropriate human model. This of course cannot include studies in intact humans. However, given the relevance of fetal membranes as important participants in preterm birth (PTB) (Truong *et al*, 2023), we have performed new experiments on human fetal membrane explants, which show that HSJ633 reduces inflammation induced by yet another danger-associated molecular pattern (DAMP) stimulant, specifically HMGB1; results are presented in the new Figure 3, and text is expanded on p11 last para and on p 18 para 1; methodology including rationale for tissue selection is detailed on p 27 para 1 & 2. Data consolidate the relevance of our findings.

2. Concerns raised by the referee #2 should be addressed textually, no further biochemical characterization of the small peptide is required. Regarding referee #2 point 1, please provide more information about the peptide design and correct the illustration in Figure 1.

We thank the Editor to specify in the present context of the paper, that only textual clarification is needed to address query about the small peptide, HSJ633. We have added more details regarding the peptide design and have revised Figure 1 accordingly; identification of region on IL6R harboring the amino acid sequence of HSJ633 is presented using AlphaFold structural model.

****** Reviewer's comments ********Referee #2 (Comments on Novelty/Model System for Author):**

Interesting model by the quality of the compound design process and principle in vitro testing is not convincing.

Referee #2 (Remarks for Author):

I am still not convinced by the additional data provided by the authors. Firstly, some important concerns have not been addressed. Figure 1 is still incorrect and does not add any helpful information about the mode of action of the novel compounds.

General address on comments by Reviewer 2:

Overall, we respect the interests of Reviewer 2 in characterizing the interaction of HSJ633 with IL-6R. Yet, we present sufficient coherent complementary substantiation of HSJ633 effects on cells and *in vivo* in a relevant physio-pharmacological condition, revealing efficacy of the newly described IL-6R modulator, HSJ633. In the context of the message conveyed, we feel that the characterization of HSJ633 provided is adequate and delivers the unmet need of a potential therapeutic candidate targeting for the first time IL-6R not only for preterm birth but importantly displaying significant benefits in neonatal outcome. Additional characterization as requested by reviewer 2 would apply to a fully separate article focused on this purpose, and thus is beyond the scope of the current paper.

1. At first I said: "1. The design of the peptides was basically described in the first section but I have not understand the rational and the overall design strategy - there is also no section in Mat and Met. Figure 1A and B "describe" the procedure and the principle. However, Figure 1A only shows a very general illustration of the IL-6R structure, including arrows pointing to some loop structures. Figure 1B should illustrate the principle of allosteric versus competitive inhibition of IL-6 signaling, but the illustration it is completely wrong. For instance, IL-6 is not a dimer. IL-6 interacts via site I with IL-6R but also with site II and site III with gp130. The illustration did not show the site II and site III interaction but instead shows some interaction between IL-6R and gp130 which is not present. Overall, peptide design has to be described in much greater detail and the illustration should be correct and help the reader to understand complex formation and the inhibitory principles. "

All my comments are still valid and have not been addressed by the authors. I still believe that a novel compound needs to be characterised in great detail during the testing and publication process.

The rationale for strategy used to develop allosteric modulators of IL-6R has been described in the Materials & Methods section, consistent with the design used for a number of other receptors, notably such as FP, IL-1R, V2R, IL-23R (Goupil *et al*, 2010; Quiniou *et al*, 2014; Quiniou *et al*, 2008; Rihakova *et al*, 2009). The essence of Fig 1A,B is simply to illustrate the general distinction between orthosteric and allosteric differences in modulating receptor-coupled signaling; it is not intended to reveal specific alterations in receptor complex interactions by HSJ633, as the latter represents a paper in itself, apart from the extended *in vivo* biology focused in the present version. Nonetheless, we have revised Fig 1A and 1B to include additional details in line with what is being requested, and we have expanded the Materials and Methods section to provide more information regarding the peptide design. Using AlphaFold, we have specifically indicated the region

from which the amino acid sequence of HSJ633 is derived (Fig. 1A) (Abramson *et al*, 2024). In essence, IL-6, IL-6R (referred to as IL6-RA), and gp130 complex (referred to as IL6-RB) form a hexamer; we have revised the schema to best show interactions of IL-6R with gp130 through what is referred to as site IIb by Boulanger *et al*, 2003. Antibodies to IL-6R interfere (eg. Tocilizumab) with ligand (IL-6) binding (Fig. 1B [right panel]) resulting with inhibition of all IL-6-coupled signals coherent with orthosteric inhibition; whereas HSJ peptides (Fig. 1B [middle panel]) bind to allosteric sites remote from the ligand binding region biasing signal transduction; concordant details are presented in the legend to Fig. 1. Once again, specific interaction between HSJ633 and IL-6R are beyond the scope of the present manuscript.

Please note that for reasons based on intellectual property (and patenting) we avoided to show specific regions from which other peptides stated in Fig. 1C are derived. This however has negligible impact on the present paper focused on HSJ633.

2. My original point 3 was not properly addressed:" Characterization of the biological activity of HSJ631 in Supplemental Figure 1C is not convincing. IL-6/sIL-6R stimulation should be included. pSTAT3 appears to be too small (around 75 kDa) or too large (around 150 kDa). Different concentrations must be used of cytokines and inhibitors. Direct binding of HSJ631 to murine and human IL-6R must be shown. Binding kinetics with purified proteins and peptides must be performed, e.g. by surface plasmon resonance (SPR) or other methods which also give binding constants. Binding experiments on cells in Figure 2B-D is not convincing. More IL-6 concentrations must be used in Figure 2D. "

As previously mentioned, while we fully agree that experiments suggested could provide additional mechanistic insights, the primary focus of the current manuscript is to investigate the therapeutic potential of HSJ633 in the prevention of preterm birth, rather than to provide an in-depth molecular characterization of HSJ633.

The reviewer reiterates the same query made previously for which we the authors feel like we have already addressed adequately, including by performing new experiments now shown in Fig. EV1. First we believe the reviewer is referring to HSJ633 rather than HSJ631. Secondly, in Fig. EV1 we show that activation of STAT3 in Ba/F3 (murine) cells stimulated with hyper-IL6 (IL-6 bound to sIL-6R) is reduced by HSJ633; (the molecular weight of STAT3 monomer is reported to be 79-86 kDa and thus doubled as dimer [Cell Signaling, #9138], consistent with western blot shown in the corresponding figure).

Regarding interaction of HSJ633 with IL-6R, we have provided several lines of evidence for such interaction at high affinity; this includes (i) using radio-labelled HSJ633 in cells with but not in cells without IL-6R, (ii) dose-dependent inhibition of inflammatory factors yielding IC₅₀ values in the low nanomolar range as per binding, and (iii) STAT3 activation in cells with but not in cells devoid of IL-6R (Fig. 2). Importantly, the concentration of IL-6 used to displace bound-HSJ633 was amply sufficient (10⁻⁴ M) as it far exceeded (10,000-fold) the IC₅₀. Consistent with focus of this paper substantiated

in our comments above, we feel that binding kinetics using purified receptor would yield little additional information in the context of the pre-clinical pharmacologic focus of this study. Hence although we acknowledge the importance of understanding the binding kinetics and precise mechanism of interaction of HSJ633, we strongly believe that this issue can be addressed in a future paper on the molecular characterization of HSJ633 binding to IL-6R.

3. The new figure (EV5B) should show that STAT3 activation in Ba/F3 (mouse) cells stimulated with hyper-IL6 (IL-6 bound to sIL-6R) is reduced by HSJ633. However, the blot presented is far from convincing. Hyper-IL-6 is typically a very good activator of pSTAT3 in Ba/F3 cells and this signal is too low to be taken seriously. My point 4 was also not addressed, albeit the authors suggested binding of their compounds in Figure 1A: "4. The binding epitopes of the peptides must be characterized in detail. "

Analysis of our blots reveal that HSJ633 does reduce STAT3 activation by hyper-IL6 (Fig. EV6). Consistent with message by the handling editor, additional biochemical characterization including detailed binding epitope is not required for this manuscript, which focuses on the pharmacological effects of HSJ633 in preterm birth models.

4. My point 5 has not been addressed, I think the samples should be analysed properly and even lower concentrations of IL-6 (less than 1µg/ml) should be tested: "5. The presented blots in Figure 7 are cropped and assembled. To directly compare activation of p38, ERK1/2, AKT and STAT3 all samples have to be analyzed on a single gel with appropriate controls. Moreover, 1 µg/ml IL-6 was used. Typically, concentrations between 1 and 10 ng/ml are sufficient."

Yes, we have addressed your comments for Fig. 8 with the following response: We agree that a single gel may have been preferred. However, during the membrane preparation, the samples were not all placed next to each other to create a cohesive image for publication. Nonetheless, we do provide the editor our membrane figures that show these results. Importantly, despite using a high dosage of IL-6 (1 µg/ml) and of HSJ633 (1µM) ERK1/2, AKT, and p38 were unaffected, whereas STAT3 was inhibited; this *in vitro* pharmacologic aspect is particularly relevant

Specific Editorial and Editorial assistant revisions:

All specific editorial and editorial assistant revision requests have been made; these are all highlighted in yellow in the manuscript (along with reviewer revisions addressed above). Also as requested, a graphical abstract has been added.

Bibliography

Boulanger MJ, Chow DC, Brevnova EE, Garcia KC (2003) Hexameric structure and assembly of the interleukin-6/IL-6 alpha-receptor/gp130 complex. *Science* 300: 2101-2104

Goupil E, Tassy D, Bourguet C, Quiniou C, Wisehart V, Pétrin D, Le Gouill C, Devost D, Zingg HH, Bouvier M *et al* (2010) A novel biased allosteric compound inhibitor of parturition selectively impedes the prostaglandin F2alpha-mediated Rho/ROCK signaling pathway. *J Biol Chem* 285: 25624-25636

Quiniou C, Domínguez-Punaro M, Cloutier F, Erfani A, Ennaciri J, Sivanesan D, Sanchez M, Chognard G, Hou X, Rivera JC *et al* (2014) Specific targeting of the IL-23 receptor, using a novel small peptide noncompetitive antagonist, decreases the inflammatory response. *Am J Physiol Regul Integr Comp Physiol* 307: R1216-1230

Quiniou C, Sapieha P, Lahaie I, Hou X, Brault S, Beauchamp M, Leduc M, Rihakova L, Joyal JS, Nadeau S *et al* (2008) Development of a novel noncompetitive antagonist of IL-1 receptor. *J Immunol* 180: 6977-6987

Rihakova L, Quiniou C, Hamdan FF, Kaul R, Brault S, Hou X, Lahaie I, Sapieha P, Hamel D, Shao Z *et al* (2009) VRQ397 (CRAVKY): a novel noncompetitive V2 receptor antagonist. *Am J Physiol Regul Integr Comp Physiol* 297: R1009-1018

Truong N, Menon R, Richardson L (2023) The Role of Fetal Membranes during Gestation, at Term, and Preterm Labor. *Placenta Reprod Med* 2

14th May 2025

Dear Dr. Chemtob,

Thank you for the submission of your revised manuscript to EMBO Molecular Medicine. I have now had the opportunity to read it and to discuss it with the other members of our editorial team. We agreed that all referee concerns and external advisor's suggestions are adequately addressed. Therefore, I am pleased to inform you that we will be able to accept your manuscript pending the following final amendments:

1) Figures:

- We note that some western blot images are spliced and merged, e.g. Fig. 8B-D and Fig EV6A. Provided source data confirm that the presented images have been spliced from the same western blot membrane. Therefore, please mark all splice sites with a strong white or black line to indicate the merging.

- Please indicate p values in the legend of Figure EV3.

2) Tables: Please upload Table EV1 in editable format e.g. .doc.

3) In the main manuscript file, please do the following:

- In Methods, provide the statement that informed consent was obtained from all human subjects and that experiments conformed to the principles set out in the WMA Declaration of Helsinki and the Department of Health and Human Services Belmont Report. Please indicate this also in the "Author Checklist".

- Remove Table legends.

4) Reagent Table: Please remove example table.

5) Synopsis:

- Please upload synopsis text.

6) As part of the EMBO Publications transparent editorial process initiative (see our Editorial at <http://embomolmed.embopress.org/content/2/9/329>), EMBO Molecular Medicine will publish online a Review Process File (RPF) to accompany accepted manuscripts. This file will be published in conjunction with your paper and will include the anonymous referee reports, your point-by-point response and all pertinent correspondence relating to the manuscript. Let us know whether you agree with the publication of the RPF and as here, if you want to remove or not any figures from it prior to publication. Please note that the Authors checklist will be published at the end of the RPF.

7) Please provide a point-by-point letter to my comments (as Word file).

I look forward to reading a new revised version of your manuscript as soon as possible.

Yours sincerely,

Zeljko Durdevic

Zeljko Durdevic
Senior Editor
EMBO Molecular Medicine

*** Instructions to submit your revised manuscript ***

- 1) a .docx formatted version of the manuscript text (including Figure legends and tables)
 - 2) Separate figure files*
 - 3) supplemental information as Expanded View and/or Appendix. Please carefully check the authors guidelines for formatting Expanded view and Appendix figures and tables at <https://www.embopress.org/page/journal/17574684/authorguide#expandedview>
 - 4) a letter INCLUDING the reviewer's reports and your detailed responses to their comments (as Word file).
 - 5) The paper explained: EMBO Molecular Medicine articles are accompanied by a summary of the articles to emphasize the major findings in the paper and their medical implications for the non-specialist reader. Please provide a draft summary of your article highlighting
 - the medical issue you are addressing,
 - the results obtained and
 - their clinical impact.This may be edited to ensure that readers understand the significance and context of the research. Please refer to any of our published articles for an example.
 - 6) Author contributions: the contribution of every author must be detailed in a separate section.
 - 7) EMBO Molecular Medicine now requires a complete author checklist (<https://www.embopress.org/page/journal/17574684/authorguide>) to be submitted with all revised manuscripts. Please use the checklist as guideline for the sort of information we need WITHIN the manuscript. The checklist should only be filled with page numbers were the information can be found. This is particularly important for animal reporting, antibody dilutions (missing) and exact values and n that should be indicted instead of a range.
 - 8) Every published paper now includes a 'Synopsis' to further enhance discoverability. Synopses are displayed on the journal webpage and are freely accessible to all readers. They include a short stand first (maximum of 300 characters, including space) as well as 2-5 one sentence bullet points that summarise the paper. Please write the bullet points to summarise the key NEW findings. They should be designed to be complementary to the abstract - i.e. not repeat the same text. We encourage inclusion of key acronyms and quantitative information (maximum of 30 words / bullet point). Please use the passive voice. Please attach these in a separate file or send them by email, we will incorporate them accordingly.
- You are also welcome to suggest a striking image or visual abstract to illustrate your article. If you do please provide a jpeg file 550 px-wide x 300-600px high.
- 9) A Conflict of Interest statement should be provided in the main text
 - 10) Please note that we now mandate that all corresponding authors list an ORCID digital identifier. This takes <90 seconds to complete. We encourage all authors to supply an ORCID identifier, which will be linked to their name for unambiguous name identification.

Currently, our records indicate that there is no ORCID associated with your account.

Please click the link below to provide an ORCID:

Link Not Available

- 11) Include a Reagents and Tools Table as part of the Methods section, which can be downloaded from our author guidelines (<https://www.embopress.org/page/journal/17574684/authorguide#structuredmethods>)

Photos 400-800 DPI

Figures are not edited by the production team. All lettering should be the same size and style; figure panels should be indicated by capital letters (A, B, C etc). Gridlines are not allowed except for log plots. Figures should be numbered in the order of their

appearance in the text with Arabic numerals. Each Figure must have a separate legend and a caption is needed for each panel.

*Additional important information regarding figures and illustrations can be found at <https://bit.ly/EMBOPressFigurePreparationGuideline>. See also figure legend preparation guidelines: <https://www.embopress.org/page/journal/17574684/authorguide#figureformat>

The authors addressed the remaining editorial issues.

20th May 2025

Dear Dr. Chemtob,

We are pleased to inform you that your manuscript is accepted for publication and is now being sent to our publisher to be included in the next available issue of EMBO Molecular Medicine.

Zeljko Durdevic
Senior Editor
EMBO Molecular Medicine
